# Learning Utilities from Demonstrations in Markov Decision Processes

**Filippo Lazzati** [1]   **Alberto Maria Metelli** [1]

## Abstract

Although it is well-known that humans commonly engage in *risk-sensitive* behaviors in the presence of stochasticity, most Inverse Reinforcement Learning (IRL) models assume a *risk-neutral* agent. As such, beyond $(i)$ introducing model misspecification, $(ii)$ they do not permit direct inference of the risk attitude of the observed agent, which can be useful in many applications. In this paper, we propose a novel model of behavior to cope with these issues. By allowing for risk sensitivity, our model alleviates $(i)$, and by explicitly representing risk attitudes through (learnable) *utility* functions, it solves $(ii)$. Then, we characterize the partial identifiability of an agent's utility under the new model and note that demonstrations from multiple environments mitigate the problem. We devise two provably-efficient algorithms for learning utilities in a finite-data regime, and we conclude with some proof-of-concept experiments to validate *both* our model and our algorithms.

## 1. Introduction

The ultimate goal of Artificial Intelligence (AI) is to construct artificial rational autonomous agents (Russell & Norvig, 2010). Such agents will interact with each other and with human beings to achieve the tasks that *we* assign to them. In this vision, a crucial feature is being able to correctly model the observed behavior of other agents. This allows a variety of applications: *descriptive*, to understand the intent of the observed agent (Russell, 1998), *predictive*, to anticipate the behavior of the observed agent (potentially in new scenarios) (Arora & Doshi, 2021), and *normative*, to imitate the observed agent because they are behaving in the "right way" (Osa et al., 2018).

Nowadays, Inverse Reinforcement Learning (IRL) pro-

[1]Politecnico di Milano, Milan, Italy. Correspondence to: Filippo Lazzati <filippo.lazzati@polimi.it>.

*Proceedings of the $42^{nd}$ International Conference on Machine Learning*, Vancouver, Canada. PMLR 267, 2025. Copyright 2025 by the author(s).

vides the most popular and powerful models of the behavior of the observed agent, named "expert". IRL models assume the existence of a reward function that *rationalizes* the expert's behavior and differ from each other based on the specific assumption of how the expert behaves based on the given reward. For instance, Ng & Russell (2000) considers the expert as playing an optimal policy, Poiani et al. (2024) considers an $\epsilon$-optimal policy, Malik et al. (2021) considers an optimal policy satisfying some constraints, Ziebart (2010); Fu et al. (2017) assume that the expert plays actions proportionally to their (soft) Q-functions, and Ramachandran & Amir (2007) assumes this probability to depend on the optimal advantage function.

These IRL models represent the expert as a *risk-neutral* agent, i.e., an agent interested in maximizing the *expected* return. However, there are many scenarios in which rational agents and humans adopt risk-sensitive strategies in the presence of stochasticity, like finance (Föllmer & Schied, 2016), revenue management (Barz, 2007), driving (Bernhard et al., 2019), and many other choice problems (Kahneman & Tversky, 1979; Kreps, 1988). In these settings, agents are not only interested in the *expected* return, but in its full *distribution* (Bellemare et al., 2023). Thus, IRL models incur in *misspecification*, that can crucially affect the descriptive, predictive, and normative power of the inferred reward function (Skalse & Abate, 2024).

In this context, in addition to misspecification, another issue of IRL models is that they do not *explicitly* represent the risk attitude of the expert, which is only indirectly captured by the reward function. We desire two different representations for the *task* of the expert (through a reward) and for its *risk attitude* (e.g., through a utility function), analogously to what is done in Inverse Constrained Reinforcement Learning (ICRL) (Malik et al., 2021). Here, the behavior of the expert is described by two parameters, a reward for modelling the task, and a cost for modelling the constraints. In this way, the reward is more easily interpretable since it does not have to capture both the task and the constraints, and, also, we can use the learned cost for performing new tasks safely (Kim et al., 2023). For these reasons, if we were able to directly learn the risk attitude of the expert *separately* from its reward function, then we could more easily understand its intent and anticipate its choices in new, unseen, scenarios (Kreps, 1988).

**Contributions.** In this paper, we introduce a new risk-sensitive model of behavior that encodes the risk attitude of an agent with a utility function. Thanks to this model, we will show that it is possible to overcome the limitations mentioned above. Our main contributions are:

- We present a new simple yet powerful *model* of behavior in Markov Decision Processes (MDPs) that rationalizes non-Markovian demonstrations (Section 3).
- We formulate *Utility Learning* (UL) as the problem of inferring the risk attitude of an agent under the new model of behavior, we characterise the partial identifiability of its utility, and we show that demonstrations in multiple environments alleviate the issue (Section 4).
- We introduce **CATY-UL** and **TRACTOR-UL**, two novel algorithms for solving the UL problem with finite data in a provably-efficient manner (Section 5).
- We conclude with proof-of-concept *experiments* that serve as an empirical validation of both the proposed model and the presented algorithms. (Section 6).

The proofs of all results are reported in Appendix C-E.

## 2. Preliminaries

The main paper's notation is below. Additional notation for the supplemental is in Appendix B.

**Notation.** For any $N \in \mathbb{N}$, we write $[\![N]\!] := \{1, \ldots, N\}$. Given set $\mathcal{X}$, we denote by $\Delta^{\mathcal{X}}$ the probability simplex on $\mathcal{X}$. Given compact $\mathcal{X} \subseteq \mathbb{R}^d, y \in \mathbb{R}^d$, we define $\Pi_{\mathcal{X}}(y) := \arg\min_{x \in \mathcal{X}} \|y - x\|_2$. A real-valued function $f : \mathbb{R} \to \mathbb{R}$ is *L-Lipschitz* if, for all $x, y \in \mathbb{R}$, we have $|f(x) - f(y)| \leq L|x - y|$. $f$ is *increasing* if, for all $x < y \in \mathbb{R}$, it holds $f(x) \leq f(y)$, and it is *strictly-increasing* if $f(x) < f(y)$. The probability distribution that puts all its mass on $z \in \mathbb{R}$ is denoted by $\delta_z$ and is called the *Dirac delta*. We represent distributions on finite support as mixtures of Dirac deltas.

**Markov Decision Processes (MDPs).** A tabular episodic Markov Decision Process (MDP) (Puterman, 1994) is a tuple $\mathcal{M} = (\mathcal{S}, \mathcal{A}, H, s_0, p, r)$, where $\mathcal{S}$ and $\mathcal{A}$ are the finite state ($S := |\mathcal{S}|$) and action ($A := |\mathcal{A}|$) spaces, $H$ is the time horizon, $s_0 \in \mathcal{S}$ is the initial state, $p : \mathcal{S} \times \mathcal{A} \times [\![H]\!] \to \Delta^{\mathcal{S}}$ is the transition model, and $r : \mathcal{S} \times \mathcal{A} \times [\![H]\!] \to [0, 1]$ is the *deterministic* reward function. The interaction of an agent with $\mathcal{M}$ generates trajectories. Let $\Omega_h := (\mathcal{S} \times \mathcal{A})^{h-1} \times \mathcal{S}$ be the set of state-action trajectories of length $h$ for all $h \in [\![H + 1]\!]$, and $\Omega := \Omega_{H+1}$. A deterministic *non-Markovian* policy $\pi = \{\pi_h\}_{h \in [\![H]\!]}$ is a sequence of functions $\pi_h : \Omega_h \to \mathcal{A}$ that, given the history up to stage $h$, i.e., $\omega = (s_1, a_1, \ldots, s_{h-1}, a_{h-1}, s_h) \in \Omega_h$, prescribes an action. A *Markovian* policy $\pi = \{\pi_h\}_{h \in [\![H]\!]}$ is a sequence of functions $\pi_h : \mathcal{S} \to \mathcal{A}$ that depend on the current state only. We use $g : \bigcup_{h \in \{2, \ldots, H+1\}} \Omega_h \to [0, H]$ to denote the return of a (partial) trajectory $\omega \in \Omega_h$, i.e.,

$g(\omega) := \sum_{h' \in [\![h-1]\!]} r_{h'}(s_{h'}, a_{h'})$. With abuse of notation, we denote by $\mathbb{P}_{p,r,\pi}$ the probability distribution over trajectories of any length induced by $\pi$ in $\mathcal{M}$ (we omit $s_0$ for simplicity), and by $\mathbb{E}_{p,r,\pi}$ the expectation w.r.t. $\mathbb{P}_{p,r,\pi}$. We define the *return distribution* $\eta^{p,r,\pi} \in \Delta^{[0,H]}$ of policy $\pi$ as $\eta^{p,r,\pi}(y) := \sum_{\omega \in \Omega: g(\omega) = y} \mathbb{P}_{p,r,\pi}(\omega)$ for all $y \in [0, H]$. The set of possible returns at $h \in [\![H + 1]\!]$ is $\mathcal{G}_h^{p,r} := \{y \in [0, h-1] \mid \exists \omega \in \Omega_h, \exists \pi : g(\omega) = y \wedge \mathbb{P}_{p,r,\pi}(\omega) > 0\}$, and $\mathcal{G}^{p,r} := \mathcal{G}_{H+1}^{p,r}$. We remark that $\mathcal{G}_h^{p,r}$ has finite cardinality for all $h$. The performance of policy $\pi$ is given by $J^\pi(p, r) := \mathbb{E}_{p,r,\pi}[\sum_{h=1}^H r_h(s_h, a_h)]$, and note that $J^\pi(p, r) = \mathbb{E}_{G \sim \eta^{p,r,\pi}}[G]$. We define the optimal performance as $J^*(p, r) := \max_\pi J^\pi(p, r)$, and the optimal policy as $\pi^* \in \arg\max_\pi J^\pi(p, r)$.

**Risk-Sensitive Markov Decision Processes (RS-MDPs).** A Risk-Sensitive Markov Decision Process (RS-MDP) (Wu & Xu, 2023) is a pair $\mathcal{M}_U := (\mathcal{M}, U)$, where $\mathcal{M} = (\mathcal{S}, \mathcal{A}, H, s_0, p, r)$ is an MDP, and $U \in \mathfrak{U}$ is a utility function in set $\mathfrak{U} := \{U' : [0, H] \to [0, H] \mid U'(0) = 0, U'(H) = H \wedge U'$ is strictly-increasing and continuous$\}$. Differently from Wu & Xu (2023), w.l.o.g., our utilities satisfy $U(H) = H$ to settle the scale. The interaction with $\mathcal{M}_U$ is the same as with $\mathcal{M}$, and the notation described earlier still applies, except for the performance of policies. The performance of policy $\pi$ is $J^\pi(U; p, r) := \mathbb{E}_{p,r,\pi}[U(\sum_{h=1}^H r_h(s_h, a_h))]$, and note that $J^\pi(U; p, r) = \mathbb{E}_{G \sim \eta^{p,r,\pi}}[U(G)]$. We define the optimal performance as $J^*(U; p, r) := \max_\pi J^\pi(U; p, r)$, the optimal policy as $\pi^* \in \arg\max_\pi J^\pi(U; p, r)$, and the set of optimal policies for $\mathcal{M}_U$ as $\Pi_{p,r}^*(U)$.

**Enlarged state space approach.** In MDPs, there always exists a *Markovian* optimal policy (Puterman, 1994), but in RS-MDPs this does not hold. The *enlarged state space approach* (Wu & Xu, 2023) is a method, proposed by Bäuerle & Rieder (2014), to compute an optimal policy in a RS-MDP. Given RS-MDP $\mathcal{M}_U = (\mathcal{S}, \mathcal{A}, H, s_0, p, r, U)$, we construct the *enlarged* state space MDP $\mathfrak{E}[\mathcal{M}_U] = (\mathcal{S}', \mathcal{A}, H, (s_0, 0), \mathfrak{p}, \mathfrak{r})$, with a different state space $\mathcal{S}' = \mathcal{S} \times \mathcal{G}_h^{p,r}$ at every $h$.[1] For every $h \in [\![H]\!]$ and $(s, y, a) \in \mathcal{S} \times \mathcal{G}_h^{p,r} \times \mathcal{A}$, the reward function $\mathfrak{r}$ is $\mathfrak{r}_h(s, y, a) = U(y + r_h(s, a))\mathbb{1}\{h = H\}$, while the dynamics $\mathfrak{p}$ assigns to the next state $(s', y') \in \mathcal{S} \times \mathcal{G}_{h+1}^{p,r}$ the probability: $\mathfrak{p}_h(s', y'|s, y, a) := p_h(s'|s, a)\mathbb{1}\{y' = y + r_h(s, a)\}$. In words, the state space is enlarged with a component that keeps track of the cumulative reward in the original RS-MDP, and the reward $\mathfrak{r}$, bounded in $[0, H]$, provides the utility of the accumulated reward at the end of the episode. A Markovian policy $\psi = \{\psi_h\}_{h \in [\![H]\!]}$ for $\mathfrak{E}[\mathcal{M}_U]$ is a sequence of mappings $\psi_h : \mathcal{S} \times \mathcal{G}_h^{p,r} \to \mathcal{A}$. Being an MDP,

---

[1] Actually, Bäuerle & Rieder (2014) use state space $\mathcal{S} \times \mathbb{R}_{\geq 0}$, while Wu & Xu (2023) use $\mathcal{S} \times [h - 1]$ for all $h \in [\![H]\!]$. Instead, we consider sets $\mathcal{S} \times \{\mathcal{G}_h^{p,r}\}_h$ to capture the minimal size required.

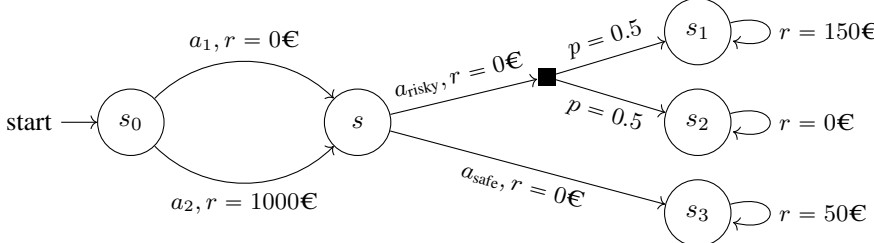

*Figure 1.* The MDP considered in Example 3.1.

we adopt for $\mathfrak{E}[\mathcal{M}_U]$ the same notation presented earlier for MDPs, by replacing $p, r, \pi$ with $\mathfrak{p}, \mathfrak{r}, \psi$. Let $\psi^*$ be the optimal *Markovian* policy for $\mathfrak{E}[\mathcal{M}_U]$. Then, Theorem 3.1 of Bäuerle & Rieder (2014) shows that the (non-Markovian) policy $\pi^*$, defined for all $h \in \{2, \ldots, H\}$ and $\omega \in \Omega_h$ as $\pi_h^*(\omega) := \psi_h^*(s_h, \sum_{h' \in [\![h-1]\!]} r_{h'}(s_{h'}, a_{h'}))$, and $\pi_1^*(s_0) = \psi_1^*(s_0, 0)$, is optimal for $\mathcal{M}_U$.

**Inverse Reinforcement Learning (IRL).** In IRL we are given demonstrations of behavior from the expert's policy $\pi^E$, and the goal is to recover the reward of the expert $r^E$ (Russell, 1998). As explained in Section 1, a *model of behavior* describes how the expert's policy $\pi^E$ is generated from $r^E$. A model suffers from *partial identifiability* if the knowledge of $\pi^E$ does not permit to recover $r^E$ (almost) uniquely (Cao et al., 2021; Metelli et al., 2021).

**Miscellaneous.** For $L > 0$, we write $\mathfrak{U}_L := \{U \in \mathfrak{U} \mid U \text{ is } L\text{-Lipschitz}\}$. For any finite set $\mathcal{X} \subseteq [0, H]$ we define $\overline{\mathfrak{U}}^{\mathcal{X}} := \{\overline{U} \in [0, H]^{|\mathcal{X}|} \mid \exists U \in \mathfrak{U}, \forall x \in \mathcal{X} : \overline{U}(x) = U(x)\}$, and $\overline{\mathfrak{U}}_L^{\mathcal{X}} := \{\overline{U} \in \overline{\mathfrak{U}}^{\mathcal{X}} \mid \exists U \in \mathfrak{U}_L, \forall x \in \mathcal{X} : \overline{U}(x) = U(x)\}$. We will denote by $\mathcal{M}_{\overline{U}}$ some RS-MDPs with $\overline{U} \in \overline{\mathfrak{U}}^{\mathcal{X}}$.

## 3. A New Model of Behavior

We aim to devise a realistic model of behavior for humans and rational agents in MDPs that complies with their sensitivity to risk. In fact, due to the stochasticity of the environment, they are likely to behave in a risk-sensitive manner. Our insight is that risk-sensitivity in MDPs gives rise to *non-Markovian* policies for both rational agents (see Bellemare et al. (2023)) and humans:

**Example 3.1.** *In the MDP of Fig. 1, we expect most people to decide what action to play in state $s$ depending on the amount of reward earned so far, since, intuitively, it makes more sense to take the risky action $a_{risky}$, that sometimes gives a large return (150€) but sometimes gives nothing (0€), when we are guaranteed to have at least 1000€ in our wallet (i.e., we have reached $s$ from $a_2$), while it may be better to take the safe action $a_{safe}$, that gives 50€ for sure, if we reached $s$ with no reward (i.e., from $a_1$). This kind of behavior is known as "decreasing" risk-aversion. (Pratt, 1964; Kreps, 1988; Wakker, 2010).*

In short, demonstrations of behavior from risk-sensitive agents in MDPs are likely to be collected by *non-Markovian* policies, whose dependency on the past history is restricted to the *amount of reward* collected so far. However, none of the existing IRL models of behavior (see Sections 1 and 7) contemplate non-Markovian policies, and, thus, they result in misspecification.[2]

For these reasons, we introduce a new model of behavior that contemplates non-Markovian policies. Given demonstrations from the expert's policy $\pi^E$ in an environment $(\mathcal{S}, \mathcal{A}, H, s_0, p)$, we assume the existence of a reward function $r^E$ and a utility function $U^E \in \mathfrak{U}$ such that:

$$\pi^E \in \arg\max_{\pi} \mathbb{E}_{p, r^E, \pi}\left[U^E\left(\sum_{h=1}^{H} r_h^E(s_h, a_h)\right)\right], \quad (1)$$

i.e., we model the expert as an *optimal agent in a RS-MDP*. The reward $r^E$ aims to capture the task of the expert, while the utility $U^E$ represents its risk attitude. Intuitively, if $p$ is deterministic, then the trajectories with the largest returns under $r^E$ are preferred. However, in presence of stochasticity, the utility $U^E$ associates weights to the returns of the trajectories to represent their true "values" for the expert. If $U^E$ is linear, then $\arg\max_{\pi} J^{\pi}(U^E; p, r^E) = \arg\max_{\pi} J^{\pi}(p, r^E)$ and the expert values each trajectory by its return under $r^E$, i.e., it is *risk-neutral*. However, if $U^E$ is convex (resp. concave), then the expert amplifies (resp. attenuates) the desirability of high-return trajectories, so that it will accept even more (resp. less) variance to play them. In this case, $U^E$ represents a *risk-seeking* (resp. *risk-averse*) expert (Kreps, 1988; Bäuerle & Rieder, 2014).

There are many arguments that support this model:

1. it generalizes the IRL model of Ng & Russell (2000), that we get if the expert is risk-neutral ($U^E$ is linear);
2. it is justified by the famous expected utility theory (von Neumann & Morgenstern, 1947), as we can interpret each policy $\pi$ as a choice that induces a lottery $\eta^{p, r^E, \pi}$ over the set of prizes (i.e., returns) $\mathcal{G}^{p, r^E}$;

---

[2]Re-modelling the MDP including the sum of the past rewards into the state would make the demonstrated policy Markovian, but, as explained in Appendix C.1, it would create various issues like a state space with a size exponential in the horizon.

3. it contemplates the existence of non-Markovian policies that depend only on the cumulative reward so far (see Bäuerle & Rieder (2014));

4. the corresponding planning problem enjoys practical tractability (Wu & Xu, 2023);

5. $U^E$ can be learned efficiently, as we show in Section 5.

**Some considerations.** If $U^E$ is linear, Eq. (1) admits a *Markovian* optimal policy (Puterman, 1994). Otherwise, the more $U^E$ deviates from linearity, the more non-Markovian policies *may* outperform Markovian policies:

**Proposition 3.1.** *There exists a RS-MDP in which the difference between the optimal performance and the performance of the best Markovian policy is* 0.5.

Next, note that, in absence of stochasticity, $U^E$ plays no role, and Eq. (1) traces back to risk-neutral behavior, as desired:

**Proposition 3.2.** *If $p$ is deterministic, then* $\arg\max_\pi J^\pi(U^E; p, r^E) = \arg\max_\pi J^\pi(p, r^E)$.

We remark that, by complying with non-Markovian policies, our model of behavior suffers from less misspecification than common IRL models. Moreover, by using $U^E$, it permits to learn a succinct and transferrable representation of the risk attitude of the expert, as we shall see later.

## 4. Utility Learning

In this and in the following sections, we focus on the problem of learning the utility $U^E$ of the expert under the assumption that it behaves as in Eq. (1). Here, we assume that the expert's policy $\pi^E$ and the dynamics $s_0, p$ are known, while in Section 5 we will estimate them from finite data.

**Problem definition and partial identifiability.** Given demonstrations collected by a policy $\pi^E$ satisfying Eq. (1), three different learning problems arise:

1. given $r^E$, learn $U^E$;
2. given $U^E$, learn $r^E$;
3. learn both $r^E$ and $U^E$.

Problem 3 is the most interesting and challenging, because it makes the least assumptions, while Problem 2, i.e., IRL, has been extensively studied in literature when $U^E$ is linear (Ng & Russell, 2000). In this paper, in analogy to ICRL (Malik et al., 2021) where the goal is to learn the constraints when $r^E$ is known, we focus on Problem 1 because it has relevant applications per se (see later in this section) and because it represents a significant step toward solving Problem 3. Thus, we will consider $r^E$ to be given and denote it with $r$ for simplicity. Let us formalize Problem 1:

**Definition 4.1** (Utility Learning (UL)). *Let $\mathcal{M} = (\mathcal{S}, \mathcal{A}, H, s_0, p, r)$ be an MDP and $\pi^E$ a (potentially non-Markovian) policy. Under the assumption that $\pi^E$ satisfies*

*Eq.* (1) *in $\mathcal{M}$ for some unknown $U^E$, the goal of* Utility Learning (UL) *is to find $U^E$.*

Does the knowledge of $\pi^E$ and $\mathcal{M}$ suffice to *uniquely* identify $U^E$? Analogously to IRL (Cao et al., 2021) and ICRL (Kim et al., 2023), the answer is *negative*, as shown in the following example (details in Appendix D).

**Example 4.1.** *Consider the MDP $\mathcal{M}$ in Fig. 2 (left), where $H=2, r_1(s_0, a_1)=1, r_1(s_0, a_2)=0.5$. Let the expert's policy $\pi^E$ prescribe $a_1$ in $s_0$. Then, all the utility functions $U \in \mathfrak{U}$ that take on values in the blue region of Fig. 2 (middle) for returns $G=1, G=1.5$, make $\pi^E$ optimal in $\mathcal{M}_U$.*

Simply put, in UL, the only information available on the unknown utility $U^E$ is that it belongs to $\mathfrak{U}$ and it makes $\pi^E$ an optimal policy in the corresponding RS-MDP. Since Example 4.1 shows that, in general, there is a set of utilities $U \in \mathfrak{U}$ satisfying this condition, we realize that $U^E$ is partially identifiable. Analogously to Metelli et al. (2021; 2023), we call such set the *feasible set* of utilities "compatible" with $\pi^E$ in $\mathcal{M}$:[3]

$$\mathcal{U}_{p,r,\pi^E} := \{U \in \mathfrak{U} \,|\, J^{\pi^E}(U; p, r) = J^*(U; p, r)\}. \quad (2)$$

**Applications.** If we knew the risk attitude of the expert, i.e., its utility $U^E$ in our model, then we could use it for applications like $(i)$ *predicting* the behavior of the expert in a new environment, $(ii)$ *imitating* the expert, or $(iii)$ *assessing* how valuable a certain behavior is from the viewpoint of the expert. UL represents an appealing problem setting for learning $U^E$ from demonstrations of behavior. However, due to partial identifiability, no learning algorithm can recover $U^E$, but, at best, it can find an *arbitrary* utility in the feasible set $\mathcal{U}_{p,r,\pi^E}$. Is this ambiguity tolerated by the applications $(i), (ii)$, and $(iii)$ above? In other words, we are interested in understanding whether *all* the utilities contained into $\mathcal{U}_{p,r,\pi^E}$ can be used in place of the true $U^E$ without incurring in large errors.[4] Unfortunately, the following propositions answer *negatively* for all $(i), (ii)$, and $(iii)$. Nevertheless, Proposition 4.5 shows that the availability of expert demonstrations from multiple environments is a possible mitigation for the issue.

Let us begin with $(i)$. We say that a utility $U$ permits to *predict* the behavior of an agent with utility $U^E$ in a new MDP $\mathcal{M}'$ if $U$ and $U^E$ induce in $\mathcal{M}'$ the same optimal policies. The next two propositions show that if the transition model or the reward function of $\mathcal{M}'$ differ from those of the original MDP $\mathcal{M}$, then there are utilities in the feasible set that get wrong in predicting the behavior of the agent with $U^E$:

**Proposition 4.1.** *There exist two MDPs $\mathcal{M} = (\mathcal{S}, \mathcal{A}, H, s_0, p, r)$, $\mathcal{M}' = (\mathcal{S}, \mathcal{A}, H, s_0, p', r)$, with $p \neq p'$,*

---

[3]In Appendix D we provide a more explicit expression.
[4]Skalse et al. (2023) conduct an analogous study for IRL.

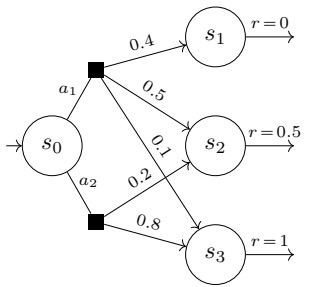 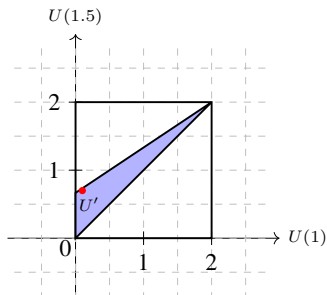 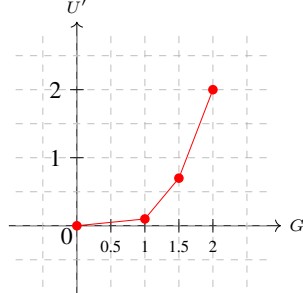

*Figure 2.* (Left) MDP of Example 4.1. (Middle) its feasible set with a sample utility $U'$. (Right) plot of $U'$ with linear interpolation.

for which there exist a policy $\pi^E$ and a pair of utilities $U_1, U_2 \in \mathcal{U}_{p,r,\pi^E}$ such that $\Pi^*_{p',r}(U_1) \cap \Pi^*_{p',r}(U_2) = \{\}$.

**Proposition 4.2.** *There exist two MDPs $\mathcal{M} = (\mathcal{S}, \mathcal{A}, H, s_0, p, r)$, $\mathcal{M}' = (\mathcal{S}, \mathcal{A}, H, s_0, p, r')$, with $r \neq r'$, for which there exist a policy $\pi^E$ and a pair of utilities $U_1, U_2 \in \mathcal{U}_{p,r,\pi^E}$ such that $\Pi^*_{p,r'}(U_1) \cap \Pi^*_{p,r'}(U_2) = \{\}$.*

Consider now $(ii)$. We say that a utility $U$ permits to *imitate* the behavior of an agent with utility $U^E$ if optimizing $U$ provides policies with a large expected utility w.r.t. $U^E$. The reason behind this definition is that, differently from IRL, we wish to imitate also the risk attitude of the observed agent. However, UL does not always allow to perform meaningful imitations:

**Proposition 4.3.** *There exists an MDP $\mathcal{M} = (\mathcal{S}, \mathcal{A}, H, s_0, p, r)$ and a policy $\pi^E$ for which there are utilities $U_1, U_2 \in \mathcal{U}_{p,r,\pi^E}$ such that, for any $\epsilon \geqslant 0$ smaller than some universal constant, there exists a policy $\pi_\epsilon$ such that $J^*(U_1; p, r) - J^{\pi_\epsilon}(U_1; p, r) = \epsilon$ and $J^*(U_2; p, r) - J^{\pi_\epsilon}(U_2; p, r) \geqslant 1$.*

Concerning $(iii)$, we say that $U$ and $U^E$ *assess* behavior in a similar way if, given any policy, they provide close values of performance. The intuition is that the expert values policies based on their alignment with its risk attitude $U^E$ w.r.t. its task $r$. Formally, we want $U$ such that $d^{all}_{p,r}(U^E, U) := \max_\pi |J^\pi(U^E; p, r) - J^\pi(U; p, r)|$ is small (Zhao et al., 2024). Nonetheless, *not* all the utilities in the feasible set are close to each other w.r.t. $d^{all}_{p,r}$:

**Proposition 4.4.** *There exists an MDP $\mathcal{M} = (\mathcal{S}, \mathcal{A}, H, s_0, p, r)$ and a policy $\pi^E$ for which there exists a pair of utilities $U_1, U_2 \in \mathcal{U}_{p,r,\pi^E}$ such that $d^{all}_{p,r}(U_1, U_2) = 1$.*

Propositions 4.1-4.4 tell us that demonstrations of behavior in a *single* MDP do not provide enough information on $U^E$ for applications $(i)$, $(ii)$, and $(iii)$.[5] Thus, we might hope that expert demonstrations in *multiple* environments can help in mitigating this issue, similarly to what

is done in IRL (Amin & Singh, 2016; Cao et al., 2021) and ICRL (Kim et al., 2023). Formally, we extend the UL problem of Definition 4.1 to a set of MDPs $\{\mathcal{M}^i\}_i$, with $\mathcal{M}^i = (\mathcal{S}^i, \mathcal{A}^i, H, s_0^i, p^i, r^i)$,[6] and policies $\{\pi^{E,i}\}_i$ by assuming that there exists a single utility $U^E$ for which Eq. (1) is satisfied for all $i$, i.e., such that $\pi^{E,i}$ is optimal for $\mathcal{M}^i_{U^E}$ for all $i$. In this extended problem setting, the feasible set will be the intersection of all the feasible sets $\mathcal{U}_{p^i, r^i, \pi^{E,i}}$. The following result proves that demonstrations in multiple environments is a *possible* solution to the partial identifiability problem.

**Proposition 4.5.** *Let $\mathcal{S}, \mathcal{A}, H$ be any state space, action space, and horizon, satisfying $S \geqslant 3, A \geqslant 2, H \geqslant 2$, and let $U^E \in \mathfrak{U}$ be any utility. If, for any possible dynamics $s_0, p$ and reward $r$, we are given the set of* all *the deterministic optimal policies of the corresponding RS-MDP $(\mathcal{S}, \mathcal{A}, H, s_0, p, r, U^E)$, then we can* uniquely *identify $U^E$.*

## 5. Online UL with Generative Model

In this section, we present two provably-efficient algorithms for solving the UL problem in a *finite-data* regime.

### 5.1. Problem Setting

We consider a finite-data version of the UL problem with demonstrations in multiple environments presented in Section 4. We let $\{\mathcal{M}^i\}_{i \in \llbracket N \rrbracket}$, with $\mathcal{M}^i = (\mathcal{S}^i, \mathcal{A}^i, H, s_0^i, p^i, r^i)$, be the $N$ MDPs with shared horizon $H$ in which an expert with utility $U^E \in \mathfrak{U}$ provides demonstrations of behavior. Specifically, for each MDP $\mathcal{M}^i$, the expert provides us with a batch dataset $\mathcal{D}^{E,i} = \{(s_1^j, a_1^j, s_2^j, \ldots, s_H^j, a_H^j, s_{H+1}^j)\}_{j \in \llbracket \tau^{E,i} \rrbracket}$ of $\tau^{E,i}$ trajectories collected by executing a policy $\pi^{E,i}$, which is optimal for the RS-MDP $\mathcal{M}^i_{U^E}$. Moreover, for every $\mathcal{M}^i$, we let $\mathcal{S}^i, \mathcal{A}^i, H, s_0^i, r^i$ be known, and we consider access to a *generative sampling model* (Azar et al., 2013) for the transition model $p^i$, which allows us to collect a sample $s' \sim p_h^i(\cdot|s, a)$ from any triple $s, a, h$ at our choice. In

---

[5]Actually, for $(ii)$ only, we can try to learn $\pi^E$ directly without passing through $U^E$, as in behavioral cloning (Osa et al., 2018).

[6]For simplicity, we let $H$ be shared.

short, we assume access to *offline* data for the expert and to *online* data for the environments, as is common in the IRL literature (Ho & Ermon, 2016).

Due to partial identifiability, the feasible set $\bigcap_i \mathcal{U}_{p^i, r^i, \pi^{E,i}}$ might contain multiple utilities. Thus, we will develop two different algorithms, one that aims to classify utilities as inside or outside the feasible set $\bigcap_i \mathcal{U}_{p^i, r^i, \pi^{E,i}}$ (**CATY-UL**, Section 5.3), and the other that aims to compute a single utility contained into it (**TRACTOR-UL**, Section 5.4). Intuitively, **CATY-UL** and **TRACTOR-UL** together permit to fully characterize the feasible set, by, respectively, learning a classification boundary and a representative item. Nevertheless, note that, because of finite data, we will be able to provide guarantees only for a relaxation of the feasible set $\mathcal{U}_\Delta \supseteq \bigcap_i \mathcal{U}_{p^i, r^i, \pi^{E,i}}$ for some $\Delta \geqslant 0$:

$$\mathcal{U}_\Delta := \Big\{ U \in \mathfrak{U} \,|\, \sum_{i \in [\![N]\!]} \overline{\mathcal{C}}_{p^i, r^i, \pi^{E,i}}(U) \leqslant \Delta \Big\}, \quad (3)$$

where $\overline{\mathcal{C}}_{p^i, r^i, \pi^{E,i}}(U)$ quantifies the *(non)compatibility* of utility $U$ with demonstrations from $\pi^{E,i}$ in $\mathcal{M}^i$ (Lazzati et al., 2024a; 2025):

$$\overline{\mathcal{C}}_{p^i, r^i, \pi^{E,i}}(U) := J^*(U; p^i, r^i) - J^{\pi^E}(U; p^i, r^i). \quad (4)$$

Intuitively, $\mathcal{U}_\Delta$ enlarges the feasible set by accepting utilities that make the policies $\pi^{E,i}$ at most $\Delta$-suboptimal. Note that, for $\Delta = 0$, we have $\mathcal{U}_\Delta = \bigcap_i \mathcal{U}_{p^i, r^i, \pi^{E,i}}$.

### 5.2. Challenges and Solution

To develop *practical* algorithms, some dimensionality challenges must be addressed. In this section, we explain how we will face them. In short, our solution permits to work with tractable approximations whose complexity is controlled by a discretization parameter $\epsilon_0 > 0$. First, we need some notation. We use symbol $\mathcal{Y}_h$ to denote an $\epsilon_0$-discretization of the real-valued interval $[0, h-1]$, i.e., we set $\mathcal{Y}_h := \{0, \epsilon_0, 2\epsilon_0, \ldots, \lfloor (h-1)/\epsilon_0 \rfloor \epsilon_0\}$ $\forall h \in [\![H+1]\!]$. Moreover, we introduce ad-hoc symbols $\mathcal{R}, \mathcal{Y}$ for the discretization of intervals $[0, 1]$ and $[0, H]$, namely, we let $\mathcal{R} := \mathcal{Y}_2$, $\mathcal{Y} := \mathcal{Y}_{H+1}$. We also set $d := |\mathcal{Y}| = \lfloor H/\epsilon_0 \rfloor$.

**Working with continuous utilities.** Utilities in $\mathfrak{U}$ are defined over the real-valued interval $[0, H]$, making them incompatible with the finite precision of computers. For this reason, we will consider *discretized* utilities. Formally, we will approximate any $U \in \mathfrak{U}$ with a $d$-dimensional vector $\overline{U} \in \overline{\mathfrak{U}} := \overline{\mathfrak{U}}^{\mathcal{Y}}$, such that $\overline{U}(y) = U(y)$ $\forall y \in \mathcal{Y}$.

**Return distributions.** In MDPs with dynamics $p$ and reward $r$, return distributions $\eta$ are supported on the set of possible returns $\mathcal{G}^{p,r} \subset [0, H]$. However, in general, the size of this set grows exponentially in the horizon $|\mathcal{G}^{p,r}| \propto (SA)^H$, causing any exact representation of $\eta$ to

---

**Algorithm 1 CATY-UL**

---

**Input:** data $\{\mathcal{D}_i^E\}_i$, threshold $\Delta$, utility $U$, discretization $\epsilon_0$, dynamics $\{\widehat{p}^i\}_i$

    `// Discretize U:`
1   $\overline{U}(y) \leftarrow U(y)$    for all $y \in \mathcal{Y}$
2 **for** $i = 1, 2, \ldots, N$ **do**
      `// Estimate` $J^{\pi^{E,i}}(U; p^i, r^i)$:
3      $\widehat{\eta}^{E,i} \leftarrow \text{ERD}(\mathcal{D}_i^E, r^i)$
4      $\widehat{J}^{E,i}(U) \leftarrow \sum_{y \in \mathcal{Y}} \widehat{\eta}^{E,i}(y) \overline{U}(y)$
      `// Estimate` $J^*(U; p^i, r^i)$:
5      $\widehat{J}^{*,i}(U), \_ \leftarrow \text{PLANNING}(\overline{U}, i, \widehat{p}^i)$
      `// Estimate` $\overline{\mathcal{C}}_{p^i, r^i, \pi^{E,i}}(U)$:
6      $\widehat{\mathcal{C}}^i(U) \leftarrow \widehat{J}^{*,i}(U) - \widehat{J}^{E,i}(U)$
7 **end**
8 class $\leftarrow$ True **if** $\sum_{i \in [\![N]\!]} \widehat{\mathcal{C}}^i(U) \leqslant \Delta$ **else** False
9 **Return** class

---

explode even for small $H$. Thus, we adopt a *categorical representation* for return distributions (Bellemare et al., 2023), that, roughly speaking, aims to approximate a distribution on $[0, H]$ with a distribution on $\mathcal{Y} \subset [0, H]$. Formally, given any $\eta \in \Delta^{[0,H]}$ with finite support, its categorical representation $\text{Proj}_{\mathcal{C}}(\eta)$ is the distribution in $\mathcal{Q} := \{q \in \Delta^{[\![d]\!]} | \sum_{j \in [\![d]\!]} q_j \delta_{y_j}\}$ ($y_j$ are the items of $\mathcal{Y}$) obtained through the categorical projection operator $\text{Proj}_{\mathcal{C}}$ (Rowland et al., 2018), reported in Eq. (8)-(9) in Appendix B.

**Optimal policies in RS-MDPs.** To compute an optimal policy in RS-MDP $\mathcal{M}_U$ with dynamics $p$ and reward $r$, the *enlarged state space approach* of Bäuerle & Rieder (2014) presented in Section 2 requires the computation of an optimal policy in the MDP $\mathfrak{E}[\mathcal{M}_U]$, whose state space is $\mathcal{S} \times \mathcal{G}_h^{p,r}$ $\forall h$. Unfortunately, we suffer again from an exponential dependence on the horizon $|\mathcal{G}_h^{p,r}| \propto (SA)^{h-1}$, that causes any exact representation of the state space and of the optimal policy of MDP $\mathfrak{E}[\mathcal{M}_U]$ to explode. To avoid this issue, we adopt the *discretization* approach of Wu & Xu (2023), which, in short, amounts to approximate sets $\mathcal{G}_h^{p,r}$ with $\mathcal{Y}_h$ by simply replacing reward $r$ with the discretized version $\overline{r}$, defined as: $\overline{r}_h(s, a) := \Pi_{\mathcal{R}}[r_h(s, a)]$ for all $s, a, h$. Crucially, since $\overline{r}_h(s, a) \in \mathcal{R}$, then the sum of $h$ rewards $\overline{r}$ belongs to $\mathcal{Y}_{h+1}$. In this manner, the sets of partial returns satisfy $\mathcal{G}_h^{p,\overline{r}} \subseteq \mathcal{Y}_h \subseteq \mathcal{Y}$ for all $h$, thus, the state space of the enlarged MDP has now a cardinality at most $Sd \leqslant \mathcal{O}(SH/\epsilon_0)$, which is no longer exponential in the horizon. In the following, we will denote by $\overline{r}^i$ the discretized version of reward $r^i$ for all $i \in [\![N]\!]$.

### 5.3. CATY-UL (CompATibilitY for Utility Learning)

The goal of **CATY-UL** (Algorithm 1) is to classify input utilities $U \in \mathfrak{U}$ based on whether they belong to set $\mathcal{U}_\Delta$ or not for some $\Delta \geqslant 0$. We implement it using two different subroutines (Lazzati et al., 2024a; 2025). First, Algo-

rithm 3 (reported in Appendix E for its simplicity) actively explores the $N$ environments $\mathcal{M}^i$ *uniformly*, by collecting $\tau^i$ samples from each transition model $p^i$, and uses these samples to construct estimates $\widehat{p}^i$. Next, Algorithm 1 uses these estimates along with the expert's data $\mathcal{D}^{E,i}$ to classify any input utility $U \in \mathfrak{U}$ w.r.t. $\mathcal{U}_\Delta$. To perform the classification, based on Eq. (3), **CATY-UL** computes estimates $\widehat{\mathcal{C}}^i(U) \approx \mathcal{C}_{p^i,r^i,\pi^{E,i}}(U)$ for all $i \in [\![N]\!]$, and, then, it outputs whether $\sum_{i \in [\![N]\!]} \widehat{\mathcal{C}}^i(U) \leqslant \Delta$ (see Line 8). To compute $\widehat{\mathcal{C}}^i(U)$, driven by Eq. (4), **CATY-UL** computes two separate estimates $\widehat{J}^{E,i}(U) \approx J^{\pi^{E,i}}(U; p^i, r^i)$ (Lines 3-4) and $\widehat{J}^{*,i}(U) \approx J^*(U; p^i, r^i)$ (Line 5), and then combines them (Line 6). Specifically, the ERD (Estimate Return Distribution) subroutine (Algorithm 5) permits to construct an estimate $\widehat{\eta}^{E,i}$ of the categorical projection $\text{Proj}_\mathcal{C}(\eta^{p^i,r^i,\pi^{E,i}})$ of the expert's return distribution $\eta^{p^i,r^i,\pi^{E,i}}$ from dataset $\mathcal{D}^{E,i}$, which is used at Line 4 to compute $\widehat{J}^{E,i}(U)$. Instead, quantity $\widehat{J}^{*,i}(U)$ is calculated at Line 5 as the optimal performance in the RS-MDP $(\mathcal{S}^i, \mathcal{A}^i, H, s_0^i, \widehat{p}^i, \overline{r}^i, \overline{U})$, which is computed through value iteration (Puterman, 1994) in the corresponding *enlarged state space MDP* using the PLANNING subroutine (Algorithm 4). **CATY-UL** enjoys the following guarantee for $L$-Lipschitz input utilities:

**Theorem 5.1.** *Let $L > 0$, $\epsilon, \delta \in (0,1)$, and let $\mathcal{U} \subseteq \mathfrak{U}_L$ be the set of utilities to classify. For all $i \in [\![N]\!]$, in case $|\mathcal{U}| = 1$, let the number of samples satisfy:*

$$\tau^{E,i} \geqslant \widetilde{\mathcal{O}}\Big(\frac{N^2 H^2}{\epsilon^2} \log \frac{N}{\delta}\Big), \ \tau^i \geqslant \widetilde{\mathcal{O}}\Big(\frac{N^2 SAH^4}{\epsilon^2} \log \frac{SAHNL}{\delta\epsilon}\Big).$$

*Otherwise, if $|\mathcal{U}| > 1$, let the number of samples satisfy:*

$$\begin{aligned} \tau^{E,i} &\geqslant \widetilde{\mathcal{O}}\Big(\frac{N^4 H^4 L^2}{\epsilon^4} \log \frac{HNL}{\delta\epsilon}\Big), \\ \tau^i &\geqslant \widetilde{\mathcal{O}}\Big(\frac{N^2 SAH^5}{\epsilon^2}\Big(S + \log \frac{SAHN}{\delta}\Big)\Big). \end{aligned} \quad (5)$$

*Then, setting $\epsilon_0 = \epsilon^2/(72HL^2N^2)$, w.p. at least $1 - \delta$, for any $\Delta \geqslant 0$, **CATY-UL** correctly classifies all the $U \in \mathcal{U}$ lying inside $\mathcal{U}_{\Delta-\epsilon}$ or outside $\mathcal{U}_{\Delta+\epsilon}$.*

Roughly speaking, this theorem says that, for a number of samples independent of $\Delta$, **CATY-UL** correctly recognizes all the utilities in $\mathcal{U}_{\Delta-\epsilon} \subseteq \mathcal{U}_\Delta$ and outside $\mathcal{U}_{\Delta+\epsilon} \supseteq \mathcal{U}_\Delta$ as, respectively, inside and outside set $\mathcal{U}_\Delta$. Intuitively, the Lipschitzianity assumption is necessary for approximating functions $U \in \mathcal{U}$ with vectors in $\overline{\mathfrak{U}}$. We remark that, if $|\mathcal{U}| = 1$, then $\propto S$ queries to the generative model suffice. Otherwise, we require $\propto S^2$ samples.

### 5.4. **TRACTOR-UL** (ex**TRACTOR** for **U**tility **L**earning)

For simplicity of presentation, we introduce some notation. For any $L > 0$, let $\overline{\mathfrak{U}}_L := \overline{\mathfrak{U}}_L^{\mathcal{Y}}$, and let $\underline{\mathfrak{U}}, \underline{\mathfrak{U}}_L, \underline{\overline{\mathfrak{U}}}, \underline{\overline{\mathfrak{U}}}_L, \mathcal{U}_\Delta$ be the analogous of, respectively, $\mathfrak{U}, \mathfrak{U}_L, \overline{\mathfrak{U}}, \overline{\mathfrak{U}}_L, \mathcal{U}_\Delta$, but containing *increasing* functions instead of *strictly-increasing*

---

**Algorithm 2** `TRACTOR-UL`

---

**Input:** data $\{\mathcal{D}_i^E\}_i$, parameters $T, K, \alpha, \overline{U}_0$, discretization $\epsilon_0$, dynamics $\{\widehat{p}^i\}_i$

10   $\widehat{\eta}^{E,i} \leftarrow \text{ERD}(\mathcal{D}_i^E, r^i)$     for $i \in [\![N]\!]$

11  **for** $t = 0, 1, \ldots, T - 1$ **do**

12       // Compute distributions $\{\widehat{\eta}_t^i\}_i$:

       **for** $i = 1, 2, \ldots, N$ **do**

13          $\neg, \widehat{\psi}_t^{*,i} \leftarrow \text{PLANNING}(\overline{U}_t, i, \widehat{p}^i)$

14          $\mathcal{D} \leftarrow \text{ROLLOUT}(\widehat{\psi}_t^{*,i}, \widehat{p}^i, \overline{r}^i, i, K)$

15          $\widehat{\eta}_t^i(y) \leftarrow \frac{1}{K} \sum_{G \in \mathcal{D}} \mathbb{1}\{G = y\}, \forall y \in \mathcal{Y}$

16       **end**

      // Update $\overline{U}_{t+1}$:

17       $g_t \leftarrow \sum_{i \in [\![N]\!]} (\widehat{\eta}_t^i - \widehat{\eta}^{E,i})$

18       $\overline{U}_{t+1} \leftarrow \Pi_{\underline{\overline{\mathfrak{U}}}_L}(\overline{U}_t - \alpha g_t)$

19  **end**

20  $\widehat{U} \leftarrow \frac{1}{T} \sum_{t=0}^{T-1} \overline{U}_t$

21  **Return** $\widehat{U}$

---

functions. **TRACTOR-UL** (Algorithm 2) is a more "practical" UL algorithm, in that it aims to compute a utility function contained into the feasible set $\bigcap_i \mathcal{U}_{p^i,r^i,\pi^{E,i}}$. As **CATY-UL**, it comprises an initial exploration phase (Algorithm 3), that collects $\tau^i$ samples to compute estimates $\widehat{p}^i$ of the transition models $p^i$, and an extraction phase (Algorithm 2), where these estimates and the expert's data $\mathcal{D}^{E,i}$ are used to compute a utility (almost) in the feasible set. Specifically, since the utilities $U$ in the feasible set satisfy $\sum_i \overline{\mathcal{C}}_{p^i,r^i,\pi^{E,i}}(U) = 0$, **TRACTOR-UL** aims to find a minimum of function $\sum_i \overline{\mathcal{C}}_{p^i,r^i,\pi^{E,i}}(\cdot)$ over the set $\underline{\mathfrak{U}}_L$. So, starting from an initial $d$-dimensional utility $\overline{U}_0 \in \underline{\mathfrak{U}}_L$, **TRACTOR-UL** computes a sequence $\overline{U}_1, \ldots, \overline{U}_T$ by performing *online projected gradient descent*[7] (Orabona, 2023) in the space of discretized $L$-Lipschitz utilities $\underline{\mathfrak{U}}_L$, where the gradient $g_t$ is computed at Line 17, and the update is carried out at Line 18. With infinite data, the gradient $g_t$ at iteration $t$ would be $\sum_i (\eta^{p^i,r^i,\pi_t^{*,i}} - \eta^{p^i,r^i,\pi^{E,i}})$, where $\pi_t^{*,i}$ is any optimal policy in RS-MDP $\mathcal{M}_{U_t}^i$, in which $U_t \in \underline{\mathfrak{U}}_L$ is any utility satisfying $U_t(y) = \overline{U}_t(y)$ for all $y \in \mathcal{Y}$. In our case, **TRACTOR-UL** uses $\widehat{\eta}^{E,i}$, computed at Line 10 in the same way as in **CATY-UL**, to approximate $\eta^{p^i,r^i,\pi^{E,i}}$, and it uses $\widehat{\eta}_t^i$, computed at Lines 13-15, to approximate $\eta^{p^i,r^i,\pi_t^{*,i}}$ by estimating $\eta^{\widehat{p}^i,\overline{r}^i,\widehat{\pi}_t^{*,i}}$, where $\widehat{\pi}_t^{*,i}$ is the optimal policy for the RS-MDP $(\mathcal{S}^i, \mathcal{A}^i, H, s_0^i, \widehat{p}^i, \overline{r}^i, \overline{U}_t)$. In short, Line 13 computes policy $\widehat{\pi}_t^{*,i}$ through the enlarged state space approach, which is subsequently played in MDP $(\mathcal{S}^i, \mathcal{A}^i, H, s_0^i, \widehat{p}^i, \overline{r}^i)$ (ROLLOUT subroutine, Algorithm 6, Line 14) to construct a dataset $\mathcal{D}$ of $K$ trajectories that is used at Line 15 to compute $\widehat{\eta}_t^i$. **TRACTOR-UL** enjoys the following guarantee:

---

[7]This approach is based on (Syed & Schapire, 2007; Schlaginhaufen & Kamgarpour, 2024).

**Theorem 5.2.** *Let $L > 0$, $\epsilon, \delta \in (0,1)$, and $U^E \in \underline{\mathfrak{U}}_L$. Assume that the projection operator $\Pi_{\overline{\underline{\mathfrak{U}}}_L}$ is implemented exactly. Let the number of samples satisfy Eq. (5). There exist values of $\epsilon_0, K, \alpha, \overline{U}_0$ (see Appendix E.5) such that, if we run* `TRACTOR-UL` *for a number of gradient iterations:*

$$T \geqslant \mathcal{O}\big(N^4 H^4 L^2 / \epsilon^4\big),$$

*then, w.p. at least $1 - \delta$, any utility $U \in \underline{\mathfrak{U}}_L$ such that $U(y) = \widehat{U}(y) \; \forall y \in \mathcal{Y}$ belongs to $\underline{\mathcal{U}}_\epsilon$.*

In other words, with high probability, `TRACTOR-UL` is guaranteed to find a utility $U$ with small $\sum_i \overline{\mathcal{C}}_{p^i, r^i, \pi^{E,i}}(U) \leqslant \epsilon$, i.e., $U$ is close to the feasible set. Note that we consider *increasing* utilities $\underline{\mathfrak{U}}_L$ instead of *strictly-increasing* $\mathfrak{U}_L$ to guarantee the closedness of the set onto which we project. Observe also that assuming that $\Pi_{\overline{\underline{\mathfrak{U}}}_L}$ can be implemented exactly simplifies the theoretical analysis, but, in practice, we are satisfied with approximations that can be computed efficiently since set $\overline{\underline{\mathfrak{U}}}_L$ is made of $\mathcal{O}(H^2/\epsilon_0^2)$ linear constraints (Appendix E.1).

## 6. Numerical Simulations

In this section, we present *proof-of-concept* experiments using data collected from lab members to provide empirical evidence to support both our model and algorithms.

**The data.** We asked to 15 participants to describe the actions they would play in an MDP with horizon $H = 5$ (see Appendix F), at varying of the state, the stage, and the *cumulative reward* collected. The reward has a monetary interpretation. To answer the questions, the participants have been provided with complete information about the MDP.[8]

**Experiment 1 - Validation of the model.** Our model of behavior, presented in Eq. (1), is the first IRL model that contemplates non-Markovian policies. To understand if this new model is more suitable than existing IRL models to describe human behavior in MDPs, we count how many participants to the study exhibited non-Markovian behavior. Intuitively, the more non-Markovianity, the better our model. What we found is that *10 participants out of 15* demonstrated a non-Markovian policy even in this very small environment, providing consistent evidence on the importance of our new model. See Appendix F.3 for additional analysis of our model on this data.

**Experiment 2 - Validation of `TRACTOR-UL`.** To understand how `TRACTOR-UL` performs in practice, we have run it on both the real-world data described earlier and on simulated data. Crucially, the executions on the participants' data reveal that, irrespective of the initial utility $\overline{U}_0$ adopted, the algorithm converges much faster using large values of step size $\alpha$. For instance, as shown in Fig. 3, the

---

[8]The data collected is not personal.

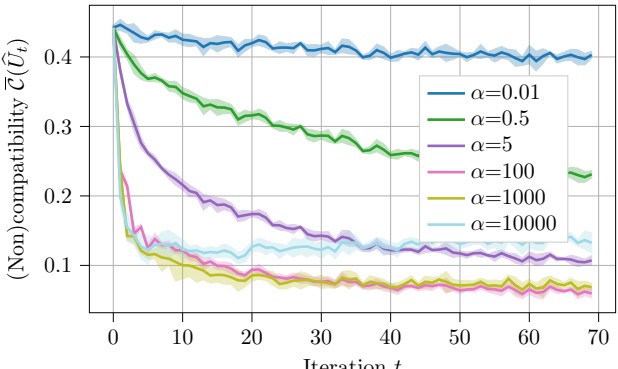

*Figure 3.* Simulations of `TRACTOR-UL` with various step sizes $\alpha$. The shaded regions are the standard deviation over 5 seeds.

best step size for using `TRACTOR-UL` to compute a utility representative of the behavior of participant 10 is $\alpha = 100$. Intuitively, this is explained by the presence of a large number of utilities in the feasible set (since we are considering demonstrations in a single environment $N = 1$), and by the projection step onto $\overline{\underline{\mathfrak{U}}}_L$, that results in small changes of utility even with large steps (see Appendix F.2.2). Next, we have run `TRACTOR-UL` on simulated data to analyze its performance on larger MDPs (increment of $S, A$) and with multiple environments (increment of $N$). We found that the number of gradient iterations necessary to achieve a certain level of (non)compatibility is affected by the increment of $N$, but not of $S, A$, as predicted by Theorem 5.2. However, note that larger $S, A$ require more execution time, because of the value iteration subroutine. Moreover, we observed that, when $N > 1$, the best step size $\alpha$ can be much smaller than $\alpha = 100$. Intuitively, the feasible set contains less utilities now, thus, we need more accurate (smaller) gradient steps to find them. More details on this experiment in Appendix F.2.3.

## 7. Related Work

In *risk-sensitive IRL* (Majumdar et al., 2017), the learner is either provided with the reward of the expert and it must infer some parameters representing its risk attitude, or the learner must infer both the reward and the risk attitude from demonstrations of behavior (Singh et al., 2018; Chen et al., 2019; Ratliff & Mazumdar, 2020; Cheng et al., 2023; Cao et al., 2024). However, these works consider problem settings and models of behavior fairly different from ours. Specifically, Majumdar et al. (2017); Singh et al. (2018); Chen et al. (2019) focus on the so-called "prepare-react" model, which is a model of environment less expressive than an MDP. Instead, Ratliff & Mazumdar (2020); Cheng et al. (2023); Cao et al. (2024) consider models of behavior in which the expert's policy is *Markovian*. More on the

related works in Appendix A.

## 8. Conclusion

In this paper, we proposed a novel risk-aware model of behavior that rationalizes non-Markovian policies in MDPs, and we presented two provably-efficient algorithms for learning the risk attitude of an agent from demonstrations. Interesting directions for future works include extending our algorithms to high-dimensional settings, studying the problem of learning both $r$ and $U$ from demonstrations, and exploring new methods to alleviate the partial identifiability.

## Acknowledgements

AI4REALNET has received funding from European Union's Horizon Europe Research and Innovation programme under the Grant Agreement No 101119527. Views and opinions expressed are however those of the author(s) only and do not necessarily reflect those of the European Union. Neither the European Union nor the granting authority can be held responsible for them.

Funded by the European Union - Next Generation EU within the project NRPP M4C2, Investment 1.,3 DD. 341 - 15 march 2022 - FAIR - Future Artificial Intelligence Research - Spoke 4 - PE00000013 - D53C22002380006.

## Impact Statement

This paper presents work whose goal is to advance the field of Machine Learning. There are many potential societal consequences of our work, none of which we feel must be specifically highlighted here.

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

# A. More on Related Work

We describe here more in detail the most relevant related works. First, we describe IRL papers with risk, i.e., those works that consider MDPs, and try to learn either the reward function or the utility or both. Next, we analyze the works that aim to learn the risk attitude (i.e., a utility function) from demonstrations of behavior (potentially in problems other than MDPs). Finally, we present other connected works.

**Inverse Reinforcement Learning with risk.**  Majumdar et al. (2017) introduces the risk-sensitive IRL problem in decision problems different from MDPs. The authors analyze two settings, one in which the expert takes a single decision, and one in which there are multiple decisions in sequence. They model the expert as a risk-aware decision-making agent acting according to a *coherent risk metric* (Artzner et al., 1999), and they consider both the case in which the reward function is known, and they try to learn the risk attitude (coherent risk metric) of the expert, and the case in which the reward is unknown, and they aim to estimate both the risk attitude and the reward function. Nevertheless, the authors analyze a model of environment, called *prepare-react model*, rather different from an MDP, since, simply put, it can be seen as an MDP in which the stochasticity is shared by all the state-action pairs at each stage $h \in [\![H]\!]$. Moreover, they consider Markovian policies. Singh et al. (2018) generalizes the work of Majumdar et al. (2017). Specifically, the biggest improvement is to consider nested optimization stages. However, the model of the environment is still rather different from an MDP. We mention also the work of Chen et al. (2019) who extend Majumdar et al. (2017) by devising an active learning framework to improve the efficiency of their learning algorithms.

Another important work is that of Ratliff & Mazumdar (2020), who study the risk-sensitive IRL problem in MDPs, by proposing a parametric model of behavior for the expert based on prospect theory (Kahneman & Tversky, 1979), and they devise a gradient-based IRL algorithm that minimizes a loss function defined on the observed behavior. This work differs from ours in that it assumes that the expert plays actions based on a softmax distribution, i.e., using a Markovian policy.

Cheng et al. (2023) proposes a model of behavior in MDPs using the conditional value-at-risk (Rockafellar & Uryasev, 2000) instead of a utility function (von Neumann & Morgenstern, 1947). Moreover, differently from ours, their model does not contemplate non-Markovian policies. Similarly to us, they analyze the partial identifiability of the parameters representing the risk attitude from demonstrations in a single environment, and propose a strategy for designing the environments in which collecting additional demonstrations in order to reduce the partial identifiability.

We shall mention also the recent pre-print of Cao et al. (2024) that proposes a novel stochastic control framework in continuous time that includes two utility functions and a generic discounting scheme under a time-varying rate. Assuming to know both the utilities and the discounting scheme, the authors show that, through state augmentation, the control problem is well-posed. In addition, the authors provide sufficient conditions for the identification of both the utilities and the discounting scheme given demonstrations of behavior. We note that there are crucial differences between this work and ours. First, the author model the expert as solving an optimization problem in which the utility function is applied to the per *stage* reward, instead we apply the utility to the entire return (see Eq. (1)). Next, they model the expert using Markovian policies.

**Learning utilities from demonstrations.**  Chajewska et al. (2001) considers an approach similar to IRL Ng & Russell (2000). Their goal is not to perform *active* preference elicitation, but, similarly to us, to use demonstrations to infer preferences. Specifically, they aim to learn utilities in sequential decision-making problems from demonstrations. However, they model the problems through decision trees, which are different from MDPs, and this represents the main difference between their work and ours. Indeed, decision trees are simpler since there is no notion of reward function at intermediate states. In this manner, they are able to devise (backward induction) algorithms to learn utilities in decision trees through linear constraints similar to those devised by Ng & Russell (2000) in IRL. It is interesting to notice that they adopt a Bayesian approach to extract a single utility from the feasible set constructed, and not an heuristic like that of Ng & Russell (2000). They assume a prior $p(u)$ over the true utility function $u$, and approximate the posterior w.r.t. the feasible set of utilities $\mathcal{U}$ using Markov Chain Monte Carlo (MCMC).

Shukla et al. (2017) faces the problem of learning human utilities from (video) demonstrations, with the aim of generating meaningful tasks based on the learned utilities. However, differently from us, they consider the stochastic context-free And-Or graph (STC-AOG) framework (Xiong et al., 2016), instead of MDPs.

Lei (2020) considers the problem of learning utilities from demonstrations similarly to Chajewska et al. (2001), but with the difference of considering *influence diagrams* instead of decision trees. Since any influence diagram can be expanded

into a decision tree, authors adopt a strategy similar to Chajewska et al. (2001).

**Others.** Shah et al. (2019) aims to learn the behavioral model of the expert from demonstrations. However, they do not consider a specific model like us (i.e., Eq. (1)), but use a differentiable planner (neural network) to learn the planner. In principle, they can fit any behavioral model (also risk-sentitive models) given the huge expressive power of neural networks. However, their approach requires a lot of demonstrations, even across multiple MDPs. Moreover, this approach does not permit to learn a utility function as a simple, interpretable, and transferrable representation of the risk attitude of the expert.

## B. Additional Notation

In this appendix, we introduce additional notation that will be used in other appendices.

**Miscellaneous.** For any probability distribution $\nu \in \Delta^{\mathbb{R}}$, we denote its cumulative density function by $F_{\nu}$. Let $\nu \in \Delta^{\mathbb{R}}$ be a probability distribution on $\mathbb{R}$; then, for any $y \in [0, 1]$, we define the *generalized inverse* $F_{\nu}^{-1}(y)$ as:

$$F_{\nu}^{-1}(y) := \inf_{x \in \mathbb{R}} \{F_{\nu}(x) \geqslant y\}.$$

We define the *1-Wasserstein distance* $w_1 : \Delta^{\mathbb{R}} \times \Delta^{\mathbb{R}} \to [0, \infty]$ between two probability distributions $\nu, \mu$ as:

$$w_1(\nu, \mu) := \int_0^1 |F_{\nu}^{-1}(y) - F_{\mu}^{-1}(y)| dy. \tag{6}$$

In addition, we define the *Cramér distance* $\ell_2 : \Delta^{\mathbb{R}} \times \Delta^{\mathbb{R}} \to [0, \infty]$ between two probability distributions $\nu, \mu$ as:

$$\ell_2(\nu, \mu) := \left( \int_{\mathbb{R}} (F_{\nu}(y) - F_{\mu}(y))^2 dy \right)^{1/2}. \tag{7}$$

We will use notation:

$$\mathbb{V}_{X \sim Q}[X] := \mathbb{E}_{X \sim Q}[(X - \mathbb{E}_{X \sim Q}[X])^2],$$

to denote the variance of a random variable $X \sim Q$ distributed as $Q$. Given two random variables $X \sim Q_1, Y \sim Q_2$, we denote their covariance as:

$$\text{Cov}_{X \sim Q_1, Y \sim Q_2}[X, Y] := \mathbb{E}_{X \sim Q_1, Y \sim Q_2}[(X - \mathbb{E}_{X \sim Q_1}[X])(Y - \mathbb{E}_{Y \sim Q_2}[Y])].$$

We define the *categorical projection operator* $\text{Proj}_{\mathcal{C}}$ (mentioned in Section 5), that projects onto set $\mathcal{Y} = \{y_1, y_2, \dots, y_d\}$ (the items of $\mathcal{Y}$ are ordered: $y_1 \leqslant y_2 \leqslant \dots \leqslant y_d$, with $y_1 = 0, y_2 = \epsilon_0, y_3 = 2\epsilon_0, \dots, y_d = \lfloor H/\epsilon_0 \rfloor \epsilon_0$), based on Rowland et al. (2018). For single Dirac measures on an arbitrary $y \in \mathbb{R}$, we write:

$$\text{Proj}_{\mathcal{C}}(\delta_y) := \begin{cases} \delta_{y_1} & \text{if } y \leqslant y_1 \\ \frac{y_{i+1} - y}{y_{i+1} - y_i} \delta_{y_i} + \frac{y - y_i}{y_{i+1} - y_i} \delta_{y_{i+1}} & \text{if } y_i < y \leqslant y_{i+1} \\ \delta_{y_d} & \text{if } y > y_d \end{cases}, \tag{8}$$

and we extend it affinely to finite mixtures of $M$ Dirac distributions, so that:

$$\text{Proj}_{\mathcal{C}} \left( \sum_{j \in \llbracket M \rrbracket} q_j \delta_{z_j} \right) = \sum_{j \in \llbracket M \rrbracket} q_j \text{Proj}_{\mathcal{C}}(\delta_{z_j}), \tag{9}$$

for some set of real values $\{z_j\}_{j \in \llbracket M \rrbracket}$ and weights $\{q_j\}_{j \in \llbracket M \rrbracket}$.

**Value functions.** Given an MDP $\mathcal{M} = (\mathcal{S}, \mathcal{A}, H, s_0, p, r)$ and a policy $\pi$, we define the $V$- and $Q$-functions of policy $\pi$ in MDP $\mathcal{M}$ at every $(s, a, h) \in \mathcal{S} \times \mathcal{A} \times \llbracket H \rrbracket$ respectively as $V_h^{\pi}(s; p, r) := \mathbb{E}_{p, r, \pi}[\sum_{t=h}^{H} r_t(s_t, a_t) | s_h = s]$ and $Q_h^{\pi}(s, a; p, r) := \mathbb{E}_{p, r, \pi}[\sum_{t=h}^{H} r_t(s_t, a_t) | s_h = s, a_h = a]$. We define the optimal $V$- and $Q$-functions as $V_h^*(s; p, r) := \sup_{\pi} V_h^{\pi}(s; p, r)$ and $Q_h^*(s, a; p, r) := \sup_{\pi} Q_h^{\pi}(s, a; p, r)$.

For MDPs with an enlarged state space, e.g., $(\{\mathcal{S} \times \mathcal{Y}_h\}_h, \mathcal{A}, H, (s_0, 0), \mathfrak{p}, \mathfrak{r})$, and a policy $\psi = \{\psi_h\}_h$, for all $h \in [\![H]\!]$ and $(s, y, a) \in \mathcal{S} \times \mathcal{Y}_h \times \mathcal{A}$ we denote the $V$- and $Q$-functions respectively as $V_h^\psi(s, y; \mathfrak{p}, \mathfrak{r}) := \mathbb{E}_{\mathfrak{p},\mathfrak{r},\psi}[\sum_{t=h}^H \mathfrak{r}_t(s_t, y_t, a_t)|s_h = s, y_h = y]$ and $Q_h^\psi(s, y, a; \mathfrak{p}, \mathfrak{r}) := \mathbb{E}_{\mathfrak{p},\mathfrak{r},\psi}[\sum_{t=h}^H \mathfrak{r}_t(s_t, y_t, a_t)|s_h = s, y_h = y, a_h = a]$. We denote the optimal $V$- and $Q$-functions as $V_h^*(s, y; \mathfrak{p}, \mathfrak{r}) := \sup_\psi V_h^\psi(s, y; \mathfrak{p}, \mathfrak{r})$ and $Q_h^*(s, y, a; \mathfrak{p}, \mathfrak{r}) := \sup_\psi Q_h^\psi(s, y, a; \mathfrak{p}, \mathfrak{r})$.

Observe that the notation just introduced will be extended in a straightforward manner to MDPs (MDPs with enlarged state space) that have an estimated transition model $\widehat{p}$ $(\widehat{\mathfrak{p}})$, and/or a discretized reward function $\overline{r}$ $(\overline{\mathfrak{r}})$.

# C. Additional Results and Proofs for Section 3

In Appendix C.1, we explain why including the past rewards into the state is not satisfactory, in Appendix C.2 we provide an observation on Eq. (1), while in Appendix C.3 we provide the missing proofs for Section 3.

## C.1. Drawbacks of Re-modelling the MDP

Re-modelling the MDP including the sum of the past rewards into the state would make the demonstrated policy Markovian, and so, in principle, it would allow to apply the existing IRL models meaningfully. However, since in IRL the reward function is unknown, to adopt this trick one should include into the state representation the entire sequence of past state-action pairs, causing the size of the new state space to explode, and also causing the reward function to become non-Markovian w.r.t. the original state space. If instead the reward function was known, and one just wanted to apply one of the risk-sensitive IRL models of behavior presented in Section 7 to learn the (parameters of the) risk attitude, then re-modelling the MDP would still cause the size of the new state space to explode in tabular MDPs, since, in general, there is a number of cumulative reward values that is exponential in the horizon. Moreover, it is not clear why the considered model of behavior, that was designed for the original state space (not including the past rewards), should be realistic in the new state space.

## C.2. An Observation on the Model

If we restrict the optimization problem in Eq. (1) to Markovian policies, we note that non-stationarity (i.e., the dependence of the policy on the stage $h$) and stochasticity (i.e., if the policy prescribes a lottery over actions instead of a single action) can improve the performance w.r.t. Markovian stationary deterministic policies even in stationary environments. Intuitively, the reason is that they permit to consider larger ranges of return distributions w.r.t. Markovian stationary deterministic policies.

**Proposition C.1.** *There exists a RS-MDP with stationary transition model and reward in which the best* Markovian *policy is non-stationary, and the best* stationary Markovian *policy is stochastic.*

## C.3. Proofs for Section 3

**Proposition 3.1.** *There exists a RS-MDP in which the difference between the optimal performance and the performance of the best Markovian policy is* 0.5.

*Proof.* For reasons that will be clear later, let us define symbol $x \approx 2.6$ as the solution of $x - \frac{x^2}{3.99} - 0.1 = 1$.

Consider the RS-MDP $\mathcal{M}_U = (\mathcal{S}, \mathcal{A}, H, s_0, p, r, U)$ in Fig. 4, where $\mathcal{S} = \{s_{\text{init}}, s_1, s_2, s_3, s_4, s_5, s_6\}$, $\mathcal{A} = \{a_1, a_2\}$, $H = 4$, $s_0 = s_{\text{init}}$, transition model $p$ such that:

$$p_1(s_1|s_{\text{init}}, a) = p_1(s_2|s_{\text{init}}, a) = 1/2 \quad \forall a \in \mathcal{A},$$
$$p_2(s_3|s_1, a) = p_2(s_3|s_2, a) = 1 \quad \forall a \in \mathcal{A},$$
$$p_3(s_4|s_3, a_1) = x/3.99, p_3(s_5|s_3, a_1) = 1 - x/3.99, p_3(s_6|s_3, a_2) = 1,$$

reward function $r$ defined as:

$$r_1(s_{\text{init}}, a) = 0 \quad \forall a \in \mathcal{A},$$
$$r_2(s_1, a) = 1 \quad \forall a \in \mathcal{A},$$
$$r_2(s_2, a) = 0 \quad \forall a \in \mathcal{A},$$

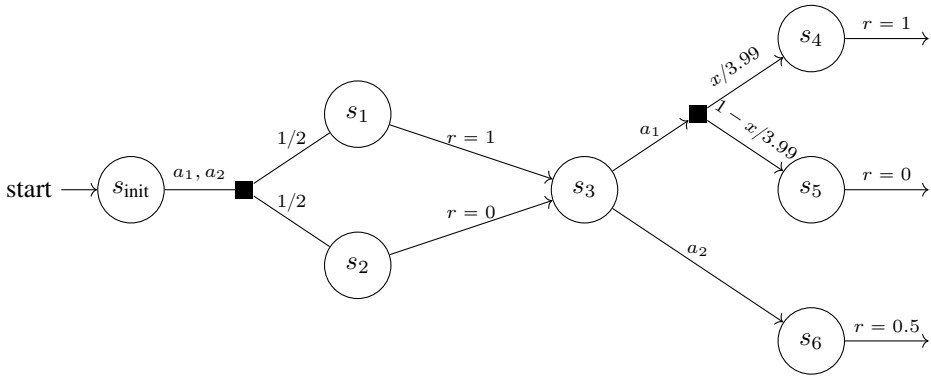

*Figure 4.* MDP for the proof of Proposition 3.1.

$$r_3(s_3, a) = 0 \quad \forall a \in \mathcal{A},$$
$$r_4(s_4, a) = 1 \quad \forall a \in \mathcal{A},$$
$$r_4(s_5, a) = 0 \quad \forall a \in \mathcal{A},$$
$$r_4(s_6, a) = 0.5 \quad \forall a \in \mathcal{A},$$

and utility function $U \in \mathfrak{U}$ that satisfies:

$$U(y) = \begin{cases} x - 0.1 & \text{if } y = 0.5 \\ x & \text{if } y = 1 \\ x + 0.1 & \text{if } y = 1.5 \\ 3.99 & \text{if } y = 2 \end{cases}.$$

Note that this entails that:

$$\frac{x}{3.99}U(2) + U(1) = U(0.5) + U(1.5). \tag{10}$$

Note also that the support of the return function of this (RS-)MDP is $\mathcal{G}^{p,r} = \{0, 0.5, 1, 1.5, 2\}$.

For $\alpha \in [0, 1]$, let $\pi^\alpha$ be the generic Markovian policy that plays action $a_1$ in $s_3$ w.p. $\alpha$ (the actions played in other states are not relevant). Then, its expected utility is:

$$
\begin{aligned}
J^{\pi^\alpha}(U; p, r) &= \frac{1}{2}\Big[\alpha\Big(\frac{x}{3.99}U(2) + (1 - \frac{x}{3.99})U(1)\Big) + (1 - \alpha)U(1.5)\Big] \\
&\quad + \frac{1}{2}\Big[\alpha\Big(\frac{x}{3.99}U(1) + (1 - \frac{x}{3.99})U(0)\Big) + (1 - \alpha)U(0.5)\Big] \\
&\overset{(1)}{=} \frac{1}{2}\Big[\alpha\Big(\frac{x}{3.99}U(2) + U(1)\Big) + (1 - \alpha)(U(1.5) + U(0.5))\Big] \\
&\overset{(2)}{=} \frac{U(1.5) + U(0.5)}{2},
\end{aligned}
$$

where at (1) we have used that $U(0) = 0$, and at (2) we have used Eq. (10).

Thus, all Markovian policies $\pi^\alpha$ have the same performance. Let us consider the non-Markovian policy $\overline{\pi}$ that, in state $s_3$, plays action $a_1$ w.p. 1 if $s_3$ is reached with cumulative reward 1, and it plays action $a_2$ w.p. 1 if $s_3$ is reached with cumulative reward 0. Then, its performance is:

$$J^{\overline{\pi}}(U; p, r) = \frac{1}{2}\Big(\frac{x}{3.99}U(2) + (1 - \frac{x}{3.99})U(1)\Big) + \frac{1}{2}U(0.5).$$

The difference in performance between the optimal performance and that of $\pi^\alpha$ is:

$$J^*(U; p, r) - J^{\pi^\alpha}(U; p, r) \geqslant J^{\overline{\pi}}(U; p, r) - J^{\pi^\alpha}(U; p, r)$$

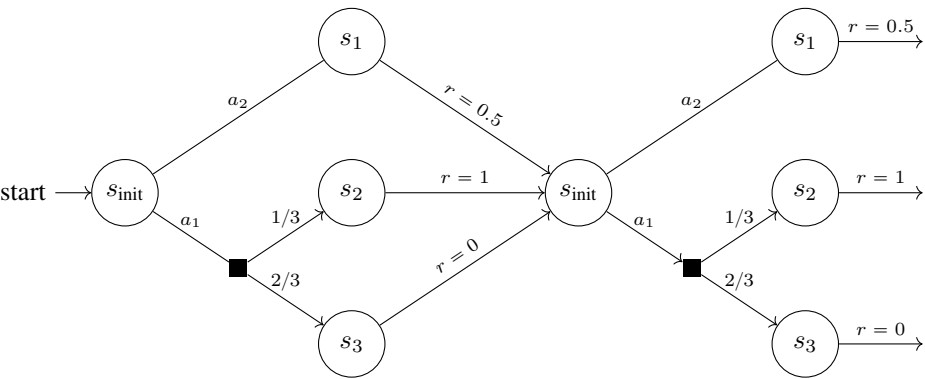

*Figure 5.* MDP for the proof of Proposition C.1.

$$= \frac{1}{2}\Big(\frac{x}{3.99}U(2) + (1 - \frac{x}{3.99})U(1)\Big) + \frac{1}{2}U(0.5) - \frac{U(1.5) + U(0.5)}{2}$$
$$= \frac{1}{2}\Big(\frac{x}{3.99}U(2) + (1 - \frac{x}{3.99})U(1) - U(1.5)\Big)$$
$$\overset{(3)}{=} \frac{1}{2}\Big(x + x - \frac{x^2}{3.99} - x - 0.1\Big)$$
$$= \frac{1}{2}\Big(x - \frac{x^2}{3.99} - 0.1\Big)$$
$$\overset{(4)}{=} 0.5,$$

where at (3) we have replaced the values of utility, and at (4) we have used the definition of $x$.

$\square$

**Proposition 3.2.** *If $p$ is deterministic, then $\arg\max_\pi J^\pi(U^E; p, r^E) = \arg\max_\pi J^\pi(p, r^E)$.*

*Proof.* If $p$ is deterministic, then the optimal policy in Eq. (1) is the policy that deterministically plays the trajectory $\omega$ with largest value of $U^E(g(\omega))$ ($g(\omega)$ denotes the return of $\omega$ under reward $r^E$). Since, by hypothesis, $U^E \in \mathfrak{U}$, then it is strictly increasing, thus such trajectory coincides with the trajectory with largest return. $\square$

**Proposition C.1.** *There exists a RS-MDP with stationary transition model and reward in which the best* Markovian *policy is non-stationary, and the best* stationary Markovian *policy is stochastic.*

*Proof.* Consider the stationary RS-MDP $\mathcal{M}_U = (\mathcal{S}, \mathcal{A}, H, s_0, p, r, U)$ depicted in Fig. 5, where $\mathcal{S} = \{s_{\text{init}}, s_1, s_2, s_3\}$, $\mathcal{A} = \{a_1, a_2\}$, $H = 4$, $s_0 = s_{\text{init}}$, stationary transition model $p$ (we omit subscript because of stationarity) such that:

$$p(s_2|s_{\text{init}}, a_1) = 1 - p(s_3|s_{\text{init}}, a_1) = 1/3,$$
$$p(s_1|s_{\text{init}}, a_2) = 1,$$
$$p(s_{\text{init}}|s, a) = 1 \quad \forall s \in \{s_1, s_2, s_3\}, \forall a \in \mathcal{A},$$

reward function $r$ defined as:

$$r(s_{\text{init}}, a) = 0 \quad \forall a \in \mathcal{A},$$
$$r(s_1, a) = 0.5 \quad \forall a \in \mathcal{A},$$
$$r(s_2, a) = 1 \quad \forall a \in \mathcal{A},$$
$$r(s_3, a) = 0 \quad \forall a \in \mathcal{A},$$

and utility function $U \in \mathfrak{U}$ that satisfies:

$$U(y) = \begin{cases} 0.15 & \text{if } y = 0.5 \\ 0.2 & \text{if } y = 1 \\ 1.8 & \text{if } y = 1.5 \\ 2 & \text{if } y = 2 \end{cases}.$$

Let $\pi^{\alpha,\beta}$ denote the general non-stationary policy that plays action $a_1$ at stage 1 w.p. $\alpha \in [0,1]$, and plays action $a_1$ at stage 2 w.p. $\beta \in [0,1]$. The performance of policy $\pi^{\alpha,\beta}$ can be written as:

$$\begin{aligned} J^{\pi^{\alpha,\beta}}(U;p,r) &= \alpha\left\{\frac{1}{3}\left[\beta\left(\frac{1}{3}U(2) + \frac{2}{3}U(1)\right) + (1-\beta)U(1.5)\right] + \frac{2}{3}\left[\beta\frac{1}{3}U(1) + (1-\beta)U(0.5)\right]\right\} \\ &\quad + (1-\alpha)\left[\beta\left(\frac{1}{3}U(1.5) + \frac{2}{3}U(0.5)\right) + (1-\beta)U(1)\right] \\ &= \alpha\beta\left[\frac{1}{9}U(2) + \frac{13}{9}U(1) - \frac{2}{3}U(1.5) - \frac{4}{3}U(0.5)\right] \\ &\quad + (\alpha+\beta)\left[\frac{1}{3}U(1.5) + \frac{2}{3}U(0.5) - U(1)\right] + U(1) \\ &= \alpha\beta\left[\frac{2}{9} + \frac{13}{45} - \frac{18}{15} - \frac{1}{5}\right] + (\alpha+\beta)\left[\frac{1}{5} + \frac{1}{10} - \frac{1}{5}\right] + \frac{1}{5} \\ &= -\frac{8}{9}\alpha\beta + \frac{1}{10}(\alpha+\beta) + \frac{1}{5}. \end{aligned}$$

To show that the best Markovian policy is non-stationary in this example, we show that the performance of non-stationary policy $\pi^{0,1}$ is better than the performance of all possible Markovian policies. The performance of $\pi^{0,1}$ is:

$$J^{\pi^{0,1}}(U;p,r) = \frac{1}{10} + \frac{1}{5} = 0.3.$$

Instead, the generic stationary policy is $\pi^{\alpha,\alpha}$, and has performance:

$$J^{\pi^{\alpha,\alpha}}(U;p,r) = -\frac{8}{9}\alpha^2 + \frac{1}{5}\alpha + \frac{1}{5}.$$

The value of $\alpha \in [0,1]$ that maximizes this objective is:

$$\frac{d}{d\alpha}J^{\pi^{\alpha,\alpha}}(U;p,r) = -\frac{16}{9}\alpha + \frac{1}{5} = 0 \iff \alpha = \frac{9}{80},$$

from which we get:

$$J^{\pi^{9/80,9/80}}(U;p,r) = \frac{169}{800} \leqslant 0.22,$$

which is smaller than $0.3 = J^{\pi^{0,1}}(U;p,r)$. This concludes the proof of the first part of the proposition.

For the second part, simply observe that, in the problem instance considered, we just obtained that the best Markovian stationary policy plays action $a_1$ w.p. $9/80$, i.e., it is stochastic. $\qquad\square$

# D. Additional Results and Proofs for Section 4

In this appendix, we provide a more explicit formulation for the feasible utility set (Appendix D.1), we present a property of the distance $d^{\text{all}}$ (Appendix D.2), and then we provide the proofs of all the results presented in Section 4 (Appendix D.3).

## D.1. A more Explicit Formulation for the Feasible Set

For any policy $\pi$, we denote by $\mathcal{S}^{p,r,\pi}$ the set of all $(s,h,y)$ state-stage-cumulative reward triples which are covered with non-zero probability by policy $\pi$ in the considered (RS-)MDP.

Thanks to this definition, we can rewrite the feasible set as follows:

**Proposition D.1.** *Let $\mathcal{M} = (\mathcal{S}, \mathcal{A}, H, s_0, p, r)$ be an MDP, and let $\pi^E$ be the expert policy. Then, the feasible utility set $\mathcal{U}_{p,r,\pi^E}$ contains all and only the utility functions that make the actions played by the expert policy optimal at all the $(s, h, y) \in \mathcal{S}^{p,r,\pi^E}$. Formally:*

$$\mathcal{U}_{p,r,\pi^E} = \left\{ U \in \mathfrak{U} \,\middle|\, \forall (s, h, y) \in \mathcal{S}^{p,r,\pi^E}, \forall a \in \mathcal{A} : \right.$$
$$\left. Q_h^*(s, y, \pi_h^E(s, y); p, r) \geqslant Q^*(s, y, a; p, r), \right.$$

*where we used the notation introduced in Appendix B.*

*Proof Sketch.* Based on Theorem 3.1 of Bäuerle & Rieder (2014) (or Theorem 1 of Wu & Xu (2023)), we have that a utility $U \in \mathfrak{U}$ belongs to the feasible set if it makes the expert policy optimal even in the enlarged state space MDP (note that it is possible to define a policy $\psi$ for the enlarged MDP because we are considering policies $\pi$ whose non-Markovianity lies only in the cumulative reward up to now). Therefore, the result follows thanks to a proof analogous to that of Lemma E.1 in Lazzati et al. (2024b), since we are simply considering a common MDP with two variables per state. $\square$

### D.2. A Property of $d^{\text{all}}$

We note that closeness under the max norm (restricted to a certain domain) implies closeness under $d^{\text{all}}$:

**Proposition D.2.** *Consider an arbitrary MDP with transition model $p$ and reward function $r$. Then, for any pair of utilities $U_1, U_2 \in \mathfrak{U}$, it holds that $d_{p,r}^{all}(U_1, U_2) \leqslant \max_{G \in \mathcal{G}^{p,r}} |U_1(G) - U_2(G)|$.*

*Proof.* For the sake of simplicity, we denote the infinity norm and the 1-norm w.r.t. set $\mathcal{G}^{p,r}$ as: $\|f\|_\infty := \max_{G \in \mathcal{G}^{p,r}} |f(G)|$ and $\|f\|_1 := \sum_{G \in \mathcal{G}^{p,r}} |f(G)|$. In addition, we overload notation and use symbols $U_1, U_2$ to denote the vectors in $[0, H]^{|\mathcal{G}^{p,r}|}$ containing, respectively, the values assigned by utility functions $U_1, U_2$ to points in set $\mathcal{G}^{p,r}$. Then, we can write:

$$d_{p,r}^{\text{all}}(U_1, U_2) := \sup_{\pi \in \Pi} |J^\pi(U_1; p, r) - J^\pi(U_2; p, r)|$$
$$= \sup_{\pi \in \Pi} |\mathbb{E}_{G \sim \eta^{p,r,\pi}}[U_1(G)] - \mathbb{E}_{G \sim \eta^{p,r,\pi}}[U_2(G)]|$$
$$= \sup_{\pi \in \Pi} |\mathbb{E}_{G \sim \eta^{p,r,\pi}}[U_1(G) - U_2(G)]|$$
$$\overset{(1)}{\leqslant} \sup_{\eta \in \Delta^{\mathcal{G}^{p,r}}} |\mathbb{E}_{G \sim \eta}[U_1(G) - U_2(G)]|$$
$$\overset{(2)}{\leqslant} \sup_{\eta \in \Delta^{\mathcal{G}^{p,r}}} \mathbb{E}_{G \sim \eta}|U_1(G) - U_2(G)|$$
$$\overset{(3)}{=} \|U_1 - U_2\|_\infty,$$

where at (1) we upper bound by considering the set of all possible distributions over set $\mathcal{G}^{p,r}$ instead of just those induced by some policies in the considered MDP, at (2) we apply triangle inequality, and at (3) we have used the fact that $\|\cdot\|_1$ and $\|\cdot\|_\infty$ are dual norms. $\square$

### D.3. Proofs for Section 4

**Example 4.1.** *Consider the MDP $\mathcal{M}$ in Fig. 2 (left), where $H = 2, r_1(s_0, a_1) = 1, r_1(s_0, a_2) = 0.5$. Let the expert's policy $\pi^E$ prescribe $a_1$ in $s_0$. Then, all the utility functions $U \in \mathfrak{U}$ that take on values in the blue region of Fig. 2 (middle) for returns $G = 1, G = 1.5$, make $\pi^E$ optimal in $\mathcal{M}_U$.*

*Proof.* A utility $U \in \mathfrak{U}$ makes $\pi^E$ optimal for $\mathcal{M}_U$ if playing $a_1$ is better than playing $a_2$: $J^{\pi^E}(U; p, r) = 0.1U(2) + 0.5U(1.5) + 0.4U(1) \geqslant 0.8U(1.5) + 0.2U(1)$. Thus, all the utilities $U \in \mathfrak{U}$, that assign to $G = 1, G = 1.5$ any of the values coloured in blue in Fig. 2 (middle), satisfy this condition. $\square$

**Proposition 4.1.** *There exist two MDPs $\mathcal{M} = (\mathcal{S}, \mathcal{A}, H, s_0, p, r)$, $\mathcal{M}' = (\mathcal{S}, \mathcal{A}, H, s_0, p', r)$, with $p \neq p'$, for which there exist a policy $\pi^E$ and a pair of utilities $U_1, U_2 \in \mathcal{U}_{p,r,\pi^E}$ such that $\Pi_{p',r}^*(U_1) \cap \Pi_{p',r}^*(U_2) = \{\}$.*

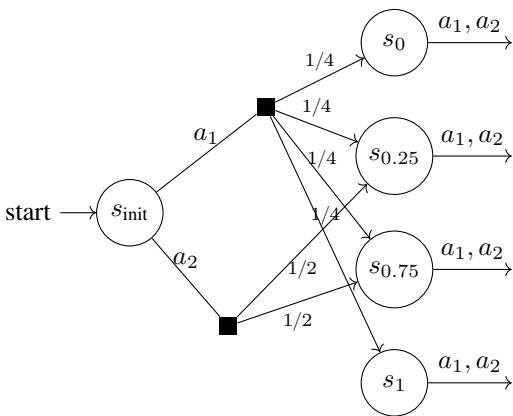

*Figure 6.* MDP for the proof of Proposition 4.1.

*Proof.* We will prove the guarantee stated in the proposition using two different pairs of MDPs: One that that satisfies $\mathcal{G}^{p',r} = \mathcal{G}^{p,r}$, i.e., for which the support of the return function coincides, and the other that does not. Let us begin with the former.

Consider a simple MDP $\mathcal{M} = (\mathcal{S}, \mathcal{A}, H, s_{\text{init}}, p, r)$ with five states $\mathcal{S} = \{s_{\text{init}}, s_0, s_{0.25}, s_{0.75}, s_1\}$, two actions $\mathcal{A} = \{a_1, a_2\}$, horizon $H = 2$, initial state $s_{\text{init}}$, transition model $p$ such that:

$$p_1(s'|s_{\text{init}}, a_1) = \begin{cases} 1/4 & \text{if } s' = s_0 \\ 1/4 & \text{if } s' = s_{0.25} \\ 1/4 & \text{if } s' = s_{0.75} \\ 1/4 & \text{if } s' = s_1 \end{cases},$$

$$p_1(s'|s_{\text{init}}, a_2) = \begin{cases} 1/2 & \text{if } s' = s_{0.25} \\ 1/2 & \text{if } s' = s_{0.75} \end{cases},$$

and reward function $r$ that assigns $r_1(s_{\text{init}}, a_1) = r_1(s_{\text{init}}, a_2) = 0$, and:

$$r_2(s, a) = \begin{cases} 0 & \text{if } s = s_0 \wedge (a = a_1 \vee a = a_2) \\ 0.25 & \text{if } s = s_{0.25} \wedge (a = a_1 \vee a = a_2) \\ 0.75 & \text{if } s = s_{0.75} \wedge (a = a_1 \vee a = a_2) \\ 1 & \text{if } s = s_1 \wedge (a = a_1 \vee a = a_2) \end{cases}.$$

Note that the support of the return function is $\mathcal{G}^{p,r} = \{0, 0.25, 0.75, 1\}$. We are given an expert's policy $\pi^E$ that prescribes action $a_1$ at stage 1 in state $s_{\text{init}}$, and arbitrary actions in other states (the specific action is not relevant). The MDP $\mathcal{M}$ is represented in Figure 6.

Now, we show that utilities $U_1, U_2 \in \mathfrak{U}$, defined in points of the support $\mathcal{G}^{p,r}$ as (and connected in arbitrary continuous strictly-increasing manner between these points):

$$U_1(G) = \begin{cases} 0 & \text{if } G = 0 \\ 0.01 & \text{if } G = 0.25 \\ 0.02 & \text{if } G = 0.75 \\ 1.99 & \text{if } G = 1 \end{cases}, \qquad U_2(G) = \begin{cases} 0 & \text{if } G = 0 \\ 0.01 & \text{if } G = 0.25 \\ 0.99 & \text{if } G = 0.75 \\ 1.99 & \text{if } G = 1 \end{cases},$$

belong to the feasible set $\mathcal{U}_{p,r,\pi^E}$, and, when transferred to the new MDP $\mathcal{M}' = (\mathcal{S}, \mathcal{A}, H, s_{\text{init}}, p', r)$, with transition model $p' \neq p$ defined as:

$$p'_1(\cdot|s_{\text{init}}, a_1) = p_1(\cdot|s_{\text{init}}, a_1),$$

$$p'_1(s'|s_{\text{init}}, a_2) = \begin{cases} 0.7 & \text{if } s' = s_0 \\ 0.3 & \text{if } s' = s_1 \end{cases},$$

impose different optimal policies, i.e., utility $U_2$ keeps making action $a_1$ optimal from state $s_{\text{init}}$ even in $\mathcal{M}'$, while $U_1$ makes action $a_2$ optimal. This proves the thesis of the proposition.

Let us begin by showing that $U_1, U_2 \in \mathcal{U}_{p,r,\pi^E}$ belong to the feasible set of $\mathcal{M}$ with policy $\pi^E$. Let $\overline{\pi}$ be the policy that plays action $a_2$ in state $s_{\text{init}}$. Then, the distribution of returns induced by policies $\pi^E$ and $\overline{\pi}$ are (we represent values only at points in $\mathcal{G}^{p,r} = \{0, 0.25, 0.75, 1\}$):

$$\eta^{p,r,\pi^E} = [1/4, 1/4, 1/4, 1/4]^\intercal$$
$$\eta^{p,r,\overline{\pi}} = [0, 1/2, 1/2, 0]^\intercal.$$

Thus, policy $\pi^E$ is optimal under some utility $U$ if and only if the values assigned by $U$ to points in $\mathcal{G}^{p,r} = \{0, 0.25, 0.75, 1\}$ (denoted, respectively, by $U^1, U^2, U^3, U^4$) satisfy:

$$U^\intercal(\eta^{p,r,\pi^E} - \eta^{p,r,\overline{\pi}}) = [1/4, -1/4, -1/4, 1/4]U = U^1 - U^2 - U^3 + U^4 \geqslant 0,$$

where we have overloaded the notation and denoted with $U := [U^1, U^2, U^3, U^4]^\intercal$ both the utility and the vector of values assigned to points in $\mathcal{G}^{p,r}$. By imposing normalization constraints ($U(0) = 0, U(2) = 2$), we get $U^1 = 0$, and by imposing also the monotonicity constraints, we get that utility $U$ is in the feasible set $\mathcal{U}_{p,r,\pi^E}$ if and only if:

$$\begin{cases} U^4 \geqslant U^2 + U^3 \\ 0 < U^2 < U^3 < U^4 < 2 \end{cases}.$$

Clearly, both utilities $U_1, U_2$ satisfy these constraints, thus they belong to the feasible set $\mathcal{U}_{p,r,\pi^E}$. Now, concerning problem $\mathcal{M}'$, the performances of $\pi^E, \overline{\pi}$ w.r.t. utilities $U_1, U_2$ are:

$$J^{\pi^E}(U_1; p', r) = \frac{1}{4}U_1(0) + \frac{1}{4}U_1(0.25) + \frac{1}{4}U_1(0.75) + \frac{1}{4}U_1(1) = 2.02/4 = 0.505,$$
$$J^{\overline{\pi}}(U_1; p', r) = 0.7U_1(0) + 0.3U_1(1) = 0.3 \times 1.99 = 0.597,$$
$$J^{\pi^E}(U_2; p', r) = \frac{1}{4}U_1(0) + \frac{1}{4}U_1(0.25) + \frac{1}{4}U_1(0.75) + \frac{1}{4}U_1(1) = 2.99/4 = 0.7475,$$
$$J^{\overline{\pi}}(U_2; p', r) = 0.7U_1(0) + 0.3U_1(1) = 0.3 \times 1.99 = 0.597.$$

Clearly, $J^{\pi^E}(U_1; p', r) < J^{\overline{\pi}}(U_1; p', r)$, but $J^{\pi^E}(U_2; p', r) > J^{\overline{\pi}}(U_2; p', r)$, thus we conclude that the set of policies induced by utilities $U_1, U_2$ in $\mathcal{M}'$ do not intersect, since they start from $s_{\text{init}}$ with different actions $\Pi^*_{p',r}(U_1) \cap \Pi^*_{p',r}(U_2) = \{\}$. This concludes the proof with an example that satisfies $\mathcal{G}^{p',r} = \mathcal{G}^{p,r}$.

If we want an example that does *not* satisfy $\mathcal{G}^{p',r} = \mathcal{G}^{p,r}$, then we can consider exactly the same example with $\mathcal{M}$ and $\mathcal{M}'$, but using $r_1(s_{\text{init}}, a_2) = 0.001$. In this manner, we see that $\mathcal{G}^{p,r} = \{0, 0.25, 0.251, 0.75, 0.751, 1\}$, and $\mathcal{G}^{p',r} = \{0, 0.001, 0.25, 0.75, 1, 1.001\}$, which are different. By choosing $U'_1, U'_2$ as:

$$U'_1(G) = \begin{cases} 0 & \text{if } G = 0 \\ 0.001 & \text{if } G = 0.001 \\ 0.01 & \text{if } G = 0.25 \\ 0.011 & \text{if } G = 0.251 \\ 0.02 & \text{if } G = 0.75 \\ 0.021 & \text{if } G = 0.751 \\ 1.99 & \text{if } G = 1 \\ 1.991 & \text{if } G = 1.001 \end{cases}, \qquad U'_2(G) = \begin{cases} 0 & \text{if } G = 0 \\ 0.001 & \text{if } G = 0.001 \\ 0.01 & \text{if } G = 0.25 \\ 0.011 & \text{if } G = 0.251 \\ 0.99 & \text{if } G = 0.75 \\ 0.991 & \text{if } G = 0.751 \\ 1.99 & \text{if } G = 1 \\ 1.991 & \text{if } G = 1.001 \end{cases},$$

it can be shown that $U'_1, U'_2$ belong to the (new) feasible set of $\mathcal{M}$, and that induce different policies in $\mathcal{M}'$. This concludes the proof. $\qquad \square$

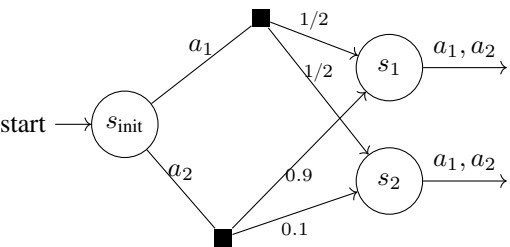

*Figure 7.* MDP for the proof of Proposition 4.2.

**Proposition 4.2.** *There exist two MDPs $\mathcal{M} = (\mathcal{S}, \mathcal{A}, H, s_0, p, r)$, $\mathcal{M}' = (\mathcal{S}, \mathcal{A}, H, s_0, p, r')$, with $r \neq r'$, for which there exist a policy $\pi^E$ and a pair of utilities $U_1, U_2 \in \mathcal{U}_{p,r,\pi^E}$ such that $\Pi^*_{p,r'}(U_1) \cap \Pi^*_{p,r'}(U_2) = \{\}$.*

*Proof.* Similarly to the proof of Proposition 4.1, we provide two examples, one with $\mathcal{G}^{p,r'} = \mathcal{G}^{p,r}$, and the other with $\mathcal{G}^{p,r'} \neq \mathcal{G}^{p,r}$. Let us begin with the former.

Consider a simple MDP $\mathcal{M} = (\mathcal{S}, \mathcal{A}, H, s_{\text{init}}, p, r)$ with three states $\mathcal{S} = \{s_{\text{init}}, s_1, s_2\}$, two actions $\mathcal{A} = \{a_1, a_2\}$, horizon $H = 2$, initial state $s_{\text{init}}$, transition model $p$ such that:

$$p_1(s'|s_{\text{init}}, a_1) = \begin{cases} 1/2 & \text{if } s' = s_1 \\ 1/2 & \text{if } s' = s_2 \end{cases},$$

$$p_1(s'|s_{\text{init}}, a_2) = \begin{cases} 0.9 & \text{if } s' = s_1 \\ 0.1 & \text{if } s' = s_2 \end{cases},$$

and reward function $r$ that assigns $r_1(s_{\text{init}}, a_1) = 0$, $r_1(s_{\text{init}}, a_2) = 0.5$, and:

$$r_2(s, a) = \begin{cases} 0 & \text{if } s = s_1 \wedge (a = a_1 \vee a = a_2) \\ 1 & \text{if } s = s_2 \wedge (a = a_1 \vee a = a_2) \end{cases}.$$

Note that the support of the return function is $\mathcal{G}^{p,r} = \{0, 0.5, 1, 1.5\}$. We are given an expert's policy $\pi^E$ that prescribes action $a_1$ at stage 1 in state $s_{\text{init}}$, and arbitrary actions in other states (the specific action is not relevant). The MDP $\mathcal{M}$ is represented in Figure 7.

Now, we show that the utilities $U_1, U_2 \in \mathfrak{U}$, defined in points of the support $\mathcal{G}^{p,r}$ as (and connected in arbitrary continuous strictly-increasing manner between these points):

$$U_1(G) = \begin{cases} 0 & \text{if } G = 0 \\ 0.1 & \text{if } G = 0.5 \\ 0.9 & \text{if } G = 1 \\ 1.5 & \text{if } G = 1.5 \end{cases}, \qquad U_2(G) = \begin{cases} 0 & \text{if } G = 0 \\ 0.1 & \text{if } G = 0.5 \\ 0.8 & \text{if } G = 1 \\ 1.5 & \text{if } G = 1.5 \end{cases},$$

belong to the feasible set $\mathcal{U}_{p,r,\pi^E}$, and, when transferred to the new MDP $\mathcal{M}' = (\mathcal{S}, \mathcal{A}, H, s_{\text{init}}, p, r')$, with reward function $r' \neq r$ defined as:

$$r'_1(s_{\text{init}}, a_1) = 0.5, \qquad r_1(s_{\text{init}}, a_2) = 0,$$

$$r'_2(s, a) = \begin{cases} 1 & \text{if } s = s_1 \wedge (a = a_1 \vee a = a_2) \\ 0 & \text{if } s = s_2 \wedge (a = a_1 \vee a = a_2) \end{cases},$$

impose different optimal policies, i.e., utility $U_2$ keeps making action $a_1$ optimal from state $s_{\text{init}}$ even in $\mathcal{M}'$, while $U_1$ makes action $a_2$ optimal. This will demonstrate the thesis of the proposition.

Let us begin by showing that $U_1, U_2 \in \mathcal{U}_{p,r,\pi^E}$ belong to the feasible set of $\mathcal{M}$ with policy $\pi^E$. Let $\bar{\pi}$ be the policy that plays action $a_2$ in state $s_{\text{init}}$. Then, the distribution of returns induced by policies $\pi^E$ and $\bar{\pi}$ are (we represent values only

at points in $\mathcal{G}^{p,r} = \{0, 0.5, 1, 1.5\}$):

$$\eta^{p,r,\pi^E} = [0.5, 0, 0.5, 0]^\intercal$$
$$\eta^{p,r,\overline{\pi}} = [0, 0.9, 0, 0.1]^\intercal.$$

Thus, policy $\pi^E$ is optimal under some utility $U$ if and only if the values assigned by $U$ to points in $\mathcal{G}^{p,r} = \{0, 0.5, 1, 1.5\}$ (denoted, respectively, by $U^1, U^2, U^3, U^4$) satisfy:

$$U^\intercal(\eta^{p,r,\pi^E} - \eta^{p,r,\overline{\pi}}) = [0.5, -0.9, 0.5, -0.1]U = 0.5U^1 - 0.9U^2 + 0.5U^3 - 0.1U^4 \geqslant 0,$$

where we have overloaded the notation and denoted with $U := [U^1, U^2, U^3, U^4]^\intercal$ both the utility and the vector of values assigned to points in $\mathcal{G}^{p,r}$. By imposing normalization constraints ($U(0) = 0, U(2) = 2$), we get $U^1 = 0$, and by imposing also the monotonicity constraints, we get that utility $U$ is in the feasible set $\mathcal{U}_{p,r,\pi^E}$ if and only if:

$$\begin{cases} U^4 \geqslant 5U^3 - 9U^2 \\ 0 < U^2 < U^3 < U^4 < 2 \end{cases}.$$

Clearly, both utilities $U_1, U_2$ satisfy these constraints, thus they belong to the feasible set $\mathcal{U}_{p,r,\pi^E}$. Now, concerning problem $\mathcal{M}'$, the performances of $\pi^E, \overline{\pi}$ w.r.t. utilities $U_1, U_2$ are:

$$J^{\pi^E}(U_1; p, r') = 0U_1(0) + 0.5U_1(0.5) + 0U_1(1) + 0.5U_1(1.5) = 1.6/2 = 0.8,$$
$$J^{\overline{\pi}}(U_1; p, r') = 0.1U_1(0) + 0U_1(0.5) + 0.9U_1(1) + 0U_1(1.5) = 0.9 \times 0.9 = 0.81,$$
$$J^{\pi^E}(U_2; p, r') = 0U_2(0) + 0.5U_2(0.5) + 0U_2(1) + 0.5U_2(1.5) = 1.6/2 = 0.8,$$
$$J^{\overline{\pi}}(U_2; p, r') = 0.1U_2(0) + 0U_2(0.5) + 0.9U_2(1) + 0U_2(1.5) = 0.9 \times 0.8 = 0.72.$$

Clearly, $J^{\pi^E}(U_1; p, r') < J^{\overline{\pi}}(U_1; p, r')$, but $J^{\pi^E}(U_2; p, r') > J^{\overline{\pi}}(U_2; p, r')$, thus we conclude that the set of policies induced by utilities $U_1, U_2$ in $\mathcal{M}'$ do not intersect, since they start from $s_{\text{init}}$ with different actions $\Pi^*_{p,r'}(U_1) \cap \Pi^*_{p,r'}(U_2) = \{\}$. This concludes the proof with an example that satisfies $\mathcal{G}^{p,r'} = \mathcal{G}^{p,r}$.

If we want an example that does *not* satisfy $\mathcal{G}^{p,r'} = \mathcal{G}^{p,r}$, then we can consider exactly the same example with $\mathcal{M}$ and $\mathcal{M}'$, but using $r'_1(s_{\text{init}}, a_2) = 0.001$. In this manner, we see that $\mathcal{G}^{p,r} = \{0, 0.5, 1, 1.5\}$, and $\mathcal{G}^{p',r} = \{0.001, 0.5, 1.001, 1.5\}$, which are different. Nevertheless, by choosing $U'_1, U'_2$ as:

$$U'_1(G) = \begin{cases} 0 & \text{if } G = 0 \\ 0.001 & \text{if } G = 0.001 \\ 0.1 & \text{if } G = 0.5 \\ 0.9 & \text{if } G = 1 \\ 0.901 & \text{if } G = 1.001 \\ 1.5 & \text{if } G = 1.5 \end{cases}, \qquad U'_2(G) = \begin{cases} 0 & \text{if } G = 0 \\ 0.001 & \text{if } G = 0.001 \\ 0.1 & \text{if } G = 0.5 \\ 0.8 & \text{if } G = 1 \\ 0.801 & \text{if } G = 1.001 \\ 1.5 & \text{if } G = 1.5 \end{cases},$$

it can be shown that $U'_1, U'_2$ still belong to the feasible set of $\mathcal{M}$ (the constraints are the same), and that induce different policies in $\mathcal{M}'$. This concludes the proof. $\qquad \square$

**Proposition 4.3.** *There exists an MDP $\mathcal{M} = (\mathcal{S}, \mathcal{A}, H, s_0, p, r)$ and a policy $\pi^E$ for which there are utilities $U_1, U_2 \in \mathcal{U}_{p,r,\pi^E}$ such that, for any $\epsilon \geqslant 0$ smaller than some universal constant, there exists a policy $\pi_\epsilon$ such that $J^*(U_1; p, r) - J^{\pi_\epsilon}(U_1; p, r) = \epsilon$ and $J^*(U_2; p, r) - J^{\pi_\epsilon}(U_2; p, r) \geqslant 1$.*

*Proof.* Consider a simple MDP $\mathcal{M} = (\mathcal{S}, \mathcal{A}, H, s_{\text{init}}, p, r)$ with four states $\mathcal{S} = \{s_{\text{init}}, s_1, s_2, s_3\}$, three actions $\mathcal{A} = \{a_1, a_2, a_3\}$, horizon $H = 2$, initial state $s_{\text{init}}$, transition model $p$ such that:

$$p_1(s_2|s_{\text{init}}, a_1) = 1, \qquad p_1(s_1|s_{\text{init}}, a_3) = 1,$$
$$p_1(s'|s_{\text{init}}, a_2) = \begin{cases} 0.91 & \text{if } s' = s_1 \\ 0.09 & \text{if } s' = s_3 \end{cases},$$

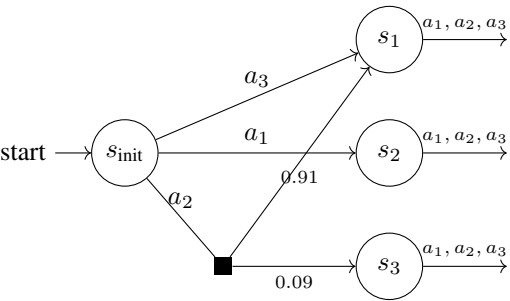

*Figure 8.* MDP for the proof of Proposition 4.3.

and reward function $r$ that assigns $r_1(s_{\text{init}}, a_1) = r_1(s_{\text{init}}, a_2) = r_1(s_{\text{init}}, a_3) = 0$, and:

$$
r_2(s, a) = \begin{cases} 0 & \text{if } s = s_1 \wedge (a = a_1 \vee a = a_2 \vee a = a_3) \\ 0.5 & \text{if } s = s_2 \wedge (a = a_1 \vee a = a_2 \vee a = a_3) \\ 1 & \text{if } s = s_3 \wedge (a = a_1 \vee a = a_2 \vee a = a_3) \end{cases} .
$$

Note that the support of the return function is $\mathcal{G}^{p,r} = \{0, 0.5, 1\}$. We are given an expert's policy $\pi^E$ that prescribes action $a_1$ at stage 1 in state $s_{\text{init}}$, and arbitrary actions in other states (the specific action is not relevant). The MDP $\mathcal{M}$ is represented in Figure 8.

Now, we show that the utilities $U_1, U_2 \in \mathfrak{U}$, defined in points of the support $\mathcal{G}^{p,r}$ as (and connected in arbitrary continuous strictly-increasing manner between these points):

$$
U_1(G) = \begin{cases} 0 & \text{if } G = 0 \\ 0.1 & \text{if } G = 0.5 \\ 0.1/0.09 & \text{if } G = 1 \end{cases}, \qquad U_2(G) = \begin{cases} 0 & \text{if } G = 0 \\ 1.099 & \text{if } G = 0.5 \\ 1.1 & \text{if } G = 1 \end{cases},
$$

belong to the feasible set $\mathcal{U}_{p,r,\pi^E}$, and that, for any $\epsilon \in [0, 0.1]$, there exists a policy $\pi$ for which it holds both that $J^*(U_1; p, r) - J^\pi(U_1; p, r) = \epsilon$ and $J^*(U_2; p, r) - J^\pi(U_2; p, r) \geqslant 1$.

First, let us show that both $U_1, U_2$ belong to the feasible utility set. Let $\pi^1, \pi^2, \pi^3$ be the policies that play, respectively, action $a_1, a_2, a_3$ in state $s_{\text{init}}$ (note that $\pi^1 = \pi^E$). Then, their performances for arbitrary utility $U$ are:

$$
\begin{aligned}
J^{\pi^1}(U; p, r) &= U(0.5), \\
J^{\pi^2}(U; p, r) &= 0.09 U(1) + 0.91 U(0) = 0.09 U(1), \\
J^{\pi^3}(U; p, r) &= U(0) = 0,
\end{aligned}
$$

where we have used the normalization condition. Replacing $U$ with $U_1$, we get $J^*(U_1; p, r) = J^{\pi^1}(U_1; p, r) = 0.1 = J^{\pi^2}(U_1; p, r) = 0.1 > J^{\pi^3}(U_1; p, r) = 0$. Instead, replacing with $U_2$, we get $J^*(U_2; p, r) = J^{\pi^1}(U_2; p, r) = 1.099 > J^{\pi^2}(U_2; p, r) = 0.09 \times 1.1 > J^{\pi^3}(U_2; p, r) = 0$. Therefore, both $U_1, U_2 \in \mathcal{U}_{p,r,\pi^E}$.

Now, for any $\alpha \in [0, 1]$ let us denote by $\pi_\alpha$ the policy that, at state $s_{\text{init}}$, plays action $a_3$ w.p. $\alpha$, and action $a_2$ w.p. $1 - \alpha$. We show that, for any $\epsilon \in [0, 0.1]$, policy $\pi_{\epsilon/0.1}$ is $\epsilon$-optimal for utility $U_1$, and its suboptimality is at least 1 under utility $U_2$. For any $\alpha \in [0, 1]$, the expected utilities of policy $\pi_\alpha$ under $U_1$ and $U_2$ are:

$$
\begin{aligned}
J^{\pi_\alpha}(U_1; p, r) &= (1 - \alpha) \times 0.09 \times U_1(1) = (1 - \alpha) \times 0.1, \\
J^{\pi_\alpha}(U_2; p, r) &= (1 - \alpha) \times 0.09 \times U_2(1) = (1 - \alpha) \times 0.099,
\end{aligned}
$$

from which we derive that the suboptimalities of such policy under $U_1$ and $U_2$ are:

$$
\begin{aligned}
J^*(U_1; p, r) - J^{\pi_\alpha}(U_1; p, r) &= 0.1 - (1 - \alpha) \times 0.1 = 0.1\alpha, \\
J^*(U_2; p, r) - J^{\pi_\alpha}(U_2; p, r) &= 1.099 - (1 - \alpha) \times 0.099 = 1 + 0.099\alpha.
\end{aligned}
$$

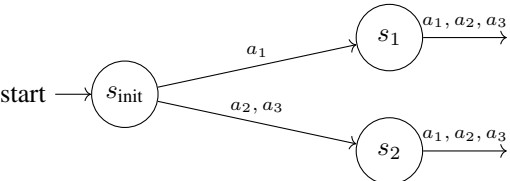

*Figure 9.* MDP for the proof of Proposition 4.4.

Thus, for any $\epsilon \in [0, 0.1]$, policy $\pi_{\epsilon/0.1}$ is $\epsilon$-optimal for utility $U_1$, but it is at least 1-suboptimal for utility $U_2$.

The intuition is that utilities $U_1$ and $U_2$ assess in completely different manners the policies that play action $a_2$, although they both describe policy $\pi^E$ as optimal. This concludes the proof. $\qquad\square$

**Proposition 4.4.** *There exists an MDP $\mathcal{M} = (\mathcal{S}, \mathcal{A}, H, s_0, p, r)$ and a policy $\pi^E$ for which there exists a pair of utilities $U_1, U_2 \in \mathcal{U}_{p, r, \pi^E}$ such that $d_{p, r}^{all}(U_1, U_2) = 1$.*

*Proof.* Consider a simple MDP $\mathcal{M} = (\mathcal{S}, \mathcal{A}, H, s_{\text{init}}, p, r)$ with three states $\mathcal{S} = \{s_{\text{init}}, s_1, s_2\}$, three actions $\mathcal{A} = \{a_1, a_2, a_3\}$, horizon $H = 2$, initial state $s_{\text{init}}$, transition model $p$ such that:

$$p_1(s_1 | s_{\text{init}}, a_1) = 1, \qquad p_1(s_2 | s_{\text{init}}, a_2) = p_1(s_2 | s_{\text{init}}, a_2) = 1,$$

and reward function $r$ that assigns $r_1(s_{\text{init}}, a_1) = r_1(s_{\text{init}}, a_2) = 0$, $r_1(s_{\text{init}}, a_2) = 1$, and:

$$r_2(s, a) = \begin{cases} 0 & \text{if } s = s_1 \wedge (a = a_1 \vee a = a_2 \vee a_3) \\ 1 & \text{if } s = s_2 \wedge (a = a_1 \vee a = a_2 \vee a_3) \end{cases}.$$

Note that the support of the return function is $\mathcal{G}^{p,r} = \{0, 1, 2\}$. We are given an expert's policy $\pi^E$ that prescribes action $a_3$ at stage 1 in state $s_{\text{init}}$, and arbitrary actions in the other states (the specific action is not relevant). The MDP $\mathcal{M}$ is represented in Figure 9.

Consider two utilities $U_1, U_2$, that take on the following values in $\mathcal{G}^{p,r}$:

$$U_1(G) = \begin{cases} 0 & \text{if } G = 0 \\ 0.1 & \text{if } G = 1 \\ 2 & \text{if } G = 2 \end{cases},$$

$$U_2(G) = \begin{cases} 0 & \text{if } G = 0 \\ 1.1 & \text{if } G = 1 \\ 2 & \text{if } G = 2 \end{cases}.$$

It is immediate that both utilities belong to the feasible set $\mathcal{U}_{p, r, \pi^E}$. Nevertheless, if we denote by $\overline{\pi}$ the policy that plays action $a_2$ in state $s_{\text{init}}$, we see that $J^{\overline{\pi}}(U_1; p, r) = 0.1$, while $J^{\overline{\pi}}(U_2; p, r) = 1.1$, so that the difference is 1. $\qquad\square$

**Proposition 4.5.** *Let $\mathcal{S}, \mathcal{A}, H$ be any state space, action space, and horizon, satisfying $S \geqslant 3, A \geqslant 2, H \geqslant 2$, and let $U^E \in \mathfrak{U}$ be any utility. If, for any possible dynamics $s_0, p$ and reward $r$, we are given the set of all the deterministic optimal policies of the corresponding RS-MDP $(\mathcal{S}, \mathcal{A}, H, s_0, p, r, U^E)$, then we can uniquely identify $U^E$.*

*Proof.* We provide a constructive proof that shows which values of $s_0, p, r$ it is sufficient to choose for recovering $U^E$ exactly. The construction is articulated into two parts. First, we aim to recover the value of $U^E(1)$, i.e., for $G = 1$; next, we recover the utility for all other possible values of return. The intuition is that we construct a Standard Gamble (SG) between two policies over the entire horizon (Wakker, 2010).

To infer $U^E(1)$, we use the $s_0, p, r$ values that provide the MDP described in Figure 10.

We consider a single initial state $s_{\text{init}}$. From here, action $a_1$ (and all actions other than $a_1$ and $a_2$) brings deterministically to state $s_1^2$, while action $a_2$ brings to state $s_3^2$ w.p. $q$ (to choose, for some $q \in [0, 1]$), and to state $s_2^2$ w.p. $1 - q$. From

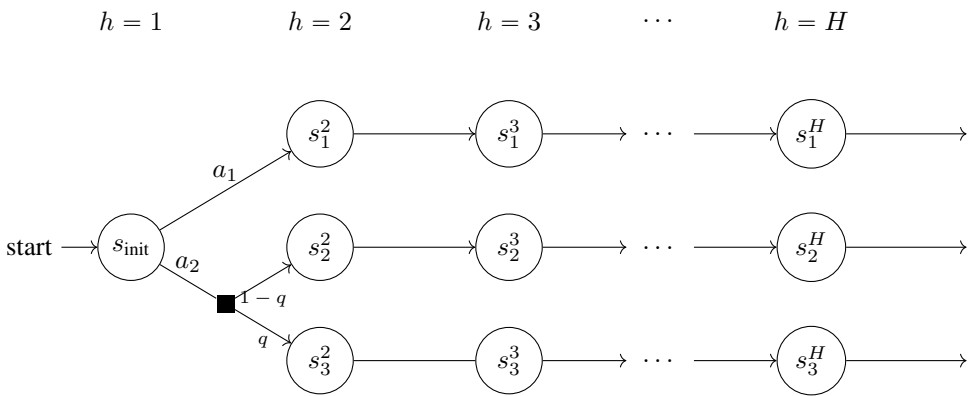

$h = 1$ $\qquad h = 2 \qquad\qquad h = 3 \qquad\qquad \cdots \qquad\qquad h = H$

*Figure 10.* MDP for the proof of Proposition 4.5.

state $s_i^2$, for any $i \in [\![3]\!]$, all actions bring deterministically to state $s_i^3$, and so on, up to state $s_i^H$. We will call the trajectory $\{s_{\text{init}}, s_i^2, s_i^3, \ldots, s_i^H\}$ the $i^{\text{th}}$ trajectory for all $i \in [\![3]\!]$, and we will write $G(i)$ to denote the sum of rewards along such trajectory. To infer the value $U^E(1)$, we select a reward $r' : \mathcal{S} \times \mathcal{A} \times [\![H]\!] \to [0, 1]$ that provides return $G(1) = 1.5$ to the first trajectory, return $G(2) = 1$ to the second trajectory, and return $G(3) = H$ to the third trajectory (this is possible because $H \geqslant 2$). By selecting, successively, all the values of $q \in [0, 1]$, we are asking to the expert to play either action $a_1$ or action $a_2$ from the initial state $s_{\text{init}}$ (we denote policies $\pi^1, \pi^2$, respectively, the policies that play actions $a_1, a_2$ in $s_{\text{init}}$). Since we are assuming that the expert will demonstrate all the possible deterministic optimal policies, there exists a value $q' \in [0, 1]$ for which the expert demonstrates both policies $\pi^1$ and $\pi^2$. Indeed, the expected utilities of policies $\pi^1, \pi^2$ for arbitrary value of $q$ are (we write $p(q)$ as the generic transition model):

$$J^{\pi^1}(U^E; p(q), r') = U^E(1.5),$$
$$J^{\pi^2}(U^E; p(q), r') = qU^E(H) + (1 - q)U^E(1) = qH + (1 - q)U^E(1),$$

and since $U^E$ is strictly-increasing, we have $U^E(1) < U^E(1.5) < U^E(H) = H$, thus there must exist $q'$ that permits to write $U^E(1.5)$ as a convex combination of the other two. This allows us to write:

$$U^E(1.5) = q'H + (1 - q')U^E(1). \tag{11}$$

Next, we select reward $r''$ that provides returns $G(1) = 1, G(2) = 0.5, G(3) = 1.5$. Thus, there must exist a $q'' \in [0, 1]$ for which the expert demonstrates both policies $\pi^1$ and $\pi^2$, allowing us to write:

$$U^E(1) = q''U^E(1.5) + (1 - q'')U^E(0.5). \tag{12}$$

Finally, we can repeat the same step with a third reward $r'''$ that provides returns $G(1) = 0.5, G(2) = 0, G(3) = 1$, and for some $q''' \in [0, 1]$ we obtain:

$$U^E(0.5) = q'''U^E(1). \tag{13}$$

By putting together Eq. (11), Eq. (12), and Eq. (13), we can retrieve $U^E(1)$:

$$\begin{cases} U^E(1.5) = q'H + (1 - q')U^E(1) \\ U^E(1) = q''U^E(1.5) + (1 - q'')U^E(0.5) \\ U^E(0.5) = q'''U^E(1) \end{cases}.$$

Now that we know $U^E(1)$, we can infer the utility for all the returns $\overline{G} \in (1, H)$ by choosing a reward that provides returns $G(1) = \overline{G}, G(2) = 1, G(3) = H$, because for some $\overline{q} \in [0, 1]$ the expert will play both policies $\pi^1$ and $\pi^2$, which allows us to write:

$$U^E(\overline{G}) = \overline{q}H + (1 - \overline{q})U^E(1),$$

and to retrieve $U^E(\overline{G})$.

Similarly, for all $\overline{G} \in (0, 1)$, we select a reward that provides returns $G(1) = \overline{G}, G(2) = 0, G(3) = 1$, and for some $\overline{q} \in [0, 1]$ we can write:

$$U^E(\overline{G}) = \overline{q}U^E(1),$$

and retrieve $U^E(\overline{G})$.

This concludes the proof. As a final remark, we stress that the initial step for inferring $U^E(1)$ cannot be dropped because there is no reward $r : \mathcal{S} \times \mathcal{A} \times \llbracket H \rrbracket \to [0, 1]$ that provides returns $G(2) = 0$ and $G(3) = H$, because both the first and second trajectories pass through action $a_2$ in state $s_{\text{init}}$. $\qquad\square$

# E. Additional Results and Proofs for Section 5

This appendix is divided in 5 parts. First, we show the complexity of implementing operator $\Pi_{\overline{\mathfrak{U}}_L}$ (Appendix E.1). In Appendix E.2, we provide the pseudocode, along with a description, of algorithms EXPLORE, PLANNING, ERD, and ROLLOUT. In Appendix E.3 we analyze the time and space complexities of **CATY-UL** and **TRACTOR-UL**. In Appendix E.4, we provide the proof of Theorem 5.1. In Appendix E.5, we provide the proof of Theorem 5.2.

## E.1. Projecting onto the Set of Discretized Utilities

Let us use the square brackets $[]$ to denote the components of vectors. Then, note that set $\overline{\mathfrak{U}}_L$ can be represented more explicitly as:

$$\overline{\mathfrak{U}}_L = \{\overline{U} \in [0, H]^d \,|\, \overline{U}[1] = 0 \wedge \overline{U}[d] = H \wedge \overline{U}[i] \leqslant \overline{U}[i+1] \; \forall i \in \llbracket d-1 \rrbracket$$
$$\wedge \; \forall i, j \in \llbracket d \rrbracket \text{ s.t. } i < j : |\overline{U}[i] - \overline{U}[j]| \leqslant L(j-i)\epsilon_0\}. \tag{14}$$

Notice that set $\overline{\mathfrak{U}}_L$ is closed and convex, since it is defined by linear constraints only. The amount of constraints scales as $\propto d^2$.

We remark that in Theorem 5.2 we assume availability of an oracle for computing the projection exactly. In practice, we can use any quadratic programming solver to approximate the projection.

## E.2. Missing Algorithms and Sub-routines

**EXPLORE** In Algorithm 3, we report the pseudo-code implementing subroutine EXPLORE. Simply put, we adopt a uniform-sampling strategy, i.e., we collect $n = \lfloor \tau/(SAH) \rfloor$ samples from each $(s, a, h) \in \mathcal{S} \times \mathcal{A} \times \llbracket H \rrbracket$ triple, that we use to compute the empirical estimate of the transition model. We return such estimate.

**PLANNING** The PLANNING sub-routine (Algorithm 4) takes in input a utility $U$, an environment index $i$, and a transition model $p$, that uses to construct the RS-MDP $\mathcal{M}_U := (\mathcal{S}^i, \mathcal{A}^i, H, s_0^i, p, \overline{r}^i, U)$. Notice that $\mathcal{M}_U \neq \mathcal{M}_{U^E}^i$, for 3 aspects. First, it uses the input transition model $p \neq p^i$; next, it consider the discretized reward $\overline{r}^i \neq r^i$; finally, it has input utility $U \neq U^E$.

PLANNING outputs two items. The optimal performance $J^*(U; p, r^i)$ for RS-MDP $\mathcal{M}_U$, and the optimal policy $\psi^* = \{\psi_h^*\}_h$ for the enlarged state space MDP $\mathfrak{E}[\mathcal{M}_U]$. However, it should be remarked that, instead of computing optimal policy $\psi^*$ for $\mathfrak{E}[\mathcal{M}_U]$ only at pairs $(s, y) \in \mathcal{S} \times \mathcal{G}_h^{p, \overline{r}^i}$ for all $h \in \llbracket H \rrbracket$, PLANNING computes the optimal policy $\psi^*$ at all pairs $(s, y) \in \mathcal{S} \times \mathcal{Y}_h$ for all $h \in \llbracket H \rrbracket$ (note that $\mathcal{G}_h^{p, \overline{r}^i} \subseteq \mathcal{Y}_h$).

The algorithm implemented in PLANNING for computing both $J^*(U; p, r^i)$ and $\psi^*$ is value iteration. The difference from common implementations of value iterations lies in the presence of an additional variable in the state. A similar pseudocode is provided in Algorithm 1 of Wu & Xu (2023).

**ERD (Estimate the Return Distribution)** The ERD sub-routine (Algorithm 5) takes in input a dataset $\mathcal{D}^E = \{\omega_j\}_j$ of state-action trajectories $\omega_j \in \Omega$ and a reward function $r$, and it computes an estimate of the return distribution w.r.t. $r$.

---

**Algorithm 3** `EXPLORE`

---

**Input:** samples budget $\tau$

22   $n \leftarrow \lfloor \tau/(SAH) \rfloor$

23   **for** $i \in \{1, 2, \ldots, N\}$ **do**

     // Initialize the transition model estimate:

24     $\widehat{p}_h^i(s'|s,a) = 0$ for all $(s,a,h,s') \in \mathcal{S} \times \mathcal{A} \times [\![H]\!] \times \mathcal{S}$

     // Collect samples:

25     **for** $(s,a,h) \in \mathcal{S} \times \mathcal{A} \times [\![H]\!]$ **do**

26       **for** $\_ \in \{1, 2, \ldots, n\}$ **do**

27         $s' \leftarrow$ sample from $p_h^i(\cdot|s,a)$

28         $\widehat{p}_h^i(s'|s,a) \leftarrow \widehat{p}_h^i(s'|s,a) + 1$

29       **end**

30     **end**

31     $\widehat{p}_h^i(\cdot|s,a) \leftarrow \widehat{p}_h^i(\cdot|s,a)/n$

32   **end**

33   **Return** $\{\widehat{p}^i\}_i$

---

**Algorithm 4** `PLANNING`

---

**Input:** utility $U$, environment index $i$, transition model $p$

   // Initialize the Q and value function at the last stage:

34   **for** $(s,y) \in \mathcal{S}^i \times \mathcal{Y}_H$ **do**

35     **for** $a \in \mathcal{A}^i$ **do**

36       $Q_H(s,y,a) \leftarrow U(y + \overline{r}_H^i(s,a))$

37     **end**

38     $V_H(s,y) \leftarrow \max_{a \in \mathcal{A}^i} Q_H(s,y,a)$

39     $\psi_H(s,y) \leftarrow \arg\max_{a \in \mathcal{A}^i} Q_H(s,y,a)$             `/* Keep just one action */`

40   **end**

   // Backward induction:

41   **for** $h = H-1, \ldots, 2, 1$ **do**

42     **for** $(s,y) \in \mathcal{S}^i \times \mathcal{Y}_h$ **do**

43       **for** $a \in \mathcal{A}^i$ **do**

44         $Q_h(s,y,a) \leftarrow \mathbb{E}_{s' \sim p_h(\cdot|s,a)}\Big[V_{h+1}(s', y + \overline{r}_h^i(s,a))\Big]$

45       **end**

46       $V_h(s,y) \leftarrow \max_{a \in \mathcal{A}^i} Q_h(s,y,a)$

47       $\psi_h(s,y) \leftarrow \arg\max_{a \in \mathcal{A}^i} Q_h(s,y,a)$        `/* Keep just one action */`

48     **end**

49   **end**

   // Return optimal performance and policy:

50   **Return** $V_1(s_0^i, 0), \psi$

---

For every trajectory $\omega_j \in \mathcal{D}^E$, `ERD` computes the return $G_j$ of $\omega_j$ based on the input reward $r$ (Line 55). In the next lines, `ERD` simply computes the categorical projection of the mixture of Dirac deltas:

$$\widehat{\eta} = \text{Proj}_{\mathcal{C}}\Big( \sum_j \frac{1}{|\mathcal{D}^E|} \delta_{G_j} \Big),$$

where the categorical projection operator $\text{Proj}_{\mathcal{C}}$ is defined in Eq. (8).

**ROLLOUT**   `ROLLOUT` (Algorithm 6) takes in input a Markovian policy $\psi$, a transition model $p$, a reward $r$, an environment index $i$, and a number of trajectories $K$, to construct the MDP $\mathcal{M} := (\mathcal{S}^i, \mathcal{A}^i, H, s_0^i, p, r)$ obtained from MDP $\mathcal{M}^i$ by

---

**Algorithm 5** ERD - Estimate the Return Distribution

---

**Input:** dataset $\mathcal{D}^E$, reward $r$
// Initialize $\widehat{\eta}$:
51 **for** $y \in \mathcal{Y}$ **do**
52 $\quad$ $\widehat{\eta}(y) \leftarrow 0$
53 **end**
$\quad$ // Loop over all trajectories in $\mathcal{D}^E$:
54 **for** $\omega \in \mathcal{D}^E$ **do**
$\quad\quad$ // Compute return of $\omega = \{s_1, a_1, \ldots, s_H, a_H, s_{H+1}\}$:
55 $\quad$ $G \leftarrow \sum_{h=1}^{H} r_h(s_h, a_h)$
$\quad\quad$ // Update estimate $\widehat{\eta}$:
56 $\quad$ **if** $G \leqslant 0$ **then**
57 $\quad\quad$ $\widehat{\eta}(0) \leftarrow \widehat{\eta}(0) + 1$
58 $\quad$ **end**
59 $\quad$ **else if** $G > \lfloor \frac{H}{\epsilon_0} \rfloor \epsilon_0$ **then**
60 $\quad\quad$ $\widehat{\eta}(\lfloor \frac{H}{\epsilon_0} \rfloor \epsilon_0) \leftarrow \widehat{\eta}(\lfloor \frac{H}{\epsilon_0} \rfloor \epsilon_0) + 1$
61 $\quad$ **end**
62 $\quad$ **else**
63 $\quad\quad$ $L \leftarrow \max_{y \in \mathcal{Y} \wedge y < G} y$
64 $\quad\quad$ $U \leftarrow \min_{y \in \mathcal{Y} \wedge y \geqslant G} y$
65 $\quad\quad$ $\widehat{\eta}(L) \leftarrow \widehat{\eta}(L) + \frac{U-G}{U-L}$
66 $\quad\quad$ $\widehat{\eta}(U) \leftarrow \widehat{\eta}(U) + \frac{G-L}{U-L}$
67 $\quad$ **end**
68 **end**
$\quad$ // Normalize:
69 $\widehat{\eta} \leftarrow \widehat{\eta}/|\mathcal{D}^E|$
70 **Return** $\widehat{\eta}$

---

replacing the dynamics and reward $p^i, r^i$ with the input $p, r$.

ROLLOUT collects $K$ trajectories by playing policy $\psi$ in $\mathcal{M}$ for $K$ times, computes the return $G$ of each trajectory, and then returns a dataset $\mathcal{D}$ containing these $K$ returns. In other words, with abuse of notation, we say that the outputted dataset $\mathcal{D} = \{G_k\}_{k \in [\![K]\!]}$ is obtained by collecting $K$ samples $G_k$ from distribution $\eta^{p,r,\psi}$.

### E.3. Time and Space Complexities

The time and space complexities of the subroutines are:

- EXPLORE: *time* $= \mathcal{O}\Big(N\tau\Big)$ for collecting $\tau$ samples from the $N$ environments; *space* $= \mathcal{O}\Big(SAHN\Big)$ for storing the estimates of the transition model of the $N$ environments.

- ERD: *time* $= \mathcal{O}\Big(H\tau^E + H/\epsilon_0\Big)$ for computing the return of each trajectory demontrated by the expert and initializing an estimate of the return distribution; *space* $= \mathcal{O}\Big(H/\epsilon_0\Big)$ to store an estimate of the return distribution.

- PLANNING: *time* $= \mathcal{O}\Big(S^2AH^2/\epsilon_0\Big)$ for doing backward induction in the enlarged discretized MDP; *space* $= \mathcal{O}\Big(SAH^2/\epsilon_0\Big)$ to store a Q-function in the enlarged discretized MDP.

- ROLLOUT: *time* $= \mathcal{O}\Big(KH\Big)$ for simulating $K$ trajectories long $H$; *space* $= \mathcal{O}\Big(K\Big)$ for storing the returns of the $K$ trajectories.

Using these complexities, we derive the complexities of **CATY-UL** and **TRACTOR-UL** as:

- **CATY-UL**: *time* $= \mathcal{O}\Big(N\tau + MN\Big(H\tau^E + S^2AH^2/\epsilon_0\Big)\Big)$ for calling EXPLORE once and then both ERD and PLANNING $MN$ times, where $M$ denotes the number of input utilities to which **CATY-UL** is applied; *space* $=$

---

**Algorithm 6** ROLLOUT

---

**Input:** policy $\psi$, transition model $p$, reward $r$, environment index $i$, number of trajectories $K$

71 $\mathcal{D} \leftarrow \{\}$ // Loop over the number of trajectories:
    **for** $\_ \in \{1, 2, \dots, K\}$ **do**

72        $s \leftarrow s_0^i$  $y \leftarrow 0$                                 `/* y keeps track of the accumulated reward */`

73        **for** $h = 1$ *to* $H$ **do**

74            $a \leftarrow \psi_h(s, y)$

75            $y \leftarrow y + r_h(s, a)$

76            $s \leftarrow s'$ where $s' \sim p_h(\cdot|s, a)$

77        **end**

78        $\mathcal{D} \leftarrow \mathcal{D} \cup \{y\}$

79 **end**

80 **Return** $\mathcal{D}$

---

$\mathcal{O}\Big(SAHN + SAH^2/\epsilon_0\Big)$ where the dominant terms are for storing a transition model in EXPLORE and a Q-function in PLANNING.

- **TRACTOR-UL**: *time* $= \mathcal{O}\Big(N\tau + NH\tau^E + T\Big(NS^2AH^2/\epsilon_0 + NKH + Q_{time}\Big)\Big)$ for calling EXPLORE once, ERD $N$ times, both PLANNING and ROLLOUT $TN$ times, and executing $T$ times the Euclidean projection onto $\overline{\mathfrak{U}}_L$ using some optimization solver ($Q_{time}$ represents this term); *space* $= \mathcal{O}\Big(NH/\epsilon_0 + SAH^2/\epsilon_0 + K + Q_{space}\Big)$ for storing the $N$ estimates of return distributions, for calling PLANNING and ROLLOUT, and for executing some optimization solver for Euclidean projection ($Q_{space}$ represents this term).

Observe that the time and space complexities of the proposed algorithms are polynomial in the amount of data ($\tau, \tau^E$), in the number of environments ($N$), and in the size of the environments ($S, A, H$). Moreover, both **CATY-UL** and **TRACTOR-UL** have time complexities that grow linearly in the number of runs (resp. $M$ and $T$), and note that the complexity of **TRACTOR-UL** grows linearly also in the number of simulated trajectories ($K$) and in the complexity of the optimization solver used for the Euclidean projection ($Q_{time}$). Observe that the complexities depend on $1/\epsilon_0$, where $\epsilon_0 > 0$ is the discretization parameter.

From a theoretical perspective, if we want that, with probability at least $1 - \delta$, the outputs of **CATY-UL** and **TRACTOR-UL** are $\epsilon$-accurate, then, under the assumption that the output of the optimization solver adopted for the Euclidean projection is *exact*, Theorems 5.1 and 5.2 show that it suffices to take $\epsilon_0 = \Theta(\epsilon^2/(HN^2))$, $\tau^E \leqslant \widetilde{\mathcal{O}}\Big(\frac{N^4H^4}{\epsilon^4}\log\frac{1}{\delta}\Big)$, $\tau \leqslant \widetilde{\mathcal{O}}\Big(\frac{N^2SAH^5}{\epsilon^2}\Big(S + \log\frac{1}{\delta}\Big)\Big)$, $T \leqslant \mathcal{O}\Big(\frac{N^4H^4}{\epsilon^4}\Big)$, $K \leqslant \widetilde{\mathcal{O}}\Big(\frac{N^2H^2}{\epsilon^2}\log\frac{1}{\delta}\Big)$, for obtaining a time and space complexity for the algorithms that grow polynomially in $S, A, H, N, \frac{1}{\epsilon}, \log\frac{1}{\delta}, Q_{time}, Q_{space}$.

### E.4. Analysis of **CATY-UL**

**Theorem 5.1.** *Let $L > 0$, $\epsilon, \delta \in (0, 1)$, and let $\mathcal{U} \subseteq \mathfrak{U}_L$ be the set of utilities to classify. For all $i \in [\![N]\!]$, in case $|\mathcal{U}| = 1$, let the number of samples satisfy:*

$$\tau^{E,i} \geqslant \widetilde{\mathcal{O}}\Big(\frac{N^2H^2}{\epsilon^2}\log\frac{N}{\delta}\Big), \ \tau^i \geqslant \widetilde{\mathcal{O}}\Big(\frac{N^2SAH^4}{\epsilon^2}\log\frac{SAHNL}{\delta\epsilon}\Big).$$

*Otherwise, if $|\mathcal{U}| > 1$, let the number of samples satisfy:*

$$\begin{aligned}
\tau^{E,i} &\geqslant \widetilde{\mathcal{O}}\Big(\frac{N^4H^4L^2}{\epsilon^4}\log\frac{HNL}{\delta\epsilon}\Big), \\
\tau^i &\geqslant \widetilde{\mathcal{O}}\Big(\frac{N^2SAH^5}{\epsilon^2}\Big(S + \log\frac{SAHN}{\delta}\Big)\Big).
\end{aligned} \tag{5}$$

*Then, setting $\epsilon_0 = \epsilon^2/(72HL^2N^2)$, w.p. at least $1 - \delta$, for any $\Delta \geqslant 0$, **CATY-UL** correctly classifies* all the $U \in \mathcal{U}$ lying *inside $\mathcal{U}_{\Delta-\epsilon}$ or outside $\mathcal{U}_{\Delta+\epsilon}$.*

*Proof.* Observe that the classification carried out by **CATY-UL** complies with the statement in the theorem as long as we

can demonstrate that:

$$\mathbb{P}_{\{\mathcal{M}^i\}_i, \{\pi^{E,i}\}_i} \left( \sup_{U \in \mathcal{U}} \Big| \sum_{i \in [\![N]\!]} \overline{\mathcal{C}}_{p^i, r^i, \pi^{E,i}}(U) - \sum_{i \in [\![N]\!]} \widehat{\mathcal{C}}^i(U) \Big| \leqslant \epsilon \right) \geqslant 1 - \delta,$$

where $\mathbb{P}_{\{\mathcal{M}^i\}_i, \{\pi^{E,i}\}_i}$ represents the joint probability distribution induced by the exploration phase of **CATY-UL** and the execution of each $\pi^{E,i}$ in the corresponding $\mathcal{M}^i$.

We can rewrite this expression as:

$$\sup_{U \in \mathcal{U}} \Big| \sum_{i \in [\![N]\!]} \overline{\mathcal{C}}_{p^i, r^i, \pi^{E,i}}(U) - \sum_{i \in [\![N]\!]} \widehat{\mathcal{C}}^i(U) \Big| \leqslant \sup_{U \in \mathcal{U}} \sum_{i \in [\![N]\!]} \Big| \overline{\mathcal{C}}_{p^i, r^i, \pi^{E,i}}(U) - \widehat{\mathcal{C}}^i(U) \Big|$$

$$\overset{(1)}{\leqslant} \sum_{i \in [\![N]\!]} \sup_{U \in \mathcal{U}} \Big| \overline{\mathcal{C}}_{p^i, r^i, \pi^{E,i}}(U) - \widehat{\mathcal{C}}^i(U) \Big|,$$

where at (1) we have upper bounded the maximum of a sum with the sum of the maxima. This shows that we can obtain the result as long as we can demonstrate that, for all $i \in [\![N]\!]$, it holds that:

$$\mathbb{P}_{p^i, r^i, \pi^{E,i}} \left( \sup_{U \in \mathcal{U}} \Big| \overline{\mathcal{C}}_{p^i, r^i, \pi^{E,i}}(U) - \widehat{\mathcal{C}}^i(U) \Big| \leqslant \frac{\epsilon}{N} \right) \geqslant 1 - \frac{\delta}{N}; \tag{15}$$

the statement of the theorem would then follow from a union bound. Therefore, let us omit the $i$ index for simplicity, and let us try to obtain the bound in Eq. (15). We can write:

$$\sup_{U \in \mathcal{U}} \Big| \overline{\mathcal{C}}_{p, r, \pi^E}(U) - \widehat{\mathcal{C}}(U) \Big| := \sup_{U \in \mathcal{U}} \Big| \big( J^*(U; p, r) - J^{\pi^E}(U; p, r) \big) - \big( \widehat{J}^*(U) - \widehat{J}^E(U) \big) \Big|$$

$$\overset{(2)}{\leqslant} \sup_{U \in \mathcal{U}} \Big| J^{\pi^E}(U; p, r) - \widehat{J}^E(U) \Big| + \sup_{U \in \mathcal{U}} \Big| J^*(U; p, r) - \widehat{J}^*(U) \Big|$$

$$\overset{(3)}{=} \sup_{U \in \mathcal{U}} \Big| \underset{G \sim \eta^{p, r, \pi^E}}{\mathbb{E}} [U(G)] - \underset{G \sim \widehat{\eta}^E}{\mathbb{E}} [U(G)]$$

$$\pm \underset{G \sim \mathrm{Proj}_{\mathcal{C}}(\eta^{p, r, \pi^E})}{\mathbb{E}} [U(G)] \Big| + \sup_{U \in \mathcal{U}} \Big| J^*(U; p, r) - \widehat{J}^*(U) \Big|$$

$$\overset{(4)}{\leqslant} \sup_{U \in \mathcal{U}} \Big| \underset{G \sim \eta^{p, r, \pi^E}}{\mathbb{E}} [U(G)] - \underset{G \sim \mathrm{Proj}_{\mathcal{C}}(\eta^{p, r, \pi^E})}{\mathbb{E}} [U(G)] \Big|$$

$$+ \sup_{U \in \mathcal{U}} \Big| \underset{G \sim \mathrm{Proj}_{\mathcal{C}}(\eta^{p, r, \pi^E})}{\mathbb{E}} [U(G)] - \underset{G \sim \widehat{\eta}^E}{\mathbb{E}} [U(G)] \Big|$$

$$+ \sup_{U \in \mathcal{U}} \Big| J^*(U; p, r) - \widehat{J}^*(U) \Big|$$

$$\overset{(5)}{\leqslant} \sup_{f: f \text{ is } L\text{-Lipschitz}} \Big| \underset{G \sim \eta^{p, r, \pi^E}}{\mathbb{E}} [f(G)] - \underset{G \sim \mathrm{Proj}_{\mathcal{C}}(\eta^{p, r, \pi^E})}{\mathbb{E}} [f(G)] \Big|$$

$$+ \sup_{U \in \mathcal{U}} \Big| \underset{G \sim \mathrm{Proj}_{\mathcal{C}}(\eta^{p, r, \pi^E})}{\mathbb{E}} [U(G)] - \underset{G \sim \widehat{\eta}^E}{\mathbb{E}} [U(G)] \Big|$$

$$+ \sup_{U \in \mathcal{U}} \Big| J^*(U; p, r) - \widehat{J}^*(U) \Big|$$

$$\overset{(6)}{=} L \cdot w_1 \big( \eta^{p, r, \pi^E}, \mathrm{Proj}_{\mathcal{C}}(\eta^{p, r, \pi^E}) \big)$$

$$+ \sup_{U \in \mathcal{U}} \Big| \underset{G \sim \mathrm{Proj}_{\mathcal{C}}(\eta^{p, r, \pi^E})}{\mathbb{E}} [U(G)] - \underset{G \sim \widehat{\eta}^E}{\mathbb{E}} [U(G)] \Big|$$

$$\sup_{U \in \mathcal{U}} \Big| J^*(U; p, r) - \widehat{J}^*(U) \Big|,$$

where at (2) we have applied triangle inequality, at (3) we use the definition of $J^{\pi^E}(U; p, r)$, and that of $\widehat{J}^E(U)$ (Line 4 of **CATY-UL**), and we have added and subtracted a term, where operator $\mathrm{Proj}_{\mathcal{C}}$ is defined in Eq. (8). We remark that distribution $\eta^{p, r, \pi^E}$ may have a support that grows exponentially in $H$, while both $\widehat{\eta}^E$ and $\mathrm{Proj}_{\mathcal{C}}(\eta^{p, r, \pi^E})$ are supported on

$\mathcal{Y}$. Note that $\widehat{\eta}^E$ and $\text{Proj}_{\mathcal{C}}(\eta^{p,r,\pi^E})$ are different distributions, since the former is the projection on $\mathcal{Y}$ of an estimate of $\eta^{p,r,\pi^E}$. At (4), we apply triangle inequality, at (5) we use the hypothesis that all utilities are $L$-Lipschitz $\mathcal{U} \subseteq \mathfrak{U}_L$, and notice that $\mathfrak{U}_L$ is a subset of all $L$-Lipschitz functions $f : [0, H] \to [0, H]$, and at (6) we apply the duality formula for the 1-Wasserstein distance $w_1$ (see Eq. (6.3) in Chapter 6 of Villani (2008)).

Concerning the case $|\mathcal{U}| = 1$, we apply, for all $i \in [\![N]\!]$, Lemma E.3 with probability $\delta/(2N)$ and accuracy $\epsilon/(3N)$, and Lemma E.5 with probability $\delta/(2N)$ and accuracy $\epsilon/(3N)$, while we bound the 1-Wasserstein distance through Lemma E.1, to obtain, through an application of the union bound, that:

$$\mathbb{P}_{\{\mathcal{M}^i\}_i, \{\pi^{E,i}\}_i} \left( \sup_{U \in \mathcal{U}} \left| \sum_{i \in [\![N]\!]} \overline{\mathcal{C}}_{p^i,r^i,\pi^{E,i}}(U) - \sum_{i \in [\![N]\!]} \widehat{\mathcal{C}}^i(U) \right| \leqslant \right.$$
$$\left. NL\sqrt{2H\epsilon_0} + \epsilon/3 + NHL\epsilon_0 + \epsilon/3 \right) \geqslant 1 - \delta,$$

as long as, for all $i \in [\![N]\!]$:

$$\tau^{E,i} \geqslant \tilde{\mathcal{O}}\left( \frac{N^2 H^2 \log \frac{N}{\delta}}{\epsilon^2} \right),$$
$$\tau^i \geqslant \tilde{\mathcal{O}}\left( \frac{N^2 SAH^4}{\epsilon^2} \log \frac{SAHN}{\delta\epsilon_0} \right).$$

By setting $\epsilon_0 = \frac{\epsilon^2}{72HL^2N^2}$, we obtain that:

$$NL\sqrt{2H\epsilon_0} + NHL\epsilon_0 = \frac{\epsilon}{6} + \frac{\epsilon^2}{72LN} \leqslant \epsilon/3.$$

By putting this bound into the bound on $\tau^i$, we get the result.

When $\mathcal{U}$ is an arbitrary subset of $\mathfrak{U}_L$, we apply, for all $i \in [\![N]\!]$, Lemma E.4 with probability $\delta/(2N)$ and accuracy $\epsilon/(3N)$, and Lemma E.13 with probability $\delta/(2N)$ and accuracy $\epsilon/(3N)$, while we bound the 1-Wasserstein distance through Lemma E.1, to obtain, through an application of the union bound, that:

$$\mathbb{P}_{\{\mathcal{M}^i\}_i, \{\pi^{E,i}\}_i} \left( \sup_{U \in \mathcal{U}} \left| \sum_{i \in [\![N]\!]} \overline{\mathcal{C}}_{p^i,r^i,\pi^{E,i}}(U) - \sum_{i \in [\![N]\!]} \widehat{\mathcal{C}}^i(U) \right| \leqslant \right.$$
$$\left. NL\sqrt{2H\epsilon_0} + \epsilon/3 + NHL\epsilon_0 + \epsilon/3 \right) \geqslant 1 - \delta,$$

as long as, for all $i \in [\![N]\!]$:

$$\tau^{E,i} \geqslant \tilde{\mathcal{O}}\left( \frac{N^2 H^3}{\epsilon^2 \epsilon_0} \log \frac{HN}{\delta\epsilon_0} \right),$$
$$\tau^i \geqslant \tilde{\mathcal{O}}\left( \frac{N^2 SAH^5}{\epsilon^2} \left( S + \log \frac{SAHN}{\delta} \right) \right).$$

Again, by setting $\epsilon_0 = \frac{\epsilon^2}{72HL^2N^2}$, we obtain that:

$$NL\sqrt{2H\epsilon_0} + NHL\epsilon_0 = \frac{\epsilon}{6} + \frac{\epsilon^2}{72LN} \leqslant \epsilon/3.$$

By putting this bound into the bounds on $\tau^{E,i}$ and $\tau^i$, we get the result. $\qquad\square$

### E.4.1. LEMMAS ON THE EXPERT'S RETURN DISTRIBUTION

**Lemma E.1.** *Let the projection operator Proj$_{\mathcal{C}}$ be defined as in Eq. (8), over set $\mathcal{Y}$ with discretization $\epsilon_0$. Then, for all $i \in [\![N]\!]$, it holds that:*

$$w_1(\eta^{p^i,r^i,\pi^{E,i}}, \text{Proj}_{\mathcal{C}}(\eta^{p^i,r^i,\pi^{E,i}})) \leqslant \sqrt{2H\epsilon_0}.$$

*Proof.* For the sake of simplicity, we omit index $i \in [\![N]\!]$, but the following derivation can be applied to all the $N$ demonstrations.

By applying Lemma 5.2 of Rowland et al. (2024), replacing term $1/(1-\gamma)$ with horizon $H$, we get:

$$w_1(\eta^{p,r,\pi^E}, \mathrm{Proj}_\mathcal{C}(\eta^{p,r,\pi^E})) \leqslant \sqrt{H}\ell_2(\eta^{p,r,\pi^E}, \mathrm{Proj}_\mathcal{C}(\eta^{p,r,\pi^E})).$$

Similarly to the proof of Proposition 3 of Rowland et al. (2018), we can write:

$$
\begin{aligned}
\ell_2^2(\eta^{p,r,\pi^E}, \mathrm{Proj}_\mathcal{C}(\eta^{p,r,\pi^E})) &\overset{(1)}{:=} \int_\mathbb{R} (F_{\eta^{p,r,\pi^E}}(y) - F_{\mathrm{Proj}_\mathcal{C}(\eta^{p,r,\pi^E})}(y))^2 dy \\
&\overset{(2)}{=} \int_0^H (F_{\eta^{p,r,\pi^E}}(y) - F_{\mathrm{Proj}_\mathcal{C}(\eta^{p,r,\pi^E})}(y))^2 dy \\
&\overset{(3)}{=} \sum_{j \in [\![d-1]\!]} \int_{y_j}^{y_{j+1}} (F_{\eta^{p,r,\pi^E}}(y) - F_{\mathrm{Proj}_\mathcal{C}(\eta^{p,r,\pi^E})}(y))^2 dy \\
&\qquad + \int_{y_d}^H (F_{\eta^{p,r,\pi^E}}(y) - F_{\mathrm{Proj}_\mathcal{C}(\eta^{p,r,\pi^E})}(y))^2 dy \\
&\overset{(4)}{\leqslant} \sum_{j \in [\![d-1]\!]} \int_{y_j}^{y_{j+1}} (F_{\eta^{p,r,\pi^E}}(y) - F_{\mathrm{Proj}_\mathcal{C}(\eta^{p,r,\pi^E})}(y))^2 dy + \epsilon_0 \\
&\overset{(5)}{\leqslant} \sum_{j \in [\![d-1]\!]} \int_{y_j}^{y_{j+1}} (F_{\eta^{p,r,\pi^E}}(y_{j+1}) - F_{\eta^{p,r,\pi^E}}(y_j))^2 dy + \epsilon_0 \\
&= \sum_{j \in [\![d-1]\!]} (y_{j+1} - y_j)(F_{\eta^{p,r,\pi^E}}(y_{j+1}) - F_{\eta^{p,r,\pi^E}}(y_j))^2 + \epsilon_0 \\
&\overset{(6)}{=} \epsilon_0 \sum_{j \in [\![d-1]\!]} (F_{\eta^{p,r,\pi^E}}(y_{j+1}) - F_{\eta^{p,r,\pi^E}}(y_j))^2 + \epsilon_0 \\
&\overset{(7)}{\leqslant} \epsilon_0 \Big( \sum_{j \in [\![d-1]\!]} (F_{\eta^{p,r,\pi^E}}(y_{j+1}) - F_{\eta^{p,r,\pi^E}}(y_j)) \Big)^2 + \epsilon_0 \\
&\overset{(8)}{=} \epsilon_0 (F_{\eta^{p,r,\pi^E}}(y_d) - F_{\eta^{p,r,\pi^E}}(y_1))^2 + \epsilon_0 \\
&\leqslant 2\epsilon_0,
\end{aligned}
$$

where at (1) we have applied the definition of $\ell_2$ distance (Eq. (7)), at (2) we recognize that the two distributions $\eta^{p,r,\pi^E}, \mathrm{Proj}_\mathcal{C}(\eta^{p,r,\pi^E})$ are defined on $[0, H]$, at (3) we use the additivity property of the integral, using notation $\mathcal{Y} := \{0, \epsilon_0, 2\epsilon_0, \ldots, \lfloor H/\epsilon_0 \rfloor \epsilon_0\}$, $d := |\mathcal{Y}| = \lfloor H/\epsilon_0 \rfloor + 1$, $y_1 := 0, y_2 := \epsilon_0, y_3 := 2\epsilon_0, \ldots, y_d := \lfloor H/\epsilon_0 \rfloor \epsilon_0$, (notation introduced in Section 5). At (4) we upper bound $\int_{y_d}^H (F_{\eta^{p,r,\pi^E}}(y) - F_{\mathrm{Proj}_\mathcal{C}(\eta^{p,r,\pi^E})}(y))^2 dy \leqslant \int_{y_d}^H dy = H - y_d = H - \lfloor H/\epsilon_0 \rfloor \epsilon_0 = \epsilon_0(H/\epsilon_0 - \lfloor H/\epsilon_0 \rfloor) \leqslant \epsilon_0$ since the difference of cumulative distribution functions is bounded by 1. At (5), thanks to the definition of the projection operator $\mathrm{Proj}_\mathcal{C}$ (Eq. (8)), we notice that, for $y \in [y_j, y_{j+1}]$, it holds that $F_{\mathrm{Proj}_\mathcal{C}(\eta^{p,r,\pi^E})}(y) \in [F_{\eta^{p,r,\pi^E}}(y_j), F_{\eta^{p,r,\pi^E}}(y_{j+1})]$, thus we can upper bound the integrand through the maximum, constant, difference of cumulative distribution functions. At (6) we use the definition of set $\mathcal{Y}$, i.e., an $\epsilon_0$-covering of the $[0, H]$ interval, at (7) we use the Cauchy-Schwarz's inequality $\sum_j (x_j)^2 \leqslant (\sum_j x_j)^2$ for $x_j \geqslant 0$, and noticed that the summands are always non-negative, at (8) we apply a telescoping argument.

The result follows by taking the square root of both sides. $\qquad\square$

**Lemma E.2.** *Let $i \in [\![N]\!]$, and let $f \in [0, H]^d$ be an arbitrary $d$-dimensional vector. Denote by $G_1, G_2, \ldots, G_{\tau^{E,i}} \overset{i.i.d.}{\sim} \eta^{p^i, r^i, \pi^{E,i}}$ the random variables representing the returns of the $\tau^{E,i}$ trajectories inside dataset $\mathcal{D}^{E,i}$. Let $\hat\eta^{E,i}$ be the random output of Algorithm 5 that depends on the random variables $G_1, G_2, \ldots, G_{\tau^{E,i}}$. Then, it holds that:*

$$\mathbb{E}_{G_1, G_2, \ldots, G_{\tau^{E,i}} \sim \eta^{p^i, r^i, \pi^{E,i}}} \Big[ \mathbb{E}_{y \sim \hat\eta^{E,i}} \big[ f(y) \big] \Big] = \mathbb{E}_{y \sim Proj_\mathcal{C}(\eta^{p^i, r^i, \pi^{E,i}})} \Big[ f(y) \Big].$$

*Proof.* We omit index $i$ for simplicity, but the proof can be carried out for all $i \in [\![N]\!]$ independently. To prove the statement, we use the notation described in Appendix E.2 for the Dirac delta, to provide an explicit representation of both the distribution $\text{Proj}_{\mathcal{C}}(\eta^{p,r,\pi^E})$ and the "random" distribution $\widehat{\eta}^E$.

We consider distribution $\eta^{p,r,\pi^E}$ supported on $\mathcal{Z} := \{z_1, z_2, \ldots, z_M\} \subseteq [0, H]$, while distributions $\text{Proj}_{\mathcal{C}}(\eta^{p,r,\pi^E}), \widehat{\eta}^E$ are supported on set $\mathcal{Y} = \{y_1, y_2, \ldots, y_d\} \subseteq [0, H]$.

W.r.t. distribution $\text{Proj}_{\mathcal{C}}(\eta^{p,r,\pi^E})$, we can write:

$$
\begin{aligned}
\text{Proj}_{\mathcal{C}}(\eta^{p,r,\pi^E}) &= \text{Proj}_{\mathcal{C}}\Big( \sum_{k\in[\![M]\!]} \eta^{p,r,\pi^E}(z_k)\delta_{z_k} \Big) \\
&\overset{(1)}{=} \sum_{k\in[\![M]\!]} \eta^{p,r,\pi^E}(z_k)\text{Proj}_{\mathcal{C}}(\delta_{z_k}) \\
&\overset{(2)}{=} \sum_{k\in[\![M]\!]} \eta^{p,r,\pi^E}(z_k)\Big( \delta_{y_1}\mathbb{1}\{z_k \le y_1\} + \delta_{y_d}\mathbb{1}\{z_k > y_d\} \\
&\quad + \sum_{j\in[\![d-1]\!]} \Big( \frac{y_{j+1}-z_k}{y_{j+1}-y_j}\delta_{y_j} + \frac{z_k-y_j}{y_{j+1}-y_j}\delta_{y_{j+1}} \Big)\mathbb{1}\{z_k \in (y_j, y_{j+1}]\} \Big) \\
&= \delta_{y_1} \sum_{k\in[\![M]\!]} \eta^{p,r,\pi^E}(z_k)\Big( \mathbb{1}\{z_k \le y_1\} + \frac{y_2-z_k}{y_2-y_1}\mathbb{1}\{z_k \in (y_1, y_2]\} \Big) \\
&\quad + \sum_{j\in\{2,\ldots,d-1\}} \delta_{y_j}\Big( \sum_{k\in[\![M]\!]} \eta^{p,r,\pi^E}(z_k)\Big( \frac{y_{j+1}-z_k}{y_{j+1}-y_j}\mathbb{1}\{z_k \in (y_i, y_{j+1}]\} \\
&\quad + \frac{z_k-y_{j-1}}{y_i-y_{j-1}}\mathbb{1}\{z_k \in (y_{j-1}, y_i]\} \Big) \Big) \\
&\quad + \delta_{y_d} \sum_{k\in[\![M]\!]} \eta^{p,r,\pi^E}(z_k)\Big( \mathbb{1}\{z_k > y_d\} + \frac{z_k-y_{d-1}}{y_d-y_{d-1}}\mathbb{1}\{z_k \in (y_{d-1}, y_d]\} \Big),
\end{aligned}
$$

where at (1) we have applied the extension in Eq. (9) of the projection operator $\text{Proj}_{\mathcal{C}}$ to finite mixtures of Dirac distributions, and at (2) we have applied its definition (Eq. (8)).

Concerning distribution $\widehat{\eta}^E$, based on Algorithm 5, we can write:

$$
\begin{aligned}
\widehat{\eta}^E &= \frac{\delta_{y_1}}{\tau^E}\Big( \sum_{t\in[\![\tau^E]\!]} \Big( \mathbb{1}\{G_t \le y_1\} + \frac{y_2-G_t}{y_2-y_1}\mathbb{1}\{G_t \in (y_1, y_2]\} \Big) \Big) \\
&\quad + \sum_{j\in\{2,\ldots,d-1\}} \frac{\delta_{y_j}}{\tau^E}\Big( \sum_{t\in[\![\tau^E]\!]} \Big( \frac{y_{j+1}-G_t}{y_{j+1}-y_j}\mathbb{1}\{G_t \in (y_i, y_{j+1}]\} \\
&\quad + \frac{G_t-y_{j-1}}{y_i-y_{j-1}}\mathbb{1}\{G_t \in (y_{j-1}, y_i]\} \Big) \Big) \\
&\quad + \frac{\delta_{y_d}}{\tau^E}\Big( \sum_{t\in[\![\tau^E]\!]} \Big( \mathbb{1}\{G_t > y_d\} + \frac{G_t-y_{d-1}}{y_d-y_{d-1}}\mathbb{1}\{G_t \in (y_{d-1}, y_d]\} \Big) \Big).
\end{aligned}
$$

Now, if we take the expectation of the random vector $\widehat{\eta}^E$ w.r.t. $\eta^{p,r,\pi^E}$, we get:

$$
\begin{aligned}
&\mathbb{E}_{G_1,G_2,\ldots,G_{\tau^E}\sim\eta^{p,r,\pi^E}}\Big[ \widehat{\eta}^E \Big] \\
&= \mathbb{E}_{G_1,G_2,\ldots,G_{\tau^E}\sim\eta^{p,r,\pi^E}}\Big[ \frac{\delta_{y_1}}{\tau^E}\Big( \sum_{t\in[\![\tau^E]\!]} \Big( \mathbb{1}\{G_t \le y_1\} + \frac{y_2-G_t}{y_2-y_1}\mathbb{1}\{G_t \in (y_1, y_2]\} \Big) \Big) \\
&\quad + \sum_{j\in\{2,\ldots,d-1\}} \frac{\delta_{y_j}}{\tau^E}\Big( \sum_{t\in[\![\tau^E]\!]} \Big( \frac{y_{j+1}-G_t}{y_{j+1}-y_j}\mathbb{1}\{G_t \in (y_i, y_{j+1}]\} \\
&\quad + \frac{G_t-y_{j-1}}{y_i-y_{j-1}}\mathbb{1}\{G_t \in (y_{j-1}, y_i]\} \Big) \Big)
\end{aligned}
$$

$$+ \frac{\delta_{y_d}}{\tau^E} \Big( \sum_{t \in \llbracket \tau^E \rrbracket} \Big( \mathbb{1}\{G_t > y_d\} + \frac{G_t - y_{d-1}}{y_d - y_{d-1}} \mathbb{1}\{G_t \in (y_{d-1}, y_d]\} \Big) \Big) \Big]$$

$$\overset{(3)}{=} \mathbb{E}_{G \sim \eta^{p,r,\pi^E}} \Big[ \delta_{y_1} \Big( \mathbb{1}\{G \leqslant y_1\} + \frac{y_2 - G}{y_2 - y_1} \mathbb{1}\{G \in (y_1, y_2]\} \Big)$$

$$+ \sum_{j \in \{2, \ldots, d-1\}} \delta_{y_j} \Big( \frac{y_{j+1} - G}{y_{j+1} - y_j} \mathbb{1}\{G \in (y_i, y_{j+1}]\}$$

$$+ \frac{G - y_{j-1}}{y_i - y_{j-1}} \mathbb{1}\{G \in (y_{j-1}, y_i]\} \Big)$$

$$+ \delta_{y_d} \Big( \mathbb{1}\{G > y_d\} + \frac{G - y_{d-1}}{y_d - y_{d-1}} \mathbb{1}\{G \in (y_{d-1}, y_d]\} \Big) \Big]$$

$$\overset{(4)}{=} \delta_{y_1} \sum_{k \in \llbracket M \rrbracket} \eta^{p,r,\pi^E}(z_k) \Big( \mathbb{1}\{z_k \leqslant y_1\} + \frac{y_2 - z_k}{y_2 - y_1} \mathbb{1}\{z_k \in (y_1, y_2]\} \Big)$$

$$+ \sum_{j \in \{2, \ldots, d-1\}} \delta_{y_j} \Big( \sum_{k \in \llbracket M \rrbracket} \eta^{p,r,\pi^E}(z_k) \Big( \frac{y_{j+1} - z_k}{y_{j+1} - y_j} \mathbb{1}\{z_k \in (y_i, y_{j+1}]\}$$

$$+ \frac{z_k - y_{j-1}}{y_i - y_{j-1}} \mathbb{1}\{z_k \in (y_{j-1}, y_i]\} \Big) \Big)$$

$$+ \delta_{y_d} \sum_{k \in \llbracket M \rrbracket} \eta^{p,r,\pi^E}(z_k) \Big( \mathbb{1}\{z_k > y_d\} + \frac{z_k - y_{d-1}}{y_d - y_{d-1}} \mathbb{1}\{z_k \in (y_{d-1}, y_d]\} \Big)$$

$$\overset{(5)}{=} \text{Proj}_{\mathcal{C}}(\eta^{p,r,\pi^E}),$$

where at (3) we use the fact that $G_1, G_2, \ldots, G_{\tau^E}$ are independent and identically distributed, at (4) we apply the linearity of the expectation, we notice that $\delta_{y_j}$ does not depend on $G$ for all $j \in \llbracket d \rrbracket$, and we notice that, for any $y \in \mathcal{Y}$, it holds that $\mathbb{E}_{G \sim \eta^{p,r,\pi^E}} \big[ \mathbb{1}\{G \leqslant y\} \big] = \eta^{p,r,\pi^E}(G \leqslant y) = \sum_{k \in \llbracket M \rrbracket} \eta^{p,r,\pi^E}(z_k) \mathbb{1}\{z_k \leqslant y\}$, where we have abused notation by writing $\eta^{p,r,\pi^E}(G \leqslant y)$ to mean the probability, under distribution $\eta^{p,r,\pi^E}$, that event $\{G \leqslant y\}$ happens. Moreover, similarly, we notice that, for any $y, y' \in \mathcal{Y}$, it holds that $\mathbb{E}_{G \sim \eta^{p,r,\pi^E}} \big[ G \cdot \mathbb{1}\{G \in [y, y']\} \big] = \sum_{k \in \llbracket M \rrbracket} z_k \eta^{p,r,\pi^E}(z_k) \mathbb{1}\{z_k \in [y, y']\}$. At (5) we simply recognize $\text{Proj}_{\mathcal{C}}(\eta^{p,r,\pi^E})$ using the previous expression.

This concludes the proof because the equality of the Dirac delta representations means that the expectations of any function w.r.t. these two distributions coincide. $\square$

**Lemma E.3.** *Let $i \in \llbracket N \rrbracket$ and let $\epsilon, \delta \in (0, 1)$. If $|\mathcal{U}| = 1$, then, with probability at least $1 - \delta$, we have:*

$$\sup_{U \in \mathcal{U}} \Big| \mathbb{E}_{G \sim \text{Proj}_{\mathcal{C}}(\eta^{p^i, r^i, \pi^{E,i}})} [U(G)] - \mathbb{E}_{G \sim \widehat{\eta}^{E,i}} [U(G)] \Big| \leqslant \epsilon,$$

*as long as:*

$$\tau^E \geqslant c \frac{H^2 \log \frac{2}{\delta}}{\epsilon^2},$$

*where $c$ is some positive constant.*

*Proof.* Let $U$ be the only function inside $\mathcal{U}$. Let us omit index $i$ for simplicity. Then, we can write:

$$\Big| \mathbb{E}_{G \sim \widehat{\eta}^E} [U(G)] - \mathbb{E}_{G \sim \text{Proj}_{\mathcal{C}}(\eta^{p,r,\pi^E})} [U(G)] \Big| \overset{(1)}{=} \Big| \mathbb{E}_{G \sim \widehat{\eta}^E} [U(G)] - \mathbb{E}_{\eta^{p,r,\pi^E}} \Big[ \mathbb{E}_{G \sim \widehat{\eta}^E} [U(G)] \Big] \Big|$$

$$\overset{(2)}{\leqslant} cH \sqrt{\frac{\log \frac{2}{\delta}}{\tau^E}},$$

where at (1) we have applied Lemma E.2, and at (2) we have applied the Hoeffding's inequality noticing that function $U$ is bounded in $[0, H]$, and denoting with $c$ some positive constant.

By imposing:

$$cH\sqrt{\frac{\log\frac{2}{\delta}}{\tau^E}} \leqslant \epsilon,$$

and solving w.r.t. $\tau^E$, we get the result. $\qquad\qquad\square$

**Lemma E.4.** *Let $i \in [\![N]\!]$ and let $\epsilon, \delta \in (0, 1)$. Then, with probability at least $1 - \delta$, we have:*

$$\sup_{U\in\mathcal{U}}\Big|\mathbb{E}_{G\sim Proj_{\mathcal{C}}(\eta^{p^i,r^i,\pi^{E,i}})}[U(G)] - \mathbb{E}_{G\sim\widehat{\eta}^{E,i}}[U(G)]\Big| \leqslant \epsilon,$$

*as long as:*

$$\tau^E \geqslant \widetilde{\mathcal{O}}\Big(\frac{H^3}{\epsilon^2\epsilon_0}\log\frac{H}{\delta\epsilon_0}\Big).$$

*Proof.* Again, let us omit index $i$ for simplicity. First, for all possible functions $U \in \mathcal{U}$, we denote by $\overline{U} \in \overline{\mathfrak{U}}_L$ the function in $\overline{\mathfrak{U}}_L$ that takes on the values that the function $U$ assigns to the points of set $\mathcal{Y}$. This permits us to write:

$$\sup_{U\in\mathcal{U}}\Big|\mathbb{E}_{G\sim\widehat{\eta}^E}[U(G)] - \mathbb{E}_{G\sim\text{Proj}_{\mathcal{C}}(\eta^{p,r,\pi^E})}[U(G)]\Big|$$

$$= \sup_{\overline{U}\in\overline{\mathfrak{U}}_L}\Big|\mathbb{E}_{G\sim\widehat{\eta}^E}[\overline{U}(G)] - \mathbb{E}_{G\sim\text{Proj}_{\mathcal{C}}(\eta^{p,r,\pi^E})}[\overline{U}(G)]\Big|$$

$$\overset{(1)}{\leqslant} \sup_{\overline{U}\in[0,H]^d}\Big|\mathbb{E}_{G\sim\widehat{\eta}^E}[\overline{U}(G)] - \mathbb{E}_{G\sim\text{Proj}_{\mathcal{C}}(\eta^{p,r,\pi^E})}[\overline{U}(G)]\Big|$$

$$\overset{(2)}{=} \sup_{\overline{U}\in[0,H]^d}\Big|\mathbb{E}_{G\sim\widehat{\eta}^E}[\overline{U}(G)] - \mathbb{E}_{\eta^{p,r,\pi^E}}\Big[\mathbb{E}_{G\sim\widehat{\eta}^E}[\overline{U}(G)]\Big]\Big|,$$

where at (1) we upper bound by considering all the possible vectors $\overline{U} \in [0, H]^d$, and at (2) we apply Lemma E.2.

Now, similarly to the proof of Lemma 7.2 in Agarwal et al. (2021), we construct an $\epsilon'$-covering of set $[0, H]^d$, call it $\mathcal{N}_{\epsilon'}$, with $|\mathcal{N}_{\epsilon'}| \leqslant (1 + 2H\sqrt{d}/\epsilon')^d$ such that, for all $f \in [0, H]^d$, there exists $f' \in \mathcal{N}_{\epsilon'}$ for which $\|f - f'\|_2 \leqslant \epsilon'$. By applying a union bound over all $f' \in \mathcal{N}_{\epsilon'}$ and Lemma E.3, we have that, with probability at least $1 - \delta$, for all $f' \in \mathcal{N}_{\epsilon'}$, it holds that:

$$\Big|\mathbb{E}_{G\sim\widehat{\eta}^E}[f'(G)] - \mathbb{E}_{\eta^{p,r,\pi^E}}\Big[\mathbb{E}_{G\sim\widehat{\eta}^E}[f'(G)]\Big]\Big| \leqslant cH\sqrt{\frac{d\log\frac{2(1+2H\sqrt{d}/\epsilon')}{\delta}}{\tau^E}}. \tag{16}$$

Next, for any $f \in [0, H]^d$, denote its closest points (in 2-norm) from $\mathcal{N}_{\epsilon'}$ as $f'$. Then, we have:

$$\Big|\mathbb{E}_{G\sim\widehat{\eta}^E}[f(G)] - \mathbb{E}_{\eta^{p,r,\pi^E}}\Big[\mathbb{E}_{G\sim\widehat{\eta}^E}[f(G)]\Big]\Big|$$

$$= \Big|\mathbb{E}_{G\sim\widehat{\eta}^E}[f(G)] - \mathbb{E}_{\eta^{p,r,\pi^E}}\Big[\mathbb{E}_{G\sim\widehat{\eta}^E}[f(G)]\Big] \pm \Big(\mathbb{E}_{G\sim\widehat{\eta}^E}[f'(G)] - \mathbb{E}_{\eta^{p,r,\pi^E}}\Big[\mathbb{E}_{G\sim\widehat{\eta}^E}[f'(G)]\Big]\Big)\Big|$$

$$\overset{(3)}{\leqslant} \Big|\mathbb{E}_{G\sim\widehat{\eta}^E}[f'(G)] - \mathbb{E}_{\eta^{p,r,\pi^E}}\Big[\mathbb{E}_{G\sim\widehat{\eta}^E}[f'(G)]\Big]\Big|$$

$$+ \Big|\mathbb{E}_{G\sim\widehat{\eta}^E}[f(G) - f'(G)]\Big| + \Big|\mathbb{E}_{\eta^{p,r,\pi^E}}\Big[\mathbb{E}_{G\sim\widehat{\eta}^E}[f(G) - f'(G)]\Big]\Big|$$

$$\overset{(4)}{\leqslant} cH\sqrt{\frac{d\log\frac{2(1+2H\sqrt{d}/\epsilon')}{\delta}}{\tau^E}} + 2\epsilon'$$

$$\overset{(5)}{\leqslant} c'H\sqrt{\frac{d\log\frac{Hd\tau^E}{\delta}}{\tau^E}}$$

where at (3) we apply triangle inequality, at (4) we apply the result in Eq. (16), and the fact that, by definition of $\epsilon'$-covering, $\|f - f'\|_2 \leqslant \epsilon'$ entails that $|f(y) - f(y')| \leqslant \epsilon'$ for all $y \in \mathcal{Y}$; at (5) we set $\epsilon' = 1/\tau^E$, and we simplify.

The result follows by upper bounding $d \leqslant H/\epsilon_0 + 1$, and then by setting:

$$c'' H \sqrt{\frac{H \log \frac{H\tau^E}{\delta \epsilon_0}}{\epsilon_0 \tau^E}} \leqslant \epsilon, \tag{17}$$

and solving w.r.t. $\tau^E$, and noticing that for all $\tau^E$ greater than some constant, we can get rid of the logarithmic terms in $\tau^E$. $\qquad\square$

### E.4.2. Lemmas on the Optimal Performance for Single Utility

In this section, we will omit index $i \in [\![N]\!]$ since the following derivations can be carried out for each $i$.

We denote the arbitrary MDP in $\{\mathcal{M}^i\}_i$ as $\mathcal{M} = (\mathcal{S}, \mathcal{A}, H, s_0, p, r)$, and its analogous with discretized reward $\overline{r}$, defined at all $(s, a, h) \in \mathcal{S} \times \mathcal{A} \times [\![H]\!]$ as $\overline{r}_h(s, a) := \Pi_{\mathcal{R}}[r_h(s, a)]$, as $\overline{\mathcal{M}} := (\mathcal{S}, \mathcal{A}, H, s_0, p, \overline{r})$. We denote the analogous MDPs with empirical transition model $\widehat{p}$ as $\widehat{\mathcal{M}} = (\mathcal{S}, \mathcal{A}, H, s_0, \widehat{p}, r)$ and $\widehat{\overline{\mathcal{M}}} := (\mathcal{S}, \mathcal{A}, H, s_0, \widehat{p}, \overline{r})$.

Given any utility $U \in \mathfrak{U}_L$, we denote the corresponding RS-MDPs, respectively, as $\mathcal{M}_U, \overline{\mathcal{M}}_U, \widehat{\mathcal{M}}_U, \widehat{\overline{\mathcal{M}}}_U$. Concerning the discretized RS-MDPs $\overline{\mathcal{M}}_U$ and $\widehat{\overline{\mathcal{M}}}_U$, we denote the corresponding enlarged state space MDPs, respectively, as $\mathfrak{E}[\overline{\mathcal{M}}_U] = (\{\mathcal{S} \times \mathcal{Y}_h\}_h, \mathcal{A}, H, (s_0, 0), \mathfrak{p}, \mathfrak{r})$ and $\mathfrak{E}[\widehat{\overline{\mathcal{M}}}_U] = (\{\mathcal{S} \times \mathcal{Y}_h\}_h, \mathcal{A}, H, (s_0, 0), \widehat{\mathfrak{p}}, \mathfrak{r})$, where we decided to define such enlarged state space MDPs using the state space $\{\mathcal{S} \times \mathcal{Y}_h\}_h$ considered by Algorithm 4 (PLANNING) instead of, respectively, $\{\mathcal{S} \times \mathcal{G}_h^{p,\overline{r}}\}_h$ and $\{\mathcal{S} \times \mathcal{G}_h^{\widehat{p},\overline{r}}\}_h$. Thus, the transition models $\mathfrak{p}$ and $\widehat{\mathfrak{p}}$, from any $h \in [\![H]\!]$ and $(s, y, a) \in \mathcal{S} \times \mathcal{Y}_h \times \mathcal{A}$, assign to the next state $(s', y') \in \mathcal{S} \times \mathcal{Y}_{h+1}$ the probability: $\mathfrak{p}_h(s', y'|s, y, a) := p_h(s'|s, a)\mathbb{1}\{y' = y + \overline{r}_h(s, a)\}$ and $\widehat{\mathfrak{p}}_h(s', y'|s, y, a) := \widehat{p}_h(s'|s, a)\mathbb{1}\{y' = y + \overline{r}_h(s, a)\}$. Moreover, the reward function $\mathfrak{r}$, in any $h \in [\![H]\!]$ and $(s, y, a) \in \mathcal{S} \times \mathcal{Y}_h \times \mathcal{A}$, is $\mathfrak{r}_h(s, y, a) = 0$ if $h < H$, and $\mathfrak{r}_h(s, y, a) = U(y + \overline{r}_h(s, a))$ if $h = H$.

We will make extensive use of notation for $V$- and $Q$- functions introduced in Appendix B.

We are now ready to proceed with the analysis. In general, the analysis shares similarities to that of Theorem 3 of Wu & Xu (2023), but we use results also from Azar et al. (2013) to obtain tighter bounds.

**Lemma E.5.** *Let $\epsilon, \delta \in (0, 1)$. For any fixed $L$-Lipschitz utility function $U \in \mathfrak{U}_L$, it suffices to execute* `CATY-UL` *with:*

$$\tau \leqslant \widetilde{\mathcal{O}}\left(\frac{SAH^4}{\epsilon^2} \log \frac{SAH}{\delta \epsilon_0}\right),$$

*to obtain $\left|J^*(U; p, r) - \widehat{J}^*(U)\right| \leqslant HL\epsilon_0 + \epsilon$ w.p. $1 - \delta$.*

*Proof.* For an arbitrary utility $U \in \mathfrak{U}_L$, we can write:

$$
\begin{aligned}
|J^*(U; p, r) - \widehat{J}^*(U)| &\overset{(1)}{=} |J^*(U; p, r) - \widehat{J}^*(U) \underline{\pm J^*(\mathfrak{p}, \mathfrak{r})}| \\
&\overset{(2)}{\leqslant} |J^*(U; p, r) - J^*(\mathfrak{p}, \mathfrak{r})| + |J^*(\mathfrak{p}, \mathfrak{r}) - \widehat{J}^*(U)| \\
&\overset{(3)}{=} |J^*(U; p, r) - J^*(\mathfrak{p}, \mathfrak{r})| + |J^*(\mathfrak{p}, \mathfrak{r}) - J^*(\widehat{\mathfrak{p}}, \mathfrak{r})| \\
&\overset{(4)}{\leqslant} HL\epsilon_0 + |J^*(\mathfrak{p}, \mathfrak{r}) - J^*(\widehat{\mathfrak{p}}, \mathfrak{r})| \\
&= HL\epsilon_0 + |V_1^*(s_0, 0; \mathfrak{p}, \mathfrak{r}) - V_1^*(s_0, 0; \widehat{\mathfrak{p}}, \mathfrak{r})| \\
&\leqslant HL\epsilon_0 + \max_{h \in [\![H]\!], (s, y, a) \in \mathcal{S} \times \mathcal{Y}_h \times \mathcal{A}} |Q_h^*(s, y, a; \mathfrak{p}, \mathfrak{r}) - Q_h^*(s, y, a; \widehat{\mathfrak{p}}, \mathfrak{r})| \\
&\overset{(5)}{\leqslant} HL\epsilon_0 + \epsilon',
\end{aligned}
$$

where at (1) we add and subtract the optimal expected utility in the enlarged MDP $\mathfrak{E}[\overline{\mathcal{M}}_U]$ considered by Algorithm 4, but with the true transition model $\mathfrak{p}$. At (2) we apply triangle inequality, at (3) we recognize that the estimate $\widehat{J}^*(U)$ used

in **CATY-UL** and outputted by PLANNING (Algorithm 4) is the optimal expected utility for the discretized problem with estimated dynamics $\widehat{\mathfrak{p}}$, at (4) we use Proposition 3 of Wu & Xu (2023), since $U$ is $L$-Lipschitz, and at (5) we apply Lemma E.6 to bound the distance between $Q$-functions.

By setting:

$$\underbrace{c\sqrt{\frac{H^3 \log \frac{4SAHd}{\delta}}{n}}}_{\leqslant \epsilon/3} + \underbrace{cH^2\left(\frac{\log \frac{16SAHd}{\delta}}{n}\right)^{3/4}}_{\leqslant \epsilon/3} + \underbrace{cH^3\frac{\log \frac{16SAHd}{\delta}}{n}}_{\leqslant \epsilon/3} \leqslant \epsilon,$$

and solving w.r.t. $\epsilon$:

$$\begin{cases} n \geqslant c'\frac{H^3 \log \frac{4SAHd}{\delta}}{\epsilon^2} \\ n \geqslant c''\frac{H^{8/3} \log \frac{16SAHd}{\delta}}{\epsilon^{4/3}} \\ n \geqslant c'''\frac{H^3 \log \frac{16SAHd}{\delta}}{\epsilon} \end{cases}.$$

Taking the largest bound, we get:

$$n \geqslant c\frac{H^3 \log \frac{16SAHd}{\delta}}{\epsilon^2},$$

for some positive constant $c$. Since $d \leqslant H/\epsilon_0 + 1$, we can write:

$$\tau \geqslant c'\frac{SAH^4 \log \frac{c''SAH}{\delta\epsilon_0}}{\epsilon^2},$$

for some positive constants $c', c''$, where we used that $\tau = SAHn$. $\qquad\square$

The proof of the following lemma is organized in many lemmas, and is based on the proof of Theorem 1 of Azar et al. (2013).

**Lemma E.6.** *For any $\delta \in (0,1)$, we have:*

$$\max_{h\in[\![H]\!],(s,y,a)\in\mathcal{S}\times\mathcal{Y}_h\times\mathcal{A}} |Q_h^*(s,y,a;\mathfrak{p},\mathfrak{r}) - Q_h^*(s,y,a;\widehat{\mathfrak{p}},\mathfrak{r})| \leqslant \epsilon',$$

*w.p. at least $1-\delta$, where $\epsilon'$ is defined as:*

$$\epsilon' := c\sqrt{\frac{H^3 \log \frac{4SAHd}{\delta}}{n}} + cH^2\left(\frac{\log \frac{16SAHd}{\delta}}{n}\right)^{3/4} + cH^3\frac{\log \frac{16SAHd}{\delta}}{n},$$

*for some positive constant $c$.*

*Proof.* We upper bound one side, and then the other. For all the $h \in [\![H]\!], (s,y,a) \in \mathcal{S}\times\mathcal{Y}_h\times\mathcal{A}$, it holds that:

$$Q_h^*(s,a,y;\mathfrak{p},\mathfrak{r}) - Q_h^*(s,y,a;\widehat{\mathfrak{p}},\mathfrak{r})$$

$$\overset{(1)}{\leqslant} \mathop{\mathbb{E}}_{\widehat{\mathfrak{p}},\mathfrak{r},\psi*}\left[\sum_{h'=h}^{H}\sum_{s'\in\mathcal{S}}\left(p_{h'}(s'|s_{h'},a_{h'}) - \widehat{p}_{h'}(s'|s_{h'},a_{h'})\right)V_{h'+1}^{\psi*}(s',y_{h'+1};\mathfrak{p},\mathfrak{r})\right.$$

$$\left.\bigg| s_h = s, y_h = y, a_h = a\right]$$

$$\overset{(2)}{\leqslant} \mathop{\mathbb{E}}_{\widehat{\mathfrak{p}},\mathfrak{r},\psi*}\left[\sum_{h'=h}^{H}c\sqrt{\frac{c_1 \mathbb{V}_{s'\sim\widehat{p}_{h'}(\cdot|s_{h'},a_{h'})}[V_{h'+1}^{\psi*}(s',y_{h'+1};\widehat{\mathfrak{p}},\mathfrak{r})]}{n}} + b_2\right.$$

$$\left.\bigg| s_h = s, y_h = y, a_h = a\right]$$

$$
\begin{aligned}
&= c\sqrt{\frac{c_1}{n}}\; \mathop{\mathbb{E}}_{\widehat{\mathfrak{p}},\mathfrak{r},\psi *}\left[\sum_{h'=h}^{H}\sqrt{\mathbb{V}_{s'\sim\widehat{p}_{h'}(\cdot|s_{h'},a_{h'})}[V_{h'+1}^{\psi *}(s',y_{h'+1};\widehat{\mathfrak{p}},\mathfrak{r})]}\;\Big|\; s_h=s, y_h=y, a_h=a\right]\\
&\quad + Hb_2\\
&\overset{(3)}{\leqslant} c\sqrt{\frac{c_1}{n}}\sqrt{H^3} + Hb_2\\
&= c\sqrt{\frac{H^3\log\frac{4SAHY}{\delta}}{n}} + c'H^2\left(\frac{\log\frac{16SAHY}{\delta}}{n}\right)^{3/4} + c''H^3\frac{\log\frac{16SAHY}{\delta}}{n}\\
&=: \epsilon',
\end{aligned}
$$

where at (1) we have applied Lemma E.7, at (2) we have applied Lemma E.10 with $\delta/2$ of probability, at (3) we have applied Lemma E.12.

The proof for the other side of inequality is completely analogous, and it holds w.p. $1-\delta/2$. The result follows through the application of a union bound. $\qquad\square$

**Lemma E.7.** *For any tuple $h\in\llbracket H\rrbracket, (s,y,a)\in\mathcal{S}\times\mathcal{Y}_h\times\mathcal{A}$, it holds that:*

$$
\begin{aligned}
Q_h^*(s,y,a;\mathfrak{p},\mathfrak{r}) - Q_h^*(s,y,a;\widehat{\mathfrak{p}},\mathfrak{r}) &\leqslant \mathop{\mathbb{E}}_{\widehat{\mathfrak{p}},\mathfrak{r},\psi *}\left[\sum_{h'=h}^{H}\sum_{s'\in\mathcal{S}}\right.\\
&\left.\Big(p_{h'}(s'|s_{h'},a_{h'}) - \widehat{p}_{h'}(s'|s_{h'},a_{h'})\Big)V_{h'+1}^{\psi *}(s',y_{h'+1};\mathfrak{p},\mathfrak{r})\;\Big|\; s_h=s, y_h=y, a_h=a\right],\\
Q_h^*(s,y,a;\mathfrak{p},\mathfrak{r}) - Q_h^*(s,y,a;\widehat{\mathfrak{p}},\mathfrak{r}) &\geqslant \mathop{\mathbb{E}}_{\widehat{\mathfrak{p}},\mathfrak{r},\widehat{\psi} *}\left[\sum_{h'=h}^{H}\sum_{s'\in\mathcal{S}}\right.\\
&\left.\Big(p_{h'}(s'|s_{h'},a_{h'}) - \widehat{p}_{h'}(s'|s_{h'},a_{h'})\Big)V_{h'+1}^{\psi *}(s',y_{h'+1};\mathfrak{p},\mathfrak{r})\;\Big|\; s_h=s, y_h=y, a_h=a\right],
\end{aligned}
$$

*where $\psi^*, \widehat{\psi}^*$ are the optimal policies respectively in problems $\mathfrak{p},\mathfrak{r}$ and $\widehat{\mathfrak{p}},\mathfrak{r}$.*

*Proof.* For any $h\in\llbracket H\rrbracket, (s,y,a)\in\mathcal{S}\times\mathcal{Y}_h\times\mathcal{A}$, we can write:

$$
\begin{aligned}
&Q_h^*(s,y,a;\mathfrak{p},\mathfrak{r}) - Q_h^*(s,y,a;\widehat{\mathfrak{p}},\mathfrak{r})\\
&= Q_h^{\psi *}(s,y,a;\mathfrak{p},\mathfrak{r}) - Q_h^{\widehat{\psi} *}(s,y,a;\widehat{\mathfrak{p}},\mathfrak{r})\\
&\overset{(1)}{\leqslant} Q_h^{\psi *}(s,y,a;\mathfrak{p},\mathfrak{r}) - Q_h^{\psi *}(s,y,a;\widehat{\mathfrak{p}},\mathfrak{r})\\
&\overset{(2)}{=} \mathfrak{r}_h(s,y,a) + \sum_{(s',y')\in\mathcal{S}\times\mathcal{Y}_{h+1}}\mathfrak{p}_h(s',y'|s,y,a)V_{h+1}^{\psi *}(s',y';\mathfrak{p},\mathfrak{r})\\
&\quad - \left(\mathfrak{r}_h(s,y,a) + \sum_{(s',y')\in\mathcal{S}\times\mathcal{Y}_{h+1}}\widehat{\mathfrak{p}}_h(s',y'|s,y,a)V_{h+1}^{\psi *}(s',y';\widehat{\mathfrak{p}},\mathfrak{r})\right)\\
&\overset{(3)}{=} \sum_{(s',y')\in\mathcal{S}\times\mathcal{Y}_{h+1}}\mathfrak{p}_h(s',y'|s,y,a)V_{h+1}^{\psi *}(s',y';\mathfrak{p},\mathfrak{r})\\
&\quad - \sum_{(s',y')\in\mathcal{S}\times\mathcal{Y}_{h+1}}\widehat{\mathfrak{p}}_h(s',y'|s,y,a)V_{h+1}^{\psi *}(s',y';\widehat{\mathfrak{p}},\mathfrak{r})\\
&\quad \pm \sum_{(s',y')\in\mathcal{S}\times\mathcal{Y}_{h+1}}\widehat{\mathfrak{p}}_h(s',y'|s,y,a)V_{h+1}^{\psi *}(s',y';\mathfrak{p},\mathfrak{r})\\
&= \sum_{(s',y')\in\mathcal{S}\times\mathcal{Y}_{h+1}}\Big(\mathfrak{p}_h(s',y'|s,y,a) - \widehat{\mathfrak{p}}_h(s',y'|s,y,a)\Big)V_{h+1}^{\psi *}(s',y';\mathfrak{p},\mathfrak{r})\\
&\quad + \sum_{(s',y')\in\mathcal{S}\times\mathcal{Y}_{h+1}}\widehat{\mathfrak{p}}_h(s',y'|s,y,a)\Big(V_{h+1}^{\psi *}(s',y';\mathfrak{p},\mathfrak{r}) - V_{h+1}^{\psi *}(s',y';\widehat{\mathfrak{p}},\mathfrak{r})\Big)
\end{aligned}
$$

$$\stackrel{(4)}{=} \sum_{(s',y')\in\mathcal{S}\times\mathcal{Y}_{h+1}} \Big( p_h(s'|s,a)\mathbb{1}\{y+\overline{r}_h(s,a)=y'\}$$

$$- \widehat{p}_h(s'|s,a)\mathbb{1}\{y+\overline{r}_h(s,a)=y'\}\Big)V_{h+1}^{\psi^*}(s',y';\mathfrak{p},\mathfrak{r})$$

$$+ \sum_{(s',y')\in\mathcal{S}\times\mathcal{Y}_{h+1}} \widehat{\mathfrak{p}}_h(s',y'|s,y,a)\Big(V_{h+1}^{\psi^*}(s',y';\mathfrak{p},\mathfrak{r}) - V_{h+1}^{\psi^*}(s',y';\widehat{\mathfrak{p}},\mathfrak{r})\Big)$$

$$\stackrel{(5)}{=} \sum_{s'\in\mathcal{S}}\Big(p_h(s'|s,a)-\widehat{p}_h(s'|s,a)\Big)\sum_{y'\in\mathcal{Y}_{h+1}}\mathbb{1}\{y+\overline{r}_h(s,a)=y'\}V_{h+1}^{\psi^*}(s',y';\mathfrak{p},\mathfrak{r})$$

$$+ \sum_{(s',y')\in\mathcal{S}\times\mathcal{Y}_{h+1}} \widehat{\mathfrak{p}}_h(s',y'|s,y,a)\Big(V_{h+1}^{\psi^*}(s',y';\mathfrak{p},\mathfrak{r}) - V_{h+1}^{\psi^*}(s';\widehat{\mathfrak{p}},\mathfrak{r})\Big)$$

$$\stackrel{(6)}{=} \sum_{s'\in\mathcal{S}}\Big(p_h(s'|s,a)-\widehat{p}_h(s'|s,a)\Big)V_{h+1}^{\psi^*}(s',y+\overline{r}_h(s,a);\mathfrak{p},\mathfrak{r})$$

$$+ \sum_{(s',y')\in\mathcal{S}\times\mathcal{Y}_{h+1}} \widehat{\mathfrak{p}}_h(s',y'|s,y,a)\Big(V_{h+1}^{\psi^*}(s',y';\mathfrak{p},\mathfrak{r}) - V_{h+1}^{\psi^*}(s',y';\widehat{\mathfrak{p}},\mathfrak{r})\Big)$$

$$= \sum_{s'\in\mathcal{S}}\Big(p_h(s'|s,a)-\widehat{p}_h(s'|s,a)\Big)V_{h+1}^{\psi^*}(s',y+\overline{r}_h(s,a);\mathfrak{p},\mathfrak{r})$$

$$+ \sum_{(s',y')\in\mathcal{S}\times\mathcal{Y}_{h+1}} \widehat{\mathfrak{p}}_h(s',y'|s,y,a)$$

$$\cdot \Big(Q_{h+1}^{\psi^*}(s',y',\psi_{h+1}^*(s',y');\mathfrak{p},\mathfrak{r}) - Q_{h+1}^{\psi^*}(s',y',\psi_{h+1}^*(s',y');\widehat{\mathfrak{p}},\mathfrak{r})\Big),$$

where at (1) we have used that $\widehat{\psi}^*$ is the optimal policy in $\widehat{\mathfrak{p}},\mathfrak{r}$, and thus $Q_h^{\psi^*}(s,a;\widehat{\mathfrak{p}},\mathfrak{r}) \leqslant Q_h^{\widehat{\psi}^*}(s,a;\widehat{\mathfrak{p}},\mathfrak{r})$. At (2) we apply the Bellman equation, at (3) we add and subtract the expected under $\widehat{\mathfrak{p}}$ optimal value function under $\mathfrak{p}$, at (4) we use the definition of transition model $\mathfrak{p},\widehat{\mathfrak{p}}$, at (5) we split the summations, at (6) we recognize that the indicator function takes on value 1 only when $y+\overline{r}_h(s,a)=y'$. Finally, we unfold the recursion to obtain the result.

Concerning the second equation, for any $h\in[\![H]\!], (s,y,a)\in\mathcal{S}\times\mathcal{Y}_h\times\mathcal{A}$, we can write:

$$Q_h^*(s,y,a;\mathfrak{p},\mathfrak{r}) - Q_h^*(s,y,a;\widehat{\mathfrak{p}},\mathfrak{r})$$

$$= Q_h^{\psi^*}(s,y,a;\mathfrak{p},\mathfrak{r}) - Q_h^{\widehat{\psi}^*}(s,y,a;\widehat{\mathfrak{p}},\mathfrak{r})$$

$$\stackrel{(7)}{=} \mathfrak{r}_h(s,y,a) + \sum_{(s',y')\in\mathcal{S}\times\mathcal{Y}_{h+1}} \mathfrak{p}_h(s',y'|s,y,a)V_{h+1}^{\psi^*}(s',y';\mathfrak{p},\mathfrak{r})$$

$$- \Big(\mathfrak{r}_h(s,y,a) + \sum_{(s',y')\in\mathcal{S}\times\mathcal{Y}_{h+1}} \widehat{\mathfrak{p}}_h(s',y'|s,y,a)V_{h+1}^{\widehat{\psi}^*}(s',y';\widehat{\mathfrak{p}},\mathfrak{r})\Big)$$

$$\stackrel{(8)}{=} \sum_{(s',y')\in\mathcal{S}\times\mathcal{Y}_{h+1}} \mathfrak{p}_h(s',y'|s,y,a)V_{h+1}^{\psi^*}(s',y';\mathfrak{p},\mathfrak{r})$$

$$- \sum_{(s',y')\in\mathcal{S}\times\mathcal{Y}_{h+1}} \widehat{\mathfrak{p}}_h(s',y'|s,y,a)V_{h+1}^{\widehat{\psi}^*}(s',y';\widehat{\mathfrak{p}},\mathfrak{r})$$

$$\pm \sum_{(s',y')\in\mathcal{S}\times\mathcal{Y}_{h+1}} \widehat{\mathfrak{p}}_h(s',y'|s,y,a)V_{h+1}^{\psi^*}(s',y';\mathfrak{p},\mathfrak{r})$$

$$= \sum_{(s',y')\in\mathcal{S}\times\mathcal{Y}_{h+1}} \Big(\mathfrak{p}_h(s',y'|s,y,a) - \widehat{\mathfrak{p}}_h(s',y'|s,y,a)\Big)V_{h+1}^{\psi^*}(s',y';\mathfrak{p},\mathfrak{r})$$

$$+ \sum_{(s',y')\in\mathcal{S}\times\mathcal{Y}_{h+1}} \widehat{\mathfrak{p}}_h(s',y'|s,y,a)\Big(V_{h+1}^{\psi^*}(s',y';\mathfrak{p},\mathfrak{r}) - V_{h+1}^{\widehat{\psi}^*}(s',y';\widehat{\mathfrak{p}},\mathfrak{r})\Big)$$

$$= \sum_{(s',y')\in\mathcal{S}\times\mathcal{Y}_{h+1}} \Big(p_h(s'|s,a)\mathbb{1}\{y+\overline{r}_h(s,a)=y'\}$$

$$- \widehat{p}_h(s'|s,a)\mathbb{1}\{y+\overline{r}_h(s,a)=y'\}\Big)V_{h+1}^{\psi^*}(s',y';\mathfrak{p},\mathfrak{r})$$

$$+ \sum_{(s',y')\in\mathcal{S}\times\mathcal{Y}_{h+1}} \widehat{\mathfrak{p}}_h(s',y'|s,y,a)\Big(V_{h+1}^{\psi^*}(s',y';\mathfrak{p},\mathfrak{r}) - V_{h+1}^{\widehat{\psi}^*}(s',y';\widehat{\mathfrak{p}},\mathfrak{r})\Big)$$

$$= \sum_{s'\in\mathcal{S}} \Big(p_h(s'|s,a) - \widehat{p}_h(s'|s,a)\Big) \sum_{y'\in\mathcal{Y}_{h+1}} \mathbb{1}\{y+\overline{r}_h(s,a)=y'\}V_{h+1}^{\psi^*}(s',y';\mathfrak{p},\mathfrak{r})$$

$$+ \sum_{(s',y')\in\mathcal{S}\times\mathcal{Y}_{h+1}} \widehat{\mathfrak{p}}_h(s',y'|s,y,a)\Big(V_{h+1}^{\psi^*}(s',y';\mathfrak{p},\mathfrak{r}) - V_{h+1}^{\widehat{\psi}^*}(s',y';\widehat{\mathfrak{p}},\mathfrak{r})\Big)$$

$$= \sum_{s'\in\mathcal{S}} \Big(p_h(s'|s,a) - \widehat{p}_h(s'|s,a)\Big)V_{h+1}^{\psi^*}(s',y+\overline{r}_h(s,a);\mathfrak{p},\mathfrak{r})$$

$$+ \sum_{(s',y')\in\mathcal{S}\times\mathcal{Y}_{h+1}} \widehat{\mathfrak{p}}_h(s',y'|s,y,a)\Big(V_{h+1}^{\psi^*}(s',y';\mathfrak{p},\mathfrak{r}) - V_{h+1}^{\widehat{\psi}^*}(s',y';\widehat{\mathfrak{p}},\mathfrak{r})\Big)$$

$$\overset{(9)}{\geqslant} \sum_{s'\in\mathcal{S}} \Big(p_h(s'|s,a) - \widehat{p}_h(s'|s,a)\Big)V_{h+1}^{\psi^*}(s',y+\overline{r}_h(s,a);\mathfrak{p},\mathfrak{r})$$

$$+ \sum_{(s',y')\in\mathcal{S}\times\mathcal{Y}_{h+1}} \widehat{\mathfrak{p}}_h(s',y'|s,y,a)$$

$$\cdot \Big(Q_{h+1}^{\psi^*}(s',y',\widehat{\psi}_{h+1}^*(s',y');\mathfrak{p},\mathfrak{r}) - Q_{h+1}^{\widehat{\psi}^*}(s',y',\widehat{\psi}_{h+1}^*(s',y');\widehat{\mathfrak{p}},\mathfrak{r})\Big),$$

where at (7) we have applied the Bellman equation, at (8) we have added and subtracted a term, and at (9) we have used that $V_{h+1}^{\psi^*}(s',y';\mathfrak{p},\mathfrak{r}) = Q_{h+1}^{\psi^*}(s',y',\psi_{h+1}^*(s',y');\mathfrak{p},\mathfrak{r}) \geqslant Q_{h+1}^{\psi^*}(s',y',\widehat{\psi}_{h+1}^*(s',y');\mathfrak{p},\mathfrak{r})$, since $\psi_{h+1}^*(s',y')$ is the optimal action under $\mathfrak{p},\mathfrak{r}$, and so, it cannot be worse than action $\widehat{\psi}_{h+1}^*(s',y')$. By unfolding the recursion, we obtain the result. $\square$

**Lemma E.8.** *For any $\delta \in (0,1)$, w.p. at least $1-\delta$, it holds that:*

$$\max_{h\in[\![H]\!],(s,y)\in\mathcal{S}\times\mathcal{Y}_h} |V_h^*(s,y;\mathfrak{p},\mathfrak{r}) - V_h^{\psi^*}(s,y;\widehat{\mathfrak{p}},\mathfrak{r})| \leqslant cH^2\sqrt{\frac{\log\frac{2SAHd}{\delta}}{n}},$$

$$\max_{h\in[\![H]\!],(s,y)\in\mathcal{S}\times\mathcal{Y}_h} |V_h^*(s,y;\mathfrak{p},\mathfrak{r}) - V_h^*(s,y;\widehat{\mathfrak{p}},\mathfrak{r})| \leqslant cH^2\sqrt{\frac{\log\frac{2SAHd}{\delta}}{n}}.$$

*where $c$ is some positive constant.*

*Proof.* First, we observe that, for any $h \in [\![H]\!], (s,y) \in \mathcal{S}\times\mathcal{Y}_h$, by following passages similar to those in the proof of Lemma E.7:

$$|V_h^*(s,y;\mathfrak{p},\mathfrak{r}) - V_h^{\psi^*}(s,y;\widehat{\mathfrak{p}},\mathfrak{r})|$$

$$= |Q_h^{\psi^*}(s,y,\psi_h^*(s,y);\mathfrak{p},\mathfrak{r}) - Q_h^{\psi^*}(s,y,\psi_h^*(s,y);\widehat{\mathfrak{p}},\mathfrak{r})|$$

$$= \Big|\mathfrak{r}_h(s,y,\psi_h^*(s,y)) + \sum_{(s',y')\in\mathcal{S}\times\mathcal{Y}_{h+1}} \mathfrak{p}_h(s',y'|s,y,\psi_h^*(s,y))V_{h+1}^{\psi^*}(s',y';\mathfrak{p},\mathfrak{r})$$

$$- \Big(\mathfrak{r}_h(s,y,\psi_h^*(s,y)) + \sum_{(s',y')\in\mathcal{S}\times\mathcal{Y}_{h+1}} \widehat{\mathfrak{p}}_h(s',y'|s,y,\psi_h^*(s,y))V_{h+1}^{\psi^*}(s',y';\widehat{\mathfrak{p}},\mathfrak{r})\Big)\Big|$$

$$= \Big| \sum_{(s',y')\in\mathcal{S}\times\mathcal{Y}_{h+1}} \mathfrak{p}_h(s',y'|s,y,\psi_h^*(s,y))V_{h+1}^{\psi^*}(s',y';\mathfrak{p},\mathfrak{r})$$

$$- \sum_{(s',y')\in\mathcal{S}\times\mathcal{Y}_{h+1}} \widehat{\mathfrak{p}}_h(s',y'|s,y,\psi_h^*(s,y))V_{h+1}^{\psi^*}(s',y';\widehat{\mathfrak{p}},\mathfrak{r})$$

$$\pm \sum_{(s',y')\in\mathcal{S}\times\mathcal{Y}_{h+1}} \widehat{\mathfrak{p}}_h(s',y'|s,y,\psi_h^*(s,y))V_{h+1}^{\psi^*}(s',y';\mathfrak{p},\mathfrak{r})\Big|$$

$$= \Big| \sum_{(s',y')\in\mathcal{S}\times\mathcal{Y}_{h+1}} \Big(\mathfrak{p}_h(s',y'|s,y,\psi_h^*(s,y)) - \widehat{\mathfrak{p}}_h(s',y'|s,y,\psi_h^*(s,y))\Big)V_{h+1}^{\psi^*}(s',y';\mathfrak{p},\mathfrak{r})$$

$$+ \sum_{(s',y')\in\mathcal{S}\times\mathcal{Y}_{h+1}} \widehat{\mathfrak{p}}_h(s',y'|s,y,\psi_h^*(s,y))\Big(V_{h+1}^{\psi^*}(s',y';\mathfrak{p},\mathfrak{r}) - V_{h+1}^{\psi^*}(s',y';\widehat{\mathfrak{p}},\mathfrak{r})\Big)\Big|$$

$$= \Big| \sum_{s'\in\mathcal{S}} \Big(p_h(s'|s,\psi_h^*(s,y)) - \widehat{p}_h(s'|s,\psi_h^*(s,y))\Big)V_{h+1}^{\psi^*}(s',y+\overline{r}_h(s,\psi_h^*(s,y));\mathfrak{p},\mathfrak{r})$$

$$+ \sum_{(s',y')\in\mathcal{S}\times\mathcal{Y}_{h+1}} \widehat{\mathfrak{p}}_h(s',y'|s,y,\psi_h^*(s,y))\Big(V_{h+1}^{\psi^*}(s',y';\mathfrak{p},\mathfrak{r}) - V_{h+1}^{\psi^*}(s',y';\widehat{\mathfrak{p}},\mathfrak{r})\Big)\Big|$$

$$= \ldots$$

$$= \Big| \mathop{\mathbb{E}}_{\widehat{\mathfrak{p}},\mathfrak{r},\psi^*}\Big[ \sum_{h'=h}^{H} \sum_{s'\in\mathcal{S}} \Big(p_{h'}(s'|s_{h'},a_{h'}) - \widehat{p}_{h'}(s'|s_{h'},a_{h'})\Big)V_{h'+1}^{\psi^*}(s',y_{h'+1};\mathfrak{p},\mathfrak{r})$$

$$\Big| s_h=s, y_h=y\Big]\Big|$$

$$\overset{(1)}{\leqslant} \mathop{\mathbb{E}}_{\widehat{\mathfrak{p}},\mathfrak{r},\psi^*}\Big[ \sum_{h'=h}^{H} \Big| \sum_{s'\in\mathcal{S}} \Big(p_{h'}(s'|s_{h'},a_{h'}) - \widehat{p}_{h'}(s'|s_{h'},a_{h'})\Big)V_{h'+1}^{\psi^*}(s',y_{h'+1};\mathfrak{p},\mathfrak{r})\Big|$$

$$\Big| s_h=s, y_h=y\Big],$$

where at (1) we have brought the absolute value inside the expectation.

Similarly, for the other term, for any $h\in[\![H]\!], (s,y)\in\mathcal{S}\times\mathcal{Y}_h$, we can write:

$$|V_h^*(s,y;\mathfrak{p},\mathfrak{r}) - V_h^*(s,y;\widehat{\mathfrak{p}},\mathfrak{r})|$$

$$= |V_h^{\psi^*}(s,y;\mathfrak{p},\mathfrak{r}) - V_h^{\widehat{\psi}^*}(s,y;\widehat{\mathfrak{p}},\mathfrak{r})|$$

$$\overset{(2)}{=} |\max_{a\in\mathcal{A}}Q_h^{\psi^*}(s,y,a;\mathfrak{p},\mathfrak{r}) - \max_{a\in\mathcal{A}}Q_h^{\widehat{\psi}^*}(s,y,a;\widehat{\mathfrak{p}},\mathfrak{r})|$$

$$\overset{(3)}{\leqslant} \max_{a\in\mathcal{A}}|Q_h^{\psi^*}(s,y,a;\mathfrak{p},\mathfrak{r}) - Q_h^{\widehat{\psi}^*}(s,y,a;\widehat{\mathfrak{p}},\mathfrak{r})|$$

$$= \max_{a\in\mathcal{A}}\Big|\mathfrak{r}_h(s,y,a) + \sum_{(s',y')\in\mathcal{S}\times\mathcal{Y}_{h+1}} \mathfrak{p}_h(s',y'|s,y,a)V_{h+1}^{\psi^*}(s',y';\mathfrak{p},\mathfrak{r})$$

$$- \Big(\mathfrak{r}_h(s,y,a) + \sum_{(s',y')\in\mathcal{S}\times\mathcal{Y}_{h+1}} \widehat{\mathfrak{p}}_h(s',y'|s,y,a)V_{h+1}^{\widehat{\psi}^*}(s',y';\widehat{\mathfrak{p}},\mathfrak{r})\Big)\Big|$$

$$= \max_{a\in\mathcal{A}}\Big| \sum_{(s',y')\in\mathcal{S}\times\mathcal{Y}_{h+1}} \mathfrak{p}_h(s',y'|s,y,a)V_{h+1}^{\psi^*}(s',y';\mathfrak{p},\mathfrak{r})$$

$$- \sum_{(s',y')\in\mathcal{S}\times\mathcal{Y}_{h+1}} \widehat{\mathfrak{p}}_h(s',y'|s,y,a)V_{h+1}^{\widehat{\psi}^*}(s',y';\widehat{\mathfrak{p}},\mathfrak{r})$$

$$\pm \sum_{(s',y')\in\mathcal{S}\times\mathcal{Y}_{h+1}} \widehat{\mathfrak{p}}_h(s',y'|s,y,a)V_{h+1}^{\psi^*}(s',y';\mathfrak{p},\mathfrak{r})\Big|$$

$$= \max_{a\in\mathcal{A}}\Big| \sum_{(s',y')\in\mathcal{S}\times\mathcal{Y}_{h+1}} \Big(\mathfrak{p}_h(s',y'|s,y,a) - \widehat{\mathfrak{p}}_h(s',y'|s,y,a)\Big)V_{h+1}^{\psi^*}(s',y';\mathfrak{p},\mathfrak{r})$$

$$+ \sum_{(s',y')\in\mathcal{S}\times\mathcal{Y}_{h+1}} \widehat{\mathfrak{p}}_h(s',y'|s,y,a)\Big(V_{h+1}^{\psi^*}(s',y';\mathfrak{p},\mathfrak{r}) - V_{h+1}^{\widehat{\psi}^*}(s',y';\widehat{\mathfrak{p}},\mathfrak{r})\Big)\Big|$$

$$\overset{(4)}{\leqslant} \Big| \sum_{(s',y')\in\mathcal{S}\times\mathcal{Y}_{h+1}} \Big(\mathfrak{p}_h(s',y'|s,y,\overline{a}) - \widehat{\mathfrak{p}}_h(s',y'|s,y,\overline{a})\Big)V_{h+1}^{\psi^*}(s',y';\mathfrak{p},\mathfrak{r})\Big|$$

$$+ \Big| \sum_{(s',y')\in\mathcal{S}\times\mathcal{Y}_{h+1}} \widehat{\mathfrak{p}}_h(s',y'|s,y,\overline{a})\Big(V_{h+1}^{\psi^*}(s',y';\mathfrak{p},\mathfrak{r}) - V_{h+1}^{\widehat{\psi}^*}(s',y';\widehat{\mathfrak{p}},\mathfrak{r})\Big)\Big|$$

$$= \Big| \sum_{s'\in\mathcal{S}} \Big(p_h(s'|s,\overline{a}) - \widehat{p}_h(s'|s,\overline{a})\Big)V_{h+1}^{\psi^*}(s',y+\Pi_{\mathcal{R}}[r_h(s,\overline{a})];\mathfrak{p},\mathfrak{r})\Big|$$

$$+ \Big| \sum_{(s',y') \in \mathcal{S} \times \mathcal{Y}_{h+1}} \widehat{\mathfrak{p}}_h(s',y'|s,y,\overline{a}) \Big( V_{h+1}^{\psi^*}(s',y';\mathfrak{p},\mathfrak{r}) - V_{h+1}^{\widehat{\psi}^*}(s',y';\widehat{\mathfrak{p}},\mathfrak{r}) \Big) \Big|$$

$$\leqslant \ldots$$

$$\overset{(5)}{\leqslant} \underset{\widehat{\mathfrak{p}},\mathfrak{r},\overline{\psi}}{\mathbb{E}} \Bigg[ \sum_{h'=h}^{H} \Big| \sum_{s' \in \mathcal{S}} \Big( p_{h'}(s'|s_{h'},a_{h'}) - \widehat{p}_{h'}(s'|s_{h'},a_{h'}) \Big) V_{h'+1}^{\psi^*}(s',y_{h'+1};\mathfrak{p},\mathfrak{r}) \Big| \Bigg| s_h = s, y_h = y \Bigg],$$

where at (2) we have applied the Bellman optimality equation, at (3) we have upper bounded the difference of maxima with the maximum of the difference, at (4) we denote the maximal action by $\overline{a}$, and we apply triangle inequality; at (5) we have unfolded the recursion and called $\overline{\psi}$ the resulting policy.

Now, for some $\epsilon \in (0,1)$, let us denote by $\mathcal{E}$ the event defined as:

$$\mathcal{E} := \Bigg\{ \forall h \in [\![H]\!], (s,y,a) \in \mathcal{S} \times \mathcal{Y}_h \times \mathcal{A} :$$

$$\Big| \sum_{s' \in \mathcal{S}} \Big( p_h(s'|s,a) - \widehat{p}_h(s'|s,a) \Big) V_{h+1}^{\psi^*}(s',y+\overline{r}_h(s,a);\mathfrak{p},\mathfrak{r}) \Big| \leqslant \epsilon \Bigg\}$$

We can write:

$$\mathbb{P}(\mathcal{E}^{\complement}) = \mathbb{P}\Bigg( \exists h \in [\![H]\!], (s,y,a) \in \mathcal{S} \times \mathcal{Y}_h \times \mathcal{A} :$$

$$\Big| \sum_{s' \in \mathcal{S}} \Big( p_h(s'|s,a) - \widehat{p}_h(s'|s,a) \Big) V_{h+1}^{\psi^*}(s',y+\overline{r}_h(s,a);\mathfrak{p},\mathfrak{r}) \Big| > \epsilon \Bigg)$$

$$\overset{(6)}{\leqslant} \sum_{h \in [\![H]\!],(s,y,a) \in \mathcal{S} \times \mathcal{Y}_h \times \mathcal{A}}$$

$$\mathbb{P}\Bigg( \Big| \sum_{s' \in \mathcal{S}} \Big( p_h(s'|s,a) - \widehat{p}_h(s'|s,a) \Big) V_{h+1}^{\psi^*}(s',y+\overline{r}_h(s,a);\mathfrak{p},\mathfrak{r}) \Big| > \epsilon \Bigg)$$

$$\overset{(7)}{\leqslant} \sum_{h \in [\![H]\!],(s,y,a) \in \mathcal{S} \times \mathcal{Y}_h \times \mathcal{A}} 2e^{\frac{-2n\epsilon^2}{H^2}}$$

$$= 2SAHde^{\frac{-2n\epsilon^2}{H^2}},$$

where at (6) we have applied a union bound over all tuples $h \in [\![H]\!], (s,y,a) \in \mathcal{S} \times \mathcal{Y}_h \times \mathcal{A}$, and at (7) we have applied Hoeffding's inequality, by recalling that we collect $n$ samples (see Algorithm 3) for any $(s,a,h) \in \mathcal{S} \times \mathcal{A} \times [\![H]\!]$ triple, and that vector $V_{h+1}^{\psi^*}(\cdot, y+\overline{r}_h(s,a);\mathfrak{p},\mathfrak{r})$ bounded by $[0,H]$ is independent of the randomness in $\widehat{p}_h(\cdot|s,a)$. It should be remarked that our collection of samples depends only on $\mathcal{S} \times \mathcal{A} \times [\![H]\!]$, and not on $\mathcal{Y}_h$; such term enters the expression only through the union bound, because we have to apply Hoeffding's inequality for all the value functions considered, which are as many as $|\mathcal{Y}_h|$. Note that we use $d = |\mathcal{Y}_{H+1}|$ since it is the largest $|\mathcal{Y}_h|$ among $h \in [\![H+1]\!]$.

This probability is at most $\delta$ if:

$$2SAHde^{\frac{-2n\epsilon^2}{H^2}} \leqslant \delta \iff \epsilon \geqslant H\sqrt{\frac{\log \frac{2SAHd}{\delta}}{2n}}.$$

By plugging into the previous expressions, we obtain that, w.p. $1 - \delta$:

$$|V_h^*(s,y;\mathfrak{p},\mathfrak{r}) - V_h^{\psi^*}(s,y;\widehat{\mathfrak{p}},\mathfrak{r})|$$

$$\leqslant \underset{\widehat{\mathfrak{p}},\mathfrak{r},\psi^*}{\mathbb{E}} \Bigg[ \sum_{h'=h}^{H} \Big| \sum_{s' \in \mathcal{S}} \Big( p_{h'}(s'|s_{h'},a_{h'}) - \widehat{p}_{h'}(s'|s_{h'},a_{h'}) \Big) V_{h'+1}^{\psi^*}(s',y_{h'+1};\mathfrak{p},\mathfrak{r}) \Big|$$

$$\left. \middle| s_h = s, y_h = y\right]$$

$$\leqslant \mathop{\mathbb{E}}_{\widehat{\mathfrak{p}},\mathfrak{r},\psi*}\left[\sum_{h'=h}^{H} H\sqrt{\frac{\log\frac{2SAHd}{\delta}}{2n}}\;\middle|\; s_h = s, y_h = y\right]$$

$$= H^2\sqrt{\frac{\log\frac{2SAHd}{\delta}}{2n}},$$

and also:

$$|V_h^*(s, y; \mathfrak{p}, \mathfrak{r}) - V_h^*(s, y; \widehat{\mathfrak{p}}, \mathfrak{r})|$$

$$\leqslant \mathop{\mathbb{E}}_{\widehat{\mathfrak{p}},\mathfrak{r},\overline{\psi}}\left[\sum_{h'=h}^{H}\left|\sum_{s'\in\mathcal{S}}\Big(p_{h'}(s'|s_{h'}, a_{h'}) - \widehat{p}_{h'}(s'|s_{h'}, a_{h'})\Big)V_{h'+1}^{\psi*}(s', y_{h'+1}; \mathfrak{p}, \mathfrak{r})\right|\right.$$

$$\left. \middle| s_h = s, y_h = y\right]$$

$$\leqslant \mathop{\mathbb{E}}_{\widehat{\mathfrak{p}},\mathfrak{r},\overline{\psi}}\left[\sum_{h'=h}^{H} H\sqrt{\frac{\log\frac{2SAHd}{\delta}}{2n}}\;\middle|\; s_h = s, y_h = y\right]$$

$$= H^2\sqrt{\frac{\log\frac{2SAHd}{\delta}}{2n}}.$$

This concludes the proof.

$\square$

**Lemma E.9.** *For any $\delta \in (0, 1)$, w.p. at least $1 - \delta$, it holds that, for all $h \in [\![H]\!], (s, y, a) \in \mathcal{S} \times \mathcal{Y}_h \times \mathcal{A}$:*

$$\sqrt{\mathbb{V}_{s'\sim p_h(\cdot|s,a)}[V_{h+1}^*(s', y + \overline{r}_h(s, a); \mathfrak{p}, \mathfrak{r})]} \leqslant$$

$$\sqrt{\mathbb{V}_{s'\sim \widehat{p}_h(\cdot|s,a)}[V_{h+1}^{\psi*}(s', y + \overline{r}_h(s, a); \widehat{\mathfrak{p}}, \mathfrak{r})]} + b_1,$$

$$\sqrt{\mathbb{V}_{s'\sim p_h(\cdot|s,a)}[V_{h+1}^*(s', y + \overline{r}_h(s, a); \mathfrak{p}, \mathfrak{r})]} \leqslant$$

$$\sqrt{\mathbb{V}_{s'\sim \widehat{p}_h(\cdot|s,a)}[V_{h+1}^*(s', y + \overline{r}_h(s, a); \widehat{\mathfrak{p}}, \mathfrak{r})]} + b_1,$$

*where $b_1$ is defined as:*

$$b_1 := cH\left(\frac{\log\frac{4SAHY}{\delta}}{n}\right)^{1/4} + c'H^2\sqrt{\frac{\log\frac{4SAHY}{\delta}}{n}},$$

*for some positive constants $c, c'$.*

*Proof.* In the following, we will use $\overline{y}$ as a label for $y + \overline{r}_h(s, a)$. We begin with the first expression. We can write, for any $h \in [\![H]\!], (s, y, a) \in \mathcal{S} \times \mathcal{Y}_h \times \mathcal{A}$:

$$\mathbb{V}_{s'\sim p_h(\cdot|s,a)}[V_{h+1}^*(s', \overline{y}; \mathfrak{p}, \mathfrak{r})]$$

$$= \mathbb{V}_{s'\sim p_h(\cdot|s,a)}[V_{h+1}^*(s', \overline{y}; \mathfrak{p}, \mathfrak{r})] \pm \mathbb{V}_{s'\sim \widehat{p}_h(\cdot|s,a)}[V_{h+1}^*(s', \overline{y}; \mathfrak{p}, \mathfrak{r})]$$

$$= \Big(\mathbb{V}_{s'\sim p_h(\cdot|s,a)}[V_{h+1}^*(s', \overline{y}; \mathfrak{p}, \mathfrak{r})] - \mathbb{V}_{s'\sim \widehat{p}_h(\cdot|s,a)}[V_{h+1}^*(s', \overline{y}; \mathfrak{p}, \mathfrak{r})]\Big)$$

$$+ \mathbb{V}_{s'\sim \widehat{p}_h(\cdot|s,a)}[V_{h+1}^*(s', \overline{y}; \mathfrak{p}, \mathfrak{r})]$$

$$\overset{(1)}{=} \sum_{s'\in\mathcal{S}}\Big(p_h(s'|s, a) - \widehat{p}_h(s'|s, a)\Big)V_{h+1}^{*^2}(s', \overline{y}; \mathfrak{p}, \mathfrak{r})$$

$$- \left[ \left( \sum_{s' \in \mathcal{S}} p_h(s'|s,a) V_{h+1}^*(s', \overline{y}; \mathfrak{p}, \mathfrak{r}) \right)^2 \right.$$

$$\left. - \left( \sum_{s' \in \mathcal{S}} \widehat{p}_h(s'|s,a) V_{h+1}^*(s', \overline{y}; \mathfrak{p}, \mathfrak{r}) \right)^2 \right]$$

$$+ \mathbb{V}_{s' \sim \widehat{p}_h(\cdot|s,a)} [V_{h+1}^*(s', \overline{y}; \mathfrak{p}, \mathfrak{r}) \pm V_{h+1}^{\psi^*}(s', \overline{y}; \widehat{\mathfrak{p}}, \mathfrak{r})]$$

$$\stackrel{(2)}{=} \sum_{s' \in \mathcal{S}} \left( p_h(s'|s,a) - \widehat{p}_h(s'|s,a) \right) V_{h+1}^{*2}(s', \overline{y}; \mathfrak{p}, \mathfrak{r})$$

$$- \left[ \left( \sum_{s' \in \mathcal{S}} p_h(s'|s,a) V_{h+1}^*(s', \overline{y}; \mathfrak{p}, \mathfrak{r}) \right)^2 \right.$$

$$\left. - \left( \sum_{s' \in \mathcal{S}} \widehat{p}_h(s'|s,a) V_{h+1}^*(s', \overline{y}; \mathfrak{p}, \mathfrak{r}) \right)^2 \right]$$

$$+ \mathbb{V}_{s' \sim \widehat{p}_h(\cdot|s,a)} [V_{h+1}^*(s', \overline{y}; \mathfrak{p}, \mathfrak{r}) - V_{h+1}^{\psi^*}(s', \overline{y}; \widehat{\mathfrak{p}}, \mathfrak{r})]$$

$$+ \mathbb{V}_{s' \sim \widehat{p}_h(\cdot|s,a)} [V_{h+1}^{\psi^*}(s', \overline{y}; \widehat{\mathfrak{p}}, \mathfrak{r})]$$

$$+ 2\text{Cov}_{s' \sim \widehat{p}_h(\cdot|s,a)} [V_{h+1}^*(s', \overline{y}; \mathfrak{p}, \mathfrak{r}) - V_{h+1}^{\psi^*}(s', \overline{y}; \widehat{\mathfrak{p}}, \mathfrak{r}),$$

$$V_{h+1}^{\psi^*}(s', \overline{y}; \widehat{\mathfrak{p}}, \mathfrak{r})]$$

$$\stackrel{(3)}{\leqslant} \sum_{s' \in \mathcal{S}} \left( p_h(s'|s,a) - \widehat{p}_h(s'|s,a) \right) V_{h+1}^{*2}(s', \overline{y}; \mathfrak{p}, \mathfrak{r})$$

$$- \left[ \left( \sum_{s' \in \mathcal{S}} p_h(s'|s,a) V_{h+1}^*(s', \overline{y}; \mathfrak{p}, \mathfrak{r}) \right)^2 \right.$$

$$\left. - \left( \sum_{s' \in \mathcal{S}} \widehat{p}_h(s'|s,a) V_{h+1}^*(s', \overline{y}; \mathfrak{p}, \mathfrak{r}) \right)^2 \right]$$

$$+ \mathbb{V}_{s' \sim \widehat{p}_h(\cdot|s,a)} [V_{h+1}^*(s', \overline{y}; \mathfrak{p}, \mathfrak{r}) - V_{h+1}^{\psi^*}(s', \overline{y}; \widehat{\mathfrak{p}}, \mathfrak{r})]$$

$$+ \mathbb{V}_{s' \sim \widehat{p}_h(\cdot|s,a)} [V_{h+1}^{\psi^*}(s', \overline{y}; \widehat{\mathfrak{p}}, \mathfrak{r})]$$

$$+ 2 \Big( \mathbb{V}_{s' \sim \widehat{p}_h(\cdot|s,a)} [V_{h+1}^*(s', \overline{y}; \mathfrak{p}, \mathfrak{r}) - V_{h+1}^{\psi^*}(s', \overline{y}; \widehat{\mathfrak{p}}, \mathfrak{r})]$$

$$\cdot \mathbb{V}_{s' \sim \widehat{p}_h(\cdot|s,a)} [V_{h+1}^{\psi^*}(s', \overline{y}; \widehat{\mathfrak{p}}, \mathfrak{r})] \Big)^{1/2}$$

$$= \sum_{s' \in \mathcal{S}} \left( p_h(s'|s,a) - \widehat{p}_h(s'|s,a) \right) V_{h+1}^{*2}(s', \overline{y}; \mathfrak{p}, \mathfrak{r})$$

$$- \left[ \left( \sum_{s' \in \mathcal{S}} p_h(s'|s,a) V_{h+1}^*(s', \overline{y}; \mathfrak{p}, \mathfrak{r}) \right)^2 \right.$$

$$\left. - \left( \sum_{s' \in \mathcal{S}} \widehat{p}_h(s'|s,a) V_{h+1}^*(s', \overline{y}; \mathfrak{p}, \mathfrak{r}) \right)^2 \right]$$

$$+ \left[ \sqrt{\mathbb{V}_{s' \sim \widehat{p}_h(\cdot|s,a)} [V_{h+1}^*(s', \overline{y}; \mathfrak{p}, \mathfrak{r}) - V_{h+1}^{\psi^*}(s', \overline{y}; \widehat{\mathfrak{p}}, \mathfrak{r})]} \right.$$

$$\left. + \sqrt{\mathbb{V}_{s' \sim \widehat{p}_h(\cdot|s,a)} [V_{h+1}^{\psi^*}(s', \overline{y}; \widehat{\mathfrak{p}}, \mathfrak{r})]} \right]^2,$$

where at (1) we have used the common formula for the variance $\mathbb{V}[X] = \mathbb{E}[X^2] - \mathbb{E}[X]^2$, at (2) we have decomposed the variance of a sum as $\mathbb{V}[X+Y] = \mathbb{V}[X] + \mathbb{V}[Y] + 2\text{Cov}[X,Y]$, at (3) we have applied Cauchy-Schwarz's inequality to bound the covariance with the product of the variances $|\text{Cov}[X,Y]| \leqslant \sqrt{\mathbb{V}[X]\mathbb{V}[Y]}$.

Next, observe that:

$$\mathbb{V}_{s' \sim \widehat{p}_h(\cdot|s,a)} [V_{h+1}^*(s', \overline{y}; \mathfrak{p}, \mathfrak{r}) - V_{h+1}^{\psi^*}(s', \overline{y}; \widehat{\mathfrak{p}}, \mathfrak{r})]$$

$$\stackrel{(4)}{=} \mathbb{E}_{s' \sim \widehat{p}_h(\cdot|s,a)} [(V_{h+1}^*(s', \overline{y}; \mathfrak{p}, \mathfrak{r}) - V_{h+1}^{\psi^*}(s', \overline{y}; \widehat{\mathfrak{p}}, \mathfrak{r}))^2]$$

$$- \mathbb{E}_{s' \sim \widehat{p}_h(\cdot|s,a)} [V_{h+1}^*(s', \overline{y}; \mathfrak{p}, \mathfrak{r}) - V_{h+1}^{\psi^*}(s', \overline{y}; \widehat{\mathfrak{p}}, \mathfrak{r})]^2$$

$$\overset{(5)}{\leqslant} \mathbb{E}_{s' \sim \widehat{p}_h(\cdot|s,a)} [(V_{h+1}^*(s', \overline{y}; \mathfrak{p}, \mathfrak{r}) - V_{h+1}^{\psi^*}(s', \overline{y}; \widehat{\mathfrak{p}}, \mathfrak{r}))^2]$$

$$\overset{(6)}{\leqslant} \|(V_{h+1}^*(\cdot, \overline{y}; \mathfrak{p}, \mathfrak{r}) - V_{h+1}^{\psi^*}(\cdot, \overline{y}; \widehat{\mathfrak{p}}, \mathfrak{r}))^2\|_\infty$$

$$= \|V_{h+1}^*(\cdot, \overline{y}; \mathfrak{p}, \mathfrak{r}) - V_{h+1}^{\psi^*}(\cdot, \overline{y}; \widehat{\mathfrak{p}}, \mathfrak{r})\|_\infty^2,$$

where at (4) we have used $\mathbb{V}[X] = \mathbb{E}[X^2] - \mathbb{E}[X]^2$, at (5) we recognize that the second term is a square, thus always positive, and we remove it, and at (6) we have upper bounded the expected value, an average, through the infinity norm.

Thanks to this expression, we can continue to upper bound the previous term as:

$$\mathbb{V}_{s' \sim p_h(\cdot|s,a)}[V_{h+1}^*(s', \overline{y}; \mathfrak{p}, \mathfrak{r})]$$

$$\leqslant \sum_{s' \in \mathcal{S}} \Big( p_h(s'|s,a) - \widehat{p}_h(s'|s,a) \Big) V_{h+1}^{*\,2}(s', \overline{y}; \mathfrak{p}, \mathfrak{r})$$

$$- \Big[ \Big( \sum_{s' \in \mathcal{S}} p_h(s'|s,a) V_{h+1}^*(s', \overline{y}; \mathfrak{p}, \mathfrak{r}) \Big)^2$$

$$- \Big( \sum_{s' \in \mathcal{S}} \widehat{p}_h(s'|s,a) V_{h+1}^*(s', \overline{y}; \mathfrak{p}, \mathfrak{r}) \Big)^2 \Big]$$

$$+ \Big[ \|V_{h+1}^*(\cdot, \overline{y}; \mathfrak{p}, \mathfrak{r}) - V_{h+1}^{\psi^*}(\cdot, \overline{y}; \widehat{\mathfrak{p}}, \mathfrak{r})\|_\infty$$

$$+ \sqrt{\mathbb{V}_{s' \sim \widehat{p}_h(\cdot|s,a)}[V_{h+1}^{\psi^*}(s', \overline{y}; \widehat{\mathfrak{p}}, \mathfrak{r})]} \Big]^2$$

$$\overset{(7)}{=} \sum_{s' \in \mathcal{S}} \Big( p_h(s'|s,a) - \widehat{p}_h(s'|s,a) \Big) V_{h+1}^{*\,2}(s', \overline{y}; \mathfrak{p}, \mathfrak{r})$$

$$- \Big[ \Big( \sum_{s' \in \mathcal{S}} (p_h(s'|s,a) - \widehat{p}_h(s'|s,a)) V_{h+1}^*(s', \overline{y}; \mathfrak{p}, \mathfrak{r}) \Big)$$

$$\cdot \Big( \sum_{s' \in \mathcal{S}} (p_h(s'|s,a) + \widehat{p}_h(s'|s,a)) V_{h+1}^*(s', \overline{y}; \mathfrak{p}, \mathfrak{r}) \Big) \Big]$$

$$+ \Big[ \|V_{h+1}^*(\cdot, \overline{y}; \mathfrak{p}, \mathfrak{r}) - V_{h+1}^{\psi^*}(\cdot, \overline{y}; \widehat{\mathfrak{p}}, \mathfrak{r})\|_\infty$$

$$+ \sqrt{\mathbb{V}_{s' \sim \widehat{p}_h(\cdot|s,a)}[V_{h+1}^{\psi^*}(s', \overline{y}; \widehat{\mathfrak{p}}, \mathfrak{r})]} \Big]^2$$

$$\overset{(8)}{\leqslant} \sum_{s' \in \mathcal{S}} \Big( p_h(s'|s,a) - \widehat{p}_h(s'|s,a) \Big) V_{h+1}^{*\,2}(s', \overline{y}; \mathfrak{p}, \mathfrak{r})$$

$$- \Big[ \Big( \sum_{s' \in \mathcal{S}} (p_h(s'|s,a) - \widehat{p}_h(s'|s,a)) V_{h+1}^*(s', \overline{y}; \mathfrak{p}, \mathfrak{r}) \Big)$$

$$\cdot \Big( \sum_{s' \in \mathcal{S}} (p_h(s'|s,a) + \widehat{p}_h(s'|s,a)) V_{h+1}^*(s', \overline{y}; \mathfrak{p}, \mathfrak{r}) \Big) \Big]$$

$$+ \Big[ cH^2 \sqrt{\frac{\log \frac{4SAHd}{\delta}}{n}} + \sqrt{\mathbb{V}_{s' \sim \widehat{p}_h(\cdot|s,a)}[V_{h+1}^{\psi^*}(s', \overline{y}; \widehat{\mathfrak{p}}, \mathfrak{r})]} \Big]^2$$

$$\overset{(9)}{\leqslant} \sum_{s' \in \mathcal{S}} \Big( p_h(s'|s,a) - \widehat{p}_h(s'|s,a) \Big) V_{h+1}^{*\,2}(s', \overline{y}; \mathfrak{p}, \mathfrak{r})$$

$$+ 2H \Big| \sum_{s' \in \mathcal{S}} (p_h(s'|s,a) - \widehat{p}_h(s'|s,a)) V_{h+1}^*(s', \overline{y}; \mathfrak{p}, \mathfrak{r}) \Big|$$

$$+ \Big[ cH^2 \sqrt{\frac{\log \frac{4SAHd}{\delta}}{n}} + \sqrt{\mathbb{V}_{s' \sim \widehat{p}_h(\cdot|s,a)}[V_{h+1}^{\psi^*}(s', \overline{y}; \widehat{\mathfrak{p}}, \mathfrak{r})]} \Big]^2$$

$$\overset{(10)}{\leqslant} \sum_{s' \in \mathcal{S}} \Big( p_h(s'|s,a) - \widehat{p}_h(s'|s,a) \Big) V_{h+1}^{*\,2}(s', \overline{y}; \mathfrak{p}, \mathfrak{r})$$

$$+ 2cH^2 \sqrt{\frac{\log \frac{4SAHd}{\delta}}{n}}$$

$$+ \left[ cH^2 \sqrt{\frac{\log \frac{4SAHd}{\delta}}{n}} + \sqrt{\mathbb{V}_{s' \sim \widehat{p}_h(\cdot|s,a)}[V_{h+1}^{\psi*}(s', \overline{y}; \widehat{\mathfrak{p}}, \mathfrak{r})]} \right]^2$$

$$\overset{(11)}{\leqslant} cH^2 \sqrt{\frac{\log \frac{4SAHd}{\delta}}{n}} + 2cH^2 \sqrt{\frac{\log \frac{4SAHd}{\delta}}{n}}$$

$$+ \left[ cH^2 \sqrt{\frac{\log \frac{4SAHd}{\delta}}{n}} + \sqrt{\mathbb{V}_{s' \sim \widehat{p}_h(\cdot|s,a)}[V_{h+1}^{\psi*}(s', \overline{y}; \widehat{\mathfrak{p}}, \mathfrak{r})]} \right]^2$$

$$= 3cH^2 \sqrt{\frac{\log \frac{4SAHd}{\delta}}{n}}$$

$$+ \left[ cH^2 \sqrt{\frac{\log \frac{4SAHd}{\delta}}{n}} + \sqrt{\mathbb{V}_{s' \sim \widehat{p}_h(\cdot|s,a)}[V_{h+1}^{\psi*}(s', \overline{y}; \widehat{\mathfrak{p}}, \mathfrak{r})]} \right]^2,$$

where at (7) we have applied the common formula $x^2 - y^2 = (x - y)(x + y)$, at (8) we have applied Lemma E.8 using probability $\delta' = \delta/2$, and noticing that, for how the discretized MDP is constructed, we have that $\overline{y} \in \mathcal{Y}$, at (9) we have upper bounded the second term with the absolute value and recognized that the value function does not exceed $H$ and the sum of probabilities is no greater than 2; at (10) we recognize that, in the proof of Lemma E.8, we had already bounded that term, thus, under the event $\mathcal{E}$ which holds w.p. $1 - \delta/2$, we have that bound; at (11) we have applied Hoeffding's inequality to all tuples $h \in [\![H]\!], (s, y, a) \in \mathcal{S} \times \mathcal{Y}_h \times \mathcal{A}$ with probability $\delta/(2SAHd)$, and noticed that the square of the value function does not exceed $H^2$.

Observe that the previous formula holds for all $h \in [\![H]\!], (s, y, a) \in \mathcal{S} \times \mathcal{Y}_h \times \mathcal{A}$ w.p. $1 - \delta$ (by summing the two $\delta/2$ through a union bound). By taking the square root of both sides, we obtain:

$$\sqrt{\mathbb{V}_{s' \sim p_h(\cdot|s,a)}[V_{h+1}^*(s', \overline{y}; \mathfrak{p}, \mathfrak{r})]}$$

$$\leqslant \left( 3cH^2 \sqrt{\frac{\log \frac{4SAHd}{\delta}}{n}} + \left[ cH^2 \sqrt{\frac{\log \frac{4SAHd}{\delta}}{n}} \right. \right.$$

$$\left. \left. + \sqrt{\mathbb{V}_{s' \sim \widehat{p}_h(\cdot|s,a)}[V_{h+1}^{\psi*}(s', \overline{y}; \widehat{\mathfrak{p}}, \mathfrak{r})]} \right]^2 \right)^{1/2}$$

$$\overset{(12)}{\leqslant} \underbrace{c'H \sqrt[4]{\frac{\log \frac{4SAHY}{\delta}}{n}} + cH^2 \sqrt{\frac{\log \frac{4SAHY}{\delta}}{n}}}_{=:b_1}$$

$$+ \sqrt{\mathbb{V}_{s' \sim \widehat{p}_h(\cdot|s,a)}[V_{h+1}^{\psi*}(s', \overline{y}; \widehat{\mathfrak{p}}, \mathfrak{r})]}$$

$$= \sqrt{\mathbb{V}_{s' \sim \widehat{p}_h(\cdot|s,a)}[V_{h+1}^{\psi*}(s', \overline{y}; \widehat{\mathfrak{p}}, \mathfrak{r})]} + b_1,$$

where at (12) we have used the fact that $\sqrt{a + b} \leqslant \sqrt{a} + \sqrt{b}$.

To prove the second formula, the passages are basically the same, the only difference is that, at passage (1), we sum and subtract $V_{h+1}^{\widehat{\psi}*}(s', \overline{y}; \widehat{\mathfrak{p}}, \mathfrak{r})$ instead of $V_{h+1}^{\psi*}(s', \overline{y}; \widehat{\mathfrak{p}}, \mathfrak{r})$, and that at passage (8) we apply the other expression in Lemma E.8. This concludes the proof. $\square$

**Lemma E.10.** *For any $\delta \in (0, 1)$, define:*

$$c_1 := \log \frac{2SAHd}{\delta},$$

$$b_2 := cH \left( \frac{\log \frac{8SAHd}{\delta}}{n} \right)^{3/4} + c'H^2 \frac{\log \frac{8SAHd}{\delta}}{n},$$

*for some positive constants $c, c'$. Then, w.p. at least $1 - \delta$, we have, for all $h \in [\![H]\!], (s, y, a) \in \mathcal{S} \times \mathcal{Y}_h \times \mathcal{A}$:*

$$\sum_{s' \in \mathcal{S}} \Big( p_h(s'|s, a) - \widehat{p}_h(s'|s, a) \Big) V_{h+1}^*(s', y + \overline{r}_h(s, a); \mathfrak{p}, \mathfrak{r})$$

$$\leqslant c'' \sqrt{\frac{c_1 \mathbb{V}_{s' \sim \widehat{p}_h(\cdot|s, a)}[V_{h+1}^{\psi^*}(s', y + \overline{r}_h(s, a); \widehat{\mathfrak{p}}, \mathfrak{r})]}{n}} + b_2,$$

$$\sum_{s' \in \mathcal{S}} \Big( p_h(s'|s, a) - \widehat{p}_h(s'|s, a) \Big) V_{h+1}^*(s', y + \overline{r}_h(s, a); \mathfrak{p}, \mathfrak{r})$$

$$\geqslant -c''' \sqrt{\frac{c_1 \mathbb{V}_{s' \sim \widehat{p}_h(\cdot|s, a)}[V_{h+1}^*(s', y + \overline{r}_h(s, a); \widehat{\mathfrak{p}}, \mathfrak{r})]}{n}} + b_2,$$

*for some positive constants $c'', c'''$.*

*Proof.* Again, we will write $\overline{y}$ instead of $y + \overline{r}_h(s, a)$ for simplicity. For all $h \in [\![H]\!], (s, y, a) \in \mathcal{S} \times \mathcal{Y}_h \times \mathcal{A}$, we can write:

$$\sum_{s' \in \mathcal{S}} \Big( p_h(s'|s, a) - \widehat{p}_h(s'|s, a) \Big) V_{h+1}^*(s', \overline{y}; \mathfrak{p}, \mathfrak{r})$$

$$\overset{(1)}{\leqslant} \sqrt{\frac{2 \mathbb{V}_{s' \sim p_h(\cdot|s, a)}[V_{h+1}^*(s', \overline{y}; \mathfrak{p}, \mathfrak{r})] \log \frac{2SAHd}{\delta}}{n}} + \frac{2H \log \frac{2SAHd}{\delta}}{3n}$$

$$\overset{(2)}{\leqslant} \sqrt{\frac{2 \log \frac{2SAHd}{\delta}}{n}} \left( \sqrt{\mathbb{V}_{s' \sim \widehat{p}_h(\cdot|s, a)}[V_{h+1}^{\psi^*}(s', \overline{y}; \widehat{\mathfrak{p}}, R)]} + b_1 \right) + \frac{2H \log \frac{2SAHd}{\delta}}{3n}$$

$$\overset{(3)}{=} c \sqrt{\frac{c_1 \mathbb{V}_{s' \sim \widehat{p}_h(\cdot|s, a)}[V_{h+1}^{\psi^*}(s', \overline{y}; \widehat{\mathfrak{p}}, \mathfrak{r})]}{n}} + c' \sqrt{\frac{c_1}{n}} H \left( \frac{\log \frac{8SAHd}{\delta}}{n} \right)^{1/4}$$

$$+ c'' \sqrt{\frac{c_1}{n}} H^2 \sqrt{\frac{\log \frac{8SAHd}{\delta}}{n}} + c''' H \frac{c_1}{n}$$

$$\leqslant c \sqrt{\frac{c_1 \mathbb{V}_{s' \sim \widehat{p}_h(\cdot|s, a)}[V_{h+1}^{\psi^*}(s', \overline{y}; \widehat{\mathfrak{p}}, \mathfrak{r})]}{n}} + c' H \left( \frac{\log \frac{8SAHd}{\delta}}{n} \right)^{3/4} + c'''' H^2 \frac{\log \frac{8SAHd}{\delta}}{n},$$

where at (1) we have applied the Bernstein's inequality using $\delta/(2SAHd)$ as probability for all $h \in [\![H]\!], (s, y, a) \in \mathcal{S} \times \mathcal{Y}_h \times \mathcal{A}$, and at (2) we have applied Lemma E.9 with $\delta/2$ of probability, and a union bound to guarantee the event to hold w.p. $1 - \delta$, at (3) we use the definition of $c_1 := \log \frac{2SAHd}{\delta}$, and denoted by $c, c', c'', c'''$ some positive constants.

For the other expression, an analogous derivation can be carried out. In particular, we use the other side of the Bernstein's inequality, and the other expression in Lemma E.9. $\qquad \square$

**Lemma E.11.** *For any $h \in [\![H]\!], (s, y, a) \in \mathcal{S} \times \mathcal{Y}_h \times \mathcal{A}$ and deterministic policy $\psi$, let $\Sigma_h^\psi(s, y, a)$ be defined as:*

$$\Sigma_h^\psi(s, y, a) := \mathop{\mathbb{E}}_{\mathfrak{p}, \mathfrak{r}, \psi} \left[ \Big| \sum_{h'=h}^H \mathfrak{r}_{h'}(s_{h'}, y_{h'}, a_{h'}) - Q_h^\psi(s, y, a; \mathfrak{p}, \mathfrak{r}) \Big|^2 \,\Big|\, s_h = s, y_h = y, a_h = a \right].$$

*Then, function $\Sigma$ satisfies the Bellman equation, i.e., for any $h \in [\![H]\!], (s, y, a) \in \mathcal{S} \times \mathcal{Y}_h \times \mathcal{A}$ and deterministic policy $\psi$:*

$$\Sigma_h^\psi(s, y, a) = \mathbb{V}_{s' \sim p_h(\cdot|s, a)}[V_{h+1}^\psi(s', y + \overline{r}_h(s, a); \mathfrak{p}, \mathfrak{r})]$$

$$+ \mathop{\mathbb{E}}_{s' \sim p_h(\cdot|s, a)} [\Sigma_{h+1}^\psi(s', y + \overline{r}_h(s, a), \psi_{h+1}(s', y + \overline{r}_h(s, a)))].$$

*Proof.* For all $h \in [\![H]\!], (s, y, a) \in \mathcal{S} \times \mathcal{Y}_h \times \mathcal{A}$ and deterministic policy $\psi$, we can write (we denote $a' := \psi_{h+1}(s', y + \overline{r}_h(s, a))$ and $\overline{y} := y + \overline{r}_h(s, a)$ for notational simplicity, and we remark that $\overline{y}$ is *not* a random variable):

$$\Sigma_h^\psi(s, y, a) := \mathop{\mathbb{E}}_{\mathfrak{p}, \mathfrak{r}, \psi} \left[ \Big| \sum_{h'=h}^H \mathfrak{r}_{h'}(s_{h'}, y_{h'}, a_{h'}) - Q_h^\psi(s, y, a; \mathfrak{p}, \mathfrak{r}) \Big|^2 \,\Big|\, s_h = s, y_h = y, a_h = a \right]$$

$$\stackrel{(1)}{=} \mathop{\mathbb{E}}_{s' \sim p_h(\cdot|s,a)} \Bigg[ \mathop{\mathbb{E}}_{\mathfrak{p},\mathfrak{r},\psi} \Bigg[ \Bigg| \sum_{h'=h}^{H} \mathfrak{r}_{h'}(s_{h'}, y_{h'}, a_{h'}) - Q_{h+1}^{\psi}(s', \overline{y}, a'; \mathfrak{p}, \mathfrak{r})$$

$$- \big( Q_h^{\psi}(s, y, a; \mathfrak{p}, \mathfrak{r}) - Q_{h+1}^{\psi}(s', \overline{y}, a'; \mathfrak{p}, \mathfrak{r}) \big) \Bigg|^2$$

$$\Big| s_h = s, a_h = a, y_h = y, s_{h+1} = s' \Bigg] \Bigg]$$

$$\stackrel{(2)}{=} \mathop{\mathbb{E}}_{s' \sim p_h(\cdot|s,a)} \Bigg[ \mathop{\mathbb{E}}_{\mathfrak{p},\mathfrak{r},\psi} \Bigg[ \Bigg| \sum_{h'=h+1}^{H} \mathfrak{r}_{h'}(s_{h'}, y_{h'}, a_{h'}) - Q_{h+1}^{\psi}(s', \overline{y}, a'; \mathfrak{p}, \mathfrak{r})$$

$$- \big( Q_h^{\psi}(s, y, a; \mathfrak{p}, \mathfrak{r}) - \mathfrak{r}_h(s, y, a) - Q_{h+1}^{\psi}(s', \overline{y}, a'; \mathfrak{p}, \mathfrak{r}) \big) \Bigg|^2$$

$$\Big| s_{h+1} = s', y_{h+1} = \overline{y} \Bigg] \Bigg]$$

$$\stackrel{(3)}{=} \mathop{\mathbb{E}}_{s' \sim p_h(\cdot|s,a)} \Bigg[ \mathop{\mathbb{E}}_{\mathfrak{p},\mathfrak{r},\psi} \Bigg[ \Bigg| \sum_{h'=h+1}^{H} \mathfrak{r}_{h'}(s_{h'}, y_{h'}, a_{h'}) - Q_{h+1}^{\psi}(s', \overline{y}, a'; \mathfrak{p}, \mathfrak{r}) \Bigg|^2$$

$$\Big| s_{h+1} = s', y_{h+1} = \overline{y} \Bigg] \Bigg]$$

$$- 2 \mathop{\mathbb{E}}_{s' \sim p_h(\cdot|s,a)} \Bigg[ \big( Q_h^{\psi}(s, y, a; \mathfrak{p}, \mathfrak{r}) - \mathfrak{r}_h(s, y, a) - Q_{h+1}^{\psi}(s', \overline{y}, a'; \mathfrak{p}, \mathfrak{r}) \big)$$

$$\cdot \underbrace{\mathop{\mathbb{E}}_{\mathfrak{p},\mathfrak{r},\psi} \Bigg[ \sum_{h'=h+1}^{H} \mathfrak{r}_{h'}(s_{h'}, y_{h'}, a_{h'}) - Q_{h+1}^{\psi}(s', \overline{y}, a'; \mathfrak{p}, \mathfrak{r}) \Big| s_{h+1} = s', y_{h+1} = \overline{y} \Bigg] \Bigg]}_{=0}$$

$$+ \mathop{\mathbb{E}}_{s' \sim p_h(\cdot|s,a)} \Bigg[ \big| Q_h^{\psi}(s, y, a; \mathfrak{p}, \mathfrak{r}) - \mathfrak{r}_h(s, y, a) - Q_{h+1}^{\psi}(s', \overline{y}, a'; \mathfrak{p}, \mathfrak{r}) \big|^2 \Bigg]$$

$$\stackrel{(4)}{=} \mathop{\mathbb{E}}_{s' \sim p_h(\cdot|s,a)} \Bigg[$$

$$\underbrace{\mathop{\mathbb{E}}_{\mathfrak{p},\mathfrak{r},\psi} \Bigg[ \Bigg| \sum_{h'=h+1}^{H} \mathfrak{r}_{h'}(s_{h'}, y_{h'}, a_{h'}) - Q_{h+1}^{\psi}(s', \overline{y}, a'; \mathfrak{p}, \mathfrak{r}) \Bigg|^2 \Big| s_{h+1} = s', y_{h+1} = \overline{y} \Bigg] \Bigg]}_{= \Sigma_{h+1}^{\psi}(s', \overline{y}, a')}$$

$$+ \underbrace{\mathop{\mathbb{E}}_{s' \sim p_h(\cdot|s,a)} \Bigg[ \big| Q_h^{\psi}(s, y, a; \mathfrak{p}, \mathfrak{r}) - \mathfrak{r}_h(s, y, a) - Q_{h+1}^{\psi}(s', \overline{y}, a'; \mathfrak{p}, \mathfrak{r}) \big|^2 \Bigg]}_{=: \mathbb{V}_{s' \sim p_h(\cdot|s,a)}[Q_{h+1}^{\psi}(s', \overline{y}, a'; \mathfrak{p}, \mathfrak{r})] = \mathbb{V}_{s' \sim p_h(\cdot|s,a)}[V_{h+1}^{\psi}(s', \overline{y}; \mathfrak{p}, \mathfrak{r})]}$$

$$= \mathop{\mathbb{E}}_{s' \sim p_h(\cdot|s,a)} [\Sigma_{h+1}^{\psi}(s', \overline{y}, a')] + \mathbb{V}_{s' \sim p_h(\cdot|s,a)}[V_{h+1}^{\psi}(s', \overline{y}; \mathfrak{p}, \mathfrak{r})],$$

at (1) we add and subtract a term, at (2) we bring out the non-random reward received at $h$, at (3) we compute the square and use the linearity of expectation, at (4) we use the fact that $\mathbb{E}_{\mathfrak{p},\mathfrak{r},\psi} \big[ \sum_{h'=h+1}^{H} \mathfrak{r}_{h'}(s_{h'}, y_{h'}, a_{h'}) - Q_{h+1}^{\psi}(s', \overline{y}, a'; \mathfrak{p}, \mathfrak{r}) \big| s_{h+1} = s' \big] = Q_{h+1}^{\psi}(s', \overline{y}, a'; \mathfrak{p}, \mathfrak{r}) - Q_{h+1}^{\psi}(s', \overline{y}, a'; \mathfrak{p}, \mathfrak{r}) = 0$ because of linearity of expectation. $\qquad \square$

**Lemma E.12.** *Let $\psi$ be any policy, and let $\mathfrak{p}$ be any transition model associated to an arbitrary inner dynamics $p$. Then, for all $h \in [\![H]\!], (s, y, a) \in \mathcal{S} \times \mathcal{Y}_h \times \mathcal{A}$, it holds that:*

$$\Bigg| \mathop{\mathbb{E}}_{\mathfrak{p},\mathfrak{r},\psi} \Bigg[ \sum_{h'=h}^{H} \sqrt{\mathbb{V}_{s' \sim p_{h'}(\cdot|s_{h'}, a_{h'})}[V_{h'+1}^{\psi}(s', y_{h'+1}; \mathfrak{p}, \mathfrak{r})]} \Big| s_h = s, y_h = y, a_h = a \Bigg] \Bigg| \leqslant \sqrt{H^3}.$$

*Proof.* For all $h \in [\![H]\!], (s, y, a) \in \mathcal{S} \times \mathcal{Y}_h \times \mathcal{A}$, we can write (note that this derivation is independent of $\mathfrak{p}, p$, so we might

use even $\widehat{\mathfrak{p}}, \widehat{p}$ in the proof):

$$
\Big| \underset{\mathfrak{p},\mathfrak{r},\psi}{\mathbb{E}} \Big[ \sum_{h'=h}^{H} \sqrt{\mathbb{V}_{s'\sim p_{h'}(\cdot|s_{h'},a_{h'})}[V_{h'+1}^{\psi}(s',y_{h'+1};\mathfrak{p},\mathfrak{r})]} \,\big|\, s_h=s, y_h=y, a_h=a \Big] \Big|
$$

$$
\overset{(1)}{\leqslant} \Big| \underset{\mathfrak{p},\mathfrak{r},\psi}{\mathbb{E}} \Big[ \sqrt{H \sum_{h'=h}^{H} \mathbb{V}_{s'\sim p_{h'}(\cdot|s_{h'},a_{h'})}[V_{h'+1}^{\psi}(s',y_{h'+1};\mathfrak{p},\mathfrak{r})]} \,\big|\, s_h=s, y_h=y, a_h=a \Big] \Big|
$$

$$
\overset{(2)}{\leqslant} \sqrt{H} \sqrt{ \underset{\mathfrak{p},\mathfrak{r},\psi}{\mathbb{E}} \Big[ \sum_{h'=h}^{H} \mathbb{V}_{s'\sim p_{h'}(\cdot|s_{h'},a_{h'})}[V_{h'+1}^{\psi}(s',y_{h'+1};\mathfrak{p},\mathfrak{r})] \,\big|\, s_h=s, y_h=y, a_h=a \Big] }
$$

$$
\overset{(3)}{=} \sqrt{H} \Big( \underset{\mathfrak{p},\mathfrak{r},\psi}{\mathbb{E}} \Big[ \sum_{h'=h}^{H} \Sigma_{h'}^{\psi}(s_{h'},y_{h'},a_{h'}) - \mathbb{E}_{s'\sim p_{h'}(\cdot|s_{h'},a_{h'})}\big[ \Sigma_{h'+1}^{\psi}(s',y_{h'+1},\psi_{h'+1}(s',y_{h'+1})) \big]
$$

$$
\big| s_h=s, y_h=y, a_h=a \Big] \Big)^{1/2}
$$

$$
= \sqrt{H} \sqrt{ \underset{\mathfrak{p},\mathfrak{r},\psi}{\mathbb{E}} \Big[ \sum_{h'=h}^{H} \Sigma_{h'}^{\psi}(s_{h'},y_{h'},a_{h'}) - \Sigma_{h'+1}^{\psi}(s_{h'+1},y_{h'+1},a_{h'+1}) \,\big|\, s_h=s, y_h=y, a_h=a \Big] }
$$

$$
\overset{(4)}{=} \sqrt{H} \sqrt{ \underset{\mathfrak{p},\mathfrak{r},\psi}{\mathbb{E}} \Big[ \Sigma_{h}^{\psi}(s_{h},y_{h},a_{h}) - \underbrace{\Sigma_{H+1}^{\psi}(s_{H+1},y_{H+1},a_{H+1})}_{=0} \,\big|\, s_h=s, y_h=y, a_h=a \Big] }
$$

$$
= \sqrt{H} \sqrt{ \Sigma_{h}^{\psi}(s,y,a) }
$$

$$
\overset{(5)}{\leqslant} \sqrt{H} \sqrt{H^2}
$$

$$
= \sqrt{H^3},
$$

where at (1) we have applied the Cauchy-Schwarz's inequality, at (2) we have applied Jensen's inequality, at (3) we have applied Lemma E.11, at (4) we have used telescoping, and at (5) we have bounded $\Sigma_{h}^{\psi}(s,y,a) \leqslant H^2$ for all $h \in \llbracket H \rrbracket, (s,y,a) \in \mathcal{S} \times \mathcal{Y}_h \times \mathcal{A}$. $\qquad\square$

### E.4.3. LEMMAS ON THE OPTIMAL PERFORMANCE FOR MULTIPLE UTILITIES

To prove the following results, we will make use of the notation introduced in the previous section.

**Lemma E.13.** *Let* $\epsilon, \delta \in (0,1)$. *It suffices to execute* **`CATY-UL`** *with:*

$$
\tau \leqslant \widetilde{\mathcal{O}}\Big( \frac{SAH^5}{\epsilon^2}\Big( S + \log \frac{SAH}{\delta} \Big) \Big),
$$

*to obtain* $\sup_{U\in\mathfrak{U}_L} \big| J^*(U;p,r) - \widehat{J}^*(U) \big| \leqslant HL\epsilon_0 + \epsilon$ *w.p.* $1-\delta$.

*Proof.* Similarly to the proof of Lemma E.13, we can write:

$$
\sup_{U\in\mathfrak{U}_L} |J^*(U;p,r) - \widehat{J}^*(U)|
$$

$$
= \sup_{U\in\mathfrak{U}_L} |J^*(U;p,r) - \widehat{J}^*(U) \pm J^*(\mathfrak{p},\mathfrak{r})|
$$

$$
\leqslant \sup_{U\in\mathfrak{U}_L} |J^*(U;p,r) - J^*(\mathfrak{p},\mathfrak{r})| + \sup_{U\in\mathfrak{U}_L} |J^*(\mathfrak{p},\mathfrak{r}) - \widehat{J}^*(U)|
$$

$$
= \sup_{U\in\mathfrak{U}_L} |J^*(U;p,r) - J^*(\mathfrak{p},\mathfrak{r})| + \sup_{U\in\mathfrak{U}_L} |J^*(\mathfrak{p},\mathfrak{r}) - J^*(\widehat{\mathfrak{p}},\mathfrak{r})|
$$

$$
\leqslant HL\epsilon_0 + \sup_{U\in\mathfrak{U}_L} |J^*(\mathfrak{p},\mathfrak{r}) - J^*(\widehat{\mathfrak{p}},\mathfrak{r})|
$$

$$\overset{(1)}{\leqslant} HL\epsilon_0 + H^2\sqrt{\frac{2}{n}\Big(\log\frac{SAH}{\delta} + (S-1)\log\big(e(1+n/(S-1))\big)\Big)}$$

$$\leqslant HL\epsilon_0 + \epsilon,$$

where at (1) we have applied the formula in Lemma E.14.

By enforcing such quantity to be smaller than $\epsilon$, we get:

$$H^2\sqrt{\frac{2}{n}\Big(\log\frac{SAH}{\delta} + (S-1)\log\big(e(1+n/(S-1))\big)\Big)} \leqslant$$

$$\frac{H^2\sqrt{\log\big(e(1+n/(S-1))\big)}}{\sqrt{n}}\sqrt{2\Big(\log\frac{SAH}{\delta} + (S-1)\Big)} \leqslant \epsilon$$

$$\iff n \geqslant 2\frac{H^4}{\epsilon^2}\Big(\log\frac{SAH}{\delta} + (S-1)\Big)\log\big(e(1+n/(S-1))\big).$$

By summing over all $(s,a,h) \in \mathcal{S}\times\mathcal{A}\times[\![H]\!]$, and by applying Lemma J.3 of Lazzati et al. (2024b), we obtain that:

$$\tau = SAHn \geqslant \tilde{\mathcal{O}}\Big(\frac{SAH^5}{\epsilon^2}\Big(\log\frac{SAH}{\delta} + S\Big)\Big).$$

$\square$

**Lemma E.14.** *For any $\delta \in (0,1)$, for all utility functions $U \in \mathfrak{U}_L$ at the same time, we have:*

$$|J_h^*(\mathfrak{p},\mathfrak{r}) - J_h^*(\widehat{\mathfrak{p}},\mathfrak{r})| \leqslant H^2\sqrt{\frac{2}{n}\Big(\log\frac{SAH}{\delta} + (S-1)\log\big(e(1+n/(S-1))\big)\Big)},$$

*w.p. at least $1-\delta$.*

*Proof.* Let us denote by $\mathcal{E}$ the event defined as:

$$\mathcal{E} := \Big\{\forall n \in \mathbb{N},\, \forall h \in [\![H]\!], (s,y,a) \in \mathcal{S}\times\mathcal{Y}_h\times\mathcal{A}:$$

$$n\mathrm{KL}\big(\widehat{p}_h(\cdot|s,a)\|p_h(\cdot|s,a)\big) \leqslant \log\frac{SAH}{\delta} + (S-1)\log\big(e(1+n/(S-1))\big)\Big\}.$$

We can write:

$$\mathbb{P}(\mathcal{E}^\complement) = \mathbb{P}\Big(\exists n \in \mathbb{N},\, \exists h \in [\![H]\!], (s,y,a) \in \mathcal{S}\times\mathcal{Y}_h\times\mathcal{A}:$$

$$n\mathrm{KL}\big(\widehat{p}_h(\cdot|s,a)\|p_h(\cdot|s,a)\big) > \log\frac{SAH}{\delta} + (S-1)\log\big(e(1+n/(S-1))\big)\Big)$$

$$\overset{(1)}{=} \mathbb{P}\Big(\exists n \in \mathbb{N},\, \exists (s,a,h) \in \mathcal{S}\times\mathcal{A}\times[\![H]\!]:$$

$$n\mathrm{KL}\big(\widehat{p}_h(\cdot|s,a)\|p_h(\cdot|s,a)\big) > \log\frac{SAH}{\delta} + (S-1)\log\big(e(1+n/(S-1))\big)\Big)$$

$$\overset{(2)}{\leqslant} \sum_{(s,a,h)\in\mathcal{S}\times\mathcal{A}\times[\![H]\!]} \mathbb{P}\Big(\exists n \in \mathbb{N},\, n\mathrm{KL}\big(\widehat{p}_h(\cdot|s,a)\|p_h(\cdot|s,a)\big) >$$

$$\log\frac{SAH}{\delta} + (S-1)\log\big(e(1+n/(S-1))\big)\Big)$$

$$\overset{(3)}{\leqslant} \sum_{(s,a,h)\in\mathcal{S}\times\mathcal{A}\times[\![H]\!]} \frac{\delta}{SAH}$$

$$\leqslant \delta,$$

where at (1) we realize that there is no dependence on variable $y$, thus we can drop it,[9] at (2) we have applied a union bound over all triples $(s, a, h) \in \mathcal{S} \times \mathcal{A} \times [\![H]\!]$, and at (3) we have applied Proposition 1 of Jonsson et al. (2020).

Next, for all utilities $U \in \mathfrak{U}_L$ at the same time, for all the tuples $h \in [\![H]\!], (s, y) \in \mathcal{S} \times \mathcal{Y}_h$, we can write:

$$
\begin{aligned}
&|V_h^*(s, y; \mathfrak{p}, \mathfrak{r}) - V_h^*(s, y; \widehat{\mathfrak{p}}, \mathfrak{r})| \\
&\overset{(4)}{\leqslant} \underset{\widehat{\mathfrak{p}}, \mathfrak{r}, \overline{\psi}}{\mathbb{E}} \Bigg[ \sum_{h'=h}^{H} \Big| \sum_{s' \in \mathcal{S}} \Big( p_{h'}(s'|s_{h'}, a_{h'}) - \widehat{p}_{h'}(s'|s_{h'}, a_{h'}) \Big) V_{h'+1}^{\psi*}(s', y_{h'+1}; \mathfrak{p}, \mathfrak{r}) \Big| \\
&\qquad\qquad \Big| s_h = s, y_h = y, a_h = a \Bigg] \\
&\overset{(5)}{\leqslant} H \underset{\widehat{\mathfrak{p}}, \mathfrak{r}, \overline{\psi}}{\mathbb{E}} \Bigg[ \sum_{h'=h}^{H} \| p_{h'}(\cdot|s_{h'}, a_{h'}) - \widehat{p}_{h'}(\cdot|s_{h'}, a_{h'}) \|_1 \Big| s_h = s, y_h = y, a_h = a \Bigg] \\
&\overset{(6)}{\leqslant} H \underset{\widehat{\mathfrak{p}}, \mathfrak{r}, \overline{\psi}}{\mathbb{E}} \Bigg[ \sum_{h'=h}^{H} \sqrt{2 \mathrm{KL}(\widehat{p}_{h'}(\cdot|s_{h'}, a_{h'}) \| p_{h'}(\cdot|s_{h'}, a_{h'}))} \Big| s_h = s, y_h = y, a_h = a \Bigg] \\
&\overset{(7)}{\leqslant} H \underset{\widehat{\mathfrak{p}}, \mathfrak{r}, \overline{\psi}}{\mathbb{E}} \Bigg[ \sum_{h'=h}^{H} \sqrt{\frac{2}{n} \Big( \log \frac{SAH}{\delta} + (S-1) \log \big( e(1 + n/(S-1)) \big) \Big)} \\
&\qquad\qquad \Big| s_h = s, y_h = y, a_h = a \Bigg] \\
&= H^2 \sqrt{\frac{2}{n} \Big( \log \frac{SAH}{\delta} + (S-1) \log \big( e(1 + n/(S-1)) \big) \Big)},
\end{aligned}
$$

where at (4) we apply the formula derived in the proof of Lemma E.8 and triangle inequality, at (5) we have upper bounded with the 1-norm, defined as $\|f\|_1 := \sum_x |f(x)|$, at (6) we have applied Pinsker's inequality, at (7) we assume that concentration event $\mathcal{E}$ holds.

We remark that the guarantee provided by this theorem holds not only for $L$-Lipschitz utilities, but for all functions with the same dimensionality (since it is a bound in 1-norm). $\qquad\square$

### E.5. Analysis of `TRACTOR-UL`

**Theorem 5.2.** *Let $L > 0$, $\epsilon, \delta \in (0, 1)$, and $U^E \in \underline{\mathfrak{U}}_L$. Assume that the projection operator $\Pi_{\underline{\mathfrak{U}}_L}$ is implemented exactly. Let the number of samples satisfy Eq. (5). There exist values of $\epsilon_0, K, \alpha, \overline{U}_0$ (see Appendix E.5) such that, if we run* `TRACTOR-UL` *for a number of gradient iterations:*

$$T \geqslant \mathcal{O}\big( N^4 H^4 L^2 / \epsilon^4 \big),$$

*then, w.p. at least $1 - \delta$, any utility $U \in \underline{\mathfrak{U}}_L$ such that $U(y) = \widehat{U}(y) \ \forall y \in \mathcal{Y}$ belongs to $\underline{\mathcal{U}}_\epsilon$.*

*Proof.* The proof draws inspiration from those of Syed & Schapire (2007) and Schlaginhaufen & Kamgarpour (2024).

Given any distribution $\eta$ supported on $\mathcal{Y}$, and given any two utilities $U \in \underline{\mathfrak{U}}_L, \overline{U} \in \overline{\underline{\mathfrak{U}}}_L$ (where $U$ is a function on $[0, H]$ and $\overline{U}$ is a vector on $\mathcal{Y}$), we will abuse notation and write both $U^\intercal \eta$ and $\overline{U}^\intercal \eta$, with obvious meaning.

Moreover, for $L > 0$, we define operator $\mathfrak{C}_L : \overline{\underline{\mathfrak{U}}}_L \to 2^{\underline{\mathfrak{U}}_L}$ (where $2^{\mathcal{X}}$ denotes the power set of set $\mathcal{X}$) that, given vector $\overline{U} \in \overline{\underline{\mathfrak{U}}}_L$, returns the set $\mathfrak{C}_L(\overline{U}) := \{U \in \underline{\mathfrak{U}}_L \mid \forall y \in \mathcal{Y} : U(y) = \overline{U}(y)\}$.

First of all, we observe that the guarantee provided by the theorem follows directly by the following expression:

$$
\underset{\mathcal{M}^1, \mathcal{M}^2, \ldots, \mathcal{M}^N}{\mathbb{P}} \Big( \sup_{U \in \mathfrak{C}_L(\widehat{U})} \sum_{i \in [\![N]\!]} \overline{\mathcal{C}}_{p^i, r^i, \pi^{E,i}}(U) \leqslant \epsilon \Big) \geqslant 1 - \delta,
$$

---

[9]Therefore, differently from the event for a single utility, now there is no dependence on $d$ in the bound. Intuitively, $d$ appeared in the case of a single utility because we had to apply Hoeffding's inequality $d$ times, because we had, potentially, $d$ different value functions (as many as the states). Since now we provide the bound for all the possible value functions (1-norm bound), then the dependence on $d$ disappears.

where $\mathbb{P}_{\mathcal{M}^1, \mathcal{M}^2, \ldots, \mathcal{M}^N}$ denotes the joint probability distribution obtained by the $N$ MDPs $\{\mathcal{M}^i\}_i$.

Let us denote by $\widehat{U} := (\sum_{t=0}^{T-1} \overline{U}_t)/T$ the output of **TRACTOR-UL**. Note that $\widehat{U} \in \overline{\mathfrak{U}}_L$. We can write:

$$\sup_{U \in \mathfrak{C}_L(\widehat{U})} \sum_{i \in [\![N]\!]} \overline{\mathcal{C}}_{p^i, r^i, \pi^{E,i}}(U)$$

$$\overset{(1)}{=} \sup_{U \in \mathfrak{C}_L(\widehat{U})} \sum_{i \in [\![N]\!]} \left( J^*(U; p^i, r^i) - J^{\pi^{E,i}}(U; p^i, r^i) \pm \widehat{U}^{\intercal} \widehat{\eta}^{E,i} \right)$$

$$\overset{(2)}{\leqslant} \sup_{U \in \mathfrak{C}_L(\widehat{U})} \sum_{i \in [\![N]\!]} \left( J^*(U; p^i, r^i) - \widehat{U}^{\intercal} \widehat{\eta}^{E,i} \right) + \epsilon_1$$

$$\overset{(3)}{=} \sup_{U \in \mathfrak{C}_L(\widehat{U})} \sum_{i \in [\![N]\!]} \left( \max_{\eta \in \mathfrak{D}_i} U^{\intercal} \eta - \widehat{U}^{\intercal} \widehat{\eta}^{E,i} \right) + \epsilon_1$$

$$\overset{(4)}{=} \sup_{\substack{U_0 \in \mathfrak{C}_L(\overline{U}_0), \\ \ldots, \\ U_{T-1} \in \mathfrak{C}_L(\overline{U}_{T-1})}} \frac{1}{T} \sum_{i \in [\![N]\!]} \max_{\eta \in \mathfrak{D}_i} \sum_{t=0}^{T-1} \left( U_t^{\intercal} \eta - \overline{U}_t^{\intercal} \widehat{\eta}^{E,i} \right) + \epsilon_1$$

$$\overset{(5)}{\leqslant} \frac{1}{T} \sum_{t=0}^{T-1} \sup_{U_t \in \mathfrak{C}_L(\overline{U}_t)} \sum_{i \in [\![N]\!]} \left( \max_{\eta \in \mathfrak{D}_i} U_t^{\intercal} \eta \pm \overline{U}_t^{\intercal} \widehat{\eta}_t^i - \overline{U}_t^{\intercal} \widehat{\eta}^{E,i} \right) + \epsilon_1$$

$$\overset{(6)}{\leqslant} \frac{1}{T} \sum_{t=0}^{T-1} \sum_{i \in [\![N]\!]} \overline{U}_t^{\intercal} \left( \widehat{\eta}_t^i - \widehat{\eta}^{E,i} \right) \pm \frac{1}{T} \min_{\overline{U} \in \overline{\mathfrak{U}}_L} \sum_{t=0}^{T-1} \sum_{i \in [\![N]\!]} \overline{U}^{\intercal} \left( \widehat{\eta}_t^i - \widehat{\eta}^{E,i} \right) + \epsilon_1 + \epsilon_2$$

$$\overset{(7)}{\leqslant} \frac{1}{T} \min_{\overline{U} \in \overline{\mathfrak{U}}_L} \sum_{t=0}^{T-1} \sum_{i \in [\![N]\!]} \overline{U}^{\intercal} \left( \widehat{\eta}_t^i - \widehat{\eta}^{E,i} \right) + \epsilon_1 + \epsilon_2 + \underbrace{\frac{2HN\sqrt{H/\epsilon_0}}{\sqrt{T}}}_{=:\epsilon_3}$$

$$\overset{(8)}{\leqslant} \frac{1}{T} \sum_{t=0}^{T-1} \sum_{i \in [\![N]\!]} \overline{U}^{E,\intercal} \left( \widehat{\eta}_t^i - \widehat{\eta}^{E,i} \right) \pm U^{E,\intercal} \eta^{p^i, r^i, \pi^{E,i}} + \epsilon_1 + \epsilon_2 + \epsilon_3$$

$$\overset{(9)}{\leqslant} \frac{1}{T} \sum_{t=0}^{T-1} \sum_{i \in [\![N]\!]} \overline{U}^{E,\intercal} \widehat{\eta}_t^i \pm U^{E,\intercal} \eta^{p^i, r^i, \overline{\pi}_t^i} - U^{E,\intercal} \eta^{p^i, r^i, \pi^{E,i}} + 2\epsilon_1 + \epsilon_2 + \epsilon_3$$

$$\overset{(10)}{\leqslant} \frac{1}{T} \sum_{t=0}^{T-1} \sum_{i \in [\![N]\!]} \underbrace{U^{E,\intercal} \left( \eta^{p^i, r^i, \overline{\pi}_t^i} - \eta^{p^i, r^i, \pi^{E,i}} \right)}_{\leqslant 0} + 2\epsilon_1 + \epsilon_2 + \epsilon_3 + \epsilon_4$$

$$\overset{(11)}{\leqslant} 2\epsilon_1 + \epsilon_2 + \epsilon_3 + \epsilon_4,$$

where at (1) we apply the definition of (non)compatibility, at (2) we first upper bound the supremum of a sum with the sum of the supremum, and then we apply Lemma E.15 w.p. $\delta/3$, and denote $\epsilon_1 := NL\sqrt{2H\epsilon_0} + \sum_{i \in [\![N]\!]} cH\sqrt{\frac{H \log \frac{NH_\tau E,i}{\delta \epsilon_0}}{\epsilon_0 \tau^{E,i}}}$, at (3)

we denote by $\mathfrak{D}_i$ the set of possible return distributions in environment $i$, at (4) we use the definition of $\widehat{U}$, and realize that all functions $U \in \mathfrak{C}_L(\widehat{U})$ can be constructed based on $T$ functions $U_0 \in \mathfrak{C}_L(\overline{U}_0), \ldots, U_{T-1} \in \mathfrak{C}_L(\overline{U}_{T-1})$. At (5) we upper bound the maximum of the sum with the sum of maxima, and exchange the two summations, and we add and subtract the dot product between the (discretized) utility $U_t$ and the estimate of the return distribution computed at Line 15; moreover, we bring the sup inside the summation. At (6) we upper bound the supremum of the sum with the sum of the supremum, and we apply Lemma E.16 w.p. $\delta/3$, defining $\epsilon_2 := cNH^2\sqrt{\frac{1}{n}\left( \log \frac{SAHN}{\delta} + (S-1)\log\left(e(1 + n/(S-1))\right)\right)} + NHL\epsilon_0 + c'HN\sqrt{\frac{\log \frac{NT}{\delta}}{K}}$, and we add and subtract a term, at (7) we apply Theorem H.2 from Schlaginhaufen & Kamgarpour (2024) since set $\overline{\mathfrak{U}}_L$ is closed and convex, where $D := \max_{\overline{U}, \overline{U}' \in \overline{\mathfrak{U}}_L} \|\overline{U} - \overline{U}'\|_2 = \sqrt{d - 2H} = \sqrt{\lceil H/\epsilon_0 \rceil - 1}H \leqslant H\sqrt{H/\epsilon_0}$

(recall that we consider increasing and not strictly-increasing utilities),[10] and $\max_{\overline{U} \in \underline{\mathfrak{U}}_L} \|\nabla \sum_{i \in [\![N]\!]} \overline{U}^\intercal (\widehat{\eta}_t^i - \widehat{\eta}^{E,i})\|_2 = \|\sum_{i \in [\![N]\!]} \widehat{\eta}_t^i - \widehat{\eta}^{E,i}\|_2 \leqslant \sum_{i \in [\![N]\!]} \|\widehat{\eta}_t^i\|_1 + \|\widehat{\eta}^{E,i}\|_1 = 2N =: G$ (because $\widehat{\eta}_t^i$ and $\widehat{\eta}^{E,i}$ are probability distributions), with learning rate $\alpha = D/(G\sqrt{T}) = H\sqrt{d-2}/(2N\sqrt{T}) = \sqrt{\lceil H/\epsilon_0 \rceil - 1} H/(2N\sqrt{T})$, at (8) we upper bound the minimum over utilities with a specific choice of utility, $\overline{U}^E$, and we add and subtract a term; note that $\overline{U}^E \in \underline{\mathfrak{U}}_L$ corresponds to the expert's utility $U^E \in \underline{\mathfrak{U}}_L$ (by hypothesis), i.e., for all $y \in \mathcal{Y}$ : $\overline{U}^E(y) = U^E(y)$. Note that, by hypothesis, $U^E$ makes all the expert policies optimal, i.e., $\forall i \in [\![N]\!] : U^{E,\intercal} \eta^{p^i, r^i, \pi^{E,i}} = \sup_\pi U^{E,\intercal} \eta^{p^i, r^i, \pi}$. At (9) we note that, under the good event of Lemma E.15, we can provide an upper bound using the term in Lemma E.15 (since $U^E \in \underline{\mathfrak{U}}_L$); in addition, we sum and subtract a term that depends on some policy $\overline{\pi}_t^i$, whose existence is guaranteed by Lemma E.17, which we apply at the next step. At (10) we apply Lemma E.17 w.p. $\delta/3$, and we define as $\epsilon_4$ the upper bound times $N$. Finally, at (11) we use the hypothesis that utility $U^E$ makes the expert policy optimal in all environments.

We want that $2\epsilon_1 + \epsilon_2 + \epsilon_3 + \epsilon_4 \leqslant \epsilon$. We can rewrite the sum as:

$$2\epsilon_1 + \epsilon_2 + \epsilon_3 + \epsilon_4$$
$$= \left(2NL\sqrt{2H\epsilon_0} + \frac{3}{2}LNH\epsilon_0\right) + c\frac{HN\sqrt{H}}{\sqrt{\epsilon_0 T}}$$
$$+ c' \sum_{i \in [\![N]\!]} H\sqrt{\frac{H \log \frac{NH\tau^{E,i}}{\delta \epsilon_0}}{\epsilon_0 \tau^{E,i}}} + c'' NH\sqrt{\frac{\log \frac{NT}{\delta}}{K}}$$
$$+ c''' NH^2\sqrt{\frac{1}{n}\left(\log \frac{SAHN}{\delta} + (S-1)\log\left(e(1 + n/(S-1))\right)\right)}.$$

By imposing each term smaller than $\epsilon/5$, we find that it suffices that

$$\begin{cases} \epsilon_0 = \frac{\epsilon^2}{80N^2L^2H} \\ T \geqslant \mathcal{O}\left(\frac{N^2 H^3}{\epsilon_0 \epsilon^2}\right) \geqslant \mathcal{O}\left(\frac{N^4 H^4 L^2}{\epsilon^4}\right) \\ \tau^{E,i} \geqslant \widetilde{\mathcal{O}}\left(\frac{H^3 N^2 \log \frac{NH}{\delta \epsilon_0}}{\epsilon_0 \epsilon^2}\right) \geqslant \widetilde{\mathcal{O}}\left(\frac{H^4 N^4 L^2 \log \frac{NHL}{\delta \epsilon}}{\epsilon^4}\right) \quad \forall i \in [\![N]\!] \\ K \geqslant \widetilde{\mathcal{O}}\left(\frac{N^2 H^2 \log \frac{NT}{\delta}}{\epsilon^2}\right) \geqslant \widetilde{\mathcal{O}}\left(\frac{N^2 H^2 \log \frac{NHL}{\delta \epsilon}}{\epsilon^2}\right) \\ \tau^i \geqslant \widetilde{\mathcal{O}}\left(\frac{N^2 SAH^5}{\epsilon^2}\left(S + \log \frac{SAHN}{\delta}\right)\right) \quad \forall i \in [\![N]\!] \end{cases},$$

where we have used that $\tau^i = SAHn$ for all $i \in [\![N]\!]$, and also used Lemma J.3 of Lazzati et al. (2024b).

The statement of the theorem follows through the application of a union bound. $\qquad \square$

**Lemma E.15.** *Let $\delta \in (0,1)$. Then, it holds that, w.p. at least $1 - \delta$:*

$$\sup_{U \in \underline{\mathfrak{U}}_L} \sum_{i \in [\![N]\!]} \left| U^\intercal \widehat{\eta}^{E,i} - J^{\pi^{E,i}}(U; p^i, r^i) \right| \leqslant NL\sqrt{2H\epsilon_0} + \sum_{i \in [\![N]\!]} cH\sqrt{\frac{H \log \frac{NH\tau^{E,i}}{\delta \epsilon_0}}{\epsilon_0 \tau^{E,i}}},$$

*where $c$ is some positive constant.*

*Proof.* We can make the same derivation as in the proof of Theorem 5.1 to upper bound the objective with the sum of two terms, which can then be bounded using Lemma E.1 and the expression (Eq. (17)) obtained in the proof of Lemma E.4 w.p. $\delta/N$:

$$\sup_{U \in \underline{\mathfrak{U}}_L} \sum_{i \in [\![N]\!]} \left| U^\intercal \widehat{\eta}^{E,i} - J^{\pi^{E,i}}(U; p^i, r^i) \right|$$
$$\leqslant L \sum_{i \in [\![N]\!]} w_1(\eta^{p^i, r^i, \pi^{E,i}}, \mathrm{Proj}_\mathcal{C}(\eta^{p^i, r^i, \pi^{E,i}}))$$

---

[10]The maximum is attained by discretized utilities $\overline{U}, \overline{U}'$ that assign, respectively, $\overline{U}(y) = 0$ and $\overline{U}'(y) = H$ to all the $y \in \mathcal{Y} \setminus \{y_1, y_d\}$.

$$+ \sum_{i\in[\![N]\!]} \sup_{\overline{U}'\in[0,H]^d} \Big| \mathbb{E}_{G\sim\text{Proj}_{\mathcal{C}}(\eta^{p^i,r^i,\pi^{E,i}})}[\overline{U}'(G)] - \mathbb{E}_{G\sim\widehat{\eta}^{E,i}}[\overline{U}'(G)] \Big|$$

$$\leqslant LN\sqrt{2H\epsilon_0} + \sum_{i\in[\![N]\!]} cH\sqrt{\frac{H\log\frac{NH\tau^{E,i}}{\delta\epsilon_0}}{\epsilon_0\tau^{E,i}}}.$$

The result follows through the application of the union bound. $\qquad\square$

**Lemma E.16.** *Let $\delta \in (0,1)$. With probability at least $1-\delta$, for all $t \in \{0,1,\dots,T-1\}$, for all $i \in [\![N]\!]$, it holds that:*

$$\sup_{U_t\in\mathfrak{C}_L(\overline{U}_t)} \max_{\eta\in\mathfrak{D}_i} U_t^{\intercal}\eta - \overline{U}_t^{\intercal}\widehat{\eta}_t^i \leqslant cH^2\sqrt{\frac{1}{n}\Big(\log\frac{SAHN}{\delta} + (S-1)\log\big(e(1+n/(S-1))\big)\Big)}$$

$$+ HL\epsilon_0 + c'H\sqrt{\frac{\log\frac{NT}{\delta}}{K}},$$

*where $c,c'$ are some positive constants.*

*Proof.* We use the notation in Section 5. In particular, let policy $\widehat{\pi}_t^{*,i}$ be the optimal policy in the RS-MDP $\widehat{\mathcal{M}}_{\overline{U}_t}^i :=$ $(\mathcal{S}^i, \mathcal{A}^i, H, s_0^i, \widehat{p}^i, \overline{r}^i, \overline{U}_t)$, i.e.:

$$J^{\widehat{\pi}_t^{*,i}}(\overline{U}_t;\widehat{p}^i,\overline{r}^i) = J^*(\overline{U}_t;\widehat{p}^i,\overline{r}^i) = J^*(U_t;\widehat{p}^i,\overline{r}^i),$$

where the last passage holds trivially for all $U_t \in \mathfrak{C}_L(\overline{U}_t)$ (because there is no evaluation of utility outside $\mathcal{Y}$).

Thus, for all $t \in \{0,1,\dots,T-1\}$, we have:

$$\sup_{U_t\in\mathfrak{C}_L(\overline{U}_t)} \max_{\eta\in\mathfrak{D}_i} U_t^{\intercal}\eta - \overline{U}_t^{\intercal}\widehat{\eta}_t^i \pm J^*(U_t;\widehat{p}^i,\overline{r}^i)$$

$$\overset{(1)}{\leqslant} \sup_{U_t\in\mathfrak{C}_L(\overline{U}_t)} \Big| J^*(U_t;p^i,r^i) - J^*(U_t;\widehat{p}^i,\overline{r}^i) \Big| + \Big| \overline{U}_t^{\intercal}\Big(\widehat{\eta}_t^i - \eta^{\widehat{p}^i,\overline{r}^i,\widehat{\pi}_t^{*,i}}\Big) \Big|$$

$$\overset{(2)}{\leqslant} HL\epsilon_0 + cH^2\sqrt{\frac{1}{n}\Big(\log\frac{SAHN}{\delta} + (S-1)\log\big(e(1+n/(S-1))\big)\Big)}$$

$$+ \Big| \overline{U}_t^{\intercal}\Big(\widehat{\eta}_t^i - \eta^{\widehat{p}^i,\overline{r}^i,\widehat{\pi}_t^{*,i}}\Big) \Big|$$

$$\overset{(3)}{\leqslant} HL\epsilon_0 + cH^2\sqrt{\frac{1}{n}\Big(\log\frac{SAHN}{\delta} + (S-1)\log\big(e(1+n/(S-1))\big)\Big)}$$

$$+ c'H\sqrt{\frac{\log\frac{NT}{\delta}}{K}},$$

where at (1) we have applied the triangle inequality, and realized that in the second term there is no dependence on the value of utility outside of $\mathcal{Y}$; moreover, we have used that $J^*(U_t;\widehat{p}^i,\overline{r}^i) = \overline{U}_t^{\intercal}\eta^{\widehat{p}^i,\overline{r}^i,\widehat{\pi}_t^{*,i}}$ by definition of policy $\widehat{\pi}_t^{*,i}$. At (2) we apply Lemma E.13 (our $J^*(U_t;\widehat{p}^i,\overline{r}^i)$ has the same meaning of $\widehat{J}^*(U)$ in the lemma, and we upper bound $\sup_{U_t\in\mathfrak{C}_L(\overline{U}_t)}$ with $\sup_{U\in\underline{\mathfrak{U}}_L}$) w.p. $\delta/(2N)$,[11] and we keep the confidence bound explicit, and we upper bound $d \leqslant H/\epsilon_0 + 1$, and at (3) we observe that $\widehat{\eta}_t^i$ is the empirical estimate of distribution $\eta^{\widehat{p}^i,\overline{r}^i,\widehat{\pi}_t^{*,i}}$ (see Line 15) obtained through the sampling of $K$ sample returns $G_1, G_2, \dots, G_K \overset{\text{i.i.d.}}{\sim} \eta^{\widehat{p}^i,\overline{r}^i,\widehat{\pi}_t^{*,i}}$. Indeed, note that the policy $\widehat{\psi}_t^{*,i}$, computed at Line 13 and optimal for $\mathfrak{E}[\widehat{\mathcal{M}}_{\overline{U}_t}^i] = (\{\mathcal{S}^i\times\mathcal{Y}_h\}_h, \mathcal{A}^i, H, s_0^i, \widehat{\mathfrak{p}}^i, \mathfrak{r}_t^i)$,[12] provides policy $\widehat{\pi}_t^{*,i}$ through the formula in Section 2, thus Line 14 is actually simulating $\widehat{\pi}_t^{*,i}$ in MDP $\widehat{\mathcal{M}}^i$. Therefore, we can apply Hoeffding's inequality (e.g., see Lemma E.3) w.p. $\delta/(2TN)$.

---

[11]We remark that, in doing so, we can still apply Proposition 3 of Wu & Xu (2023) inside the proof of Lemma E.13 even though we consider *increasing* utilities instead of *strictly-increasing* utilities; indeed, it is trivial to observe that the proof of Proposition 3 of Wu & Xu (2023) does not depend on such property.

[12]See Section 2 for the meaning of $\widehat{\mathfrak{p}}^i$ and $\mathfrak{r}_t^i$; we use $\mathcal{Y}_h$ for all $h$ in the state space instead of the sets of partial returns $\{\mathcal{G}_h^{\widehat{p}^i,\overline{r}^i}\}_h$ in order to obtain policy $\widehat{\psi}_t^{*,i}$ supported on the entire $\mathcal{S}\times\mathcal{Y}_h$ space, and to make it compliant with Algorithm 4

The result follows through the application of the union bound.

We remark that in one case we use probability $\delta/(2N)$ (without $T$) while in the other we use $\delta/(2NT)$ (with $T$), because in the former we provide a guarantee for all possible utilities w.r.t. the optimal performance, thus all the $T$ steps are already included; instead, in the latter, we provide a guarantee for a single utility and for a single policy at a specific $t \in \{0, \ldots, T-1\}$, thus we have to compute a union bound with $T$. $\qquad\square$

**Lemma E.17.** *Let $\delta \in (0,1)$. With probability at least $1 - \delta$, for all $i \in [\![N]\!]$ and $t \in \{0, \ldots, T-1\}$, under the good event in Lemma E.16, there exists a policy $\overline{\pi}_t^i$ such that:*

$$\overline{U}^{E,\intercal}\widehat{\eta}_t^i - U^{E,\intercal}\eta^{p^i,r^i,\overline{\pi}^i} \leqslant LH\epsilon_0/2 + cH\sqrt{\frac{\log\frac{NT}{\delta}}{K}}$$
$$+ c'H^2\sqrt{\frac{1}{n}\Big(\log\frac{SAHN}{\delta} + (S-1)\log\big(e(1 + n/(S-1))\big)\Big)},$$

*where $c, c'$ are positive constants.*

*Proof.* First, simply observe that $\widehat{\eta}_t^i$ is the empirical estimate (see Line 15) of $\eta^{\widehat{p}^i,\overline{r}^i,\widehat{\pi}_t^{*,i}}$, thus, similarly to the proof of Lemma E.16, for all $i \in [\![N]\!]$ and $t \in \{0, 1, \ldots, T-1\}$, we can apply Hoeffding's inequality w.p. $\delta/(2TN)$:

$$\left|\overline{U}^{E,\intercal}\Big(\widehat{\eta}_t^i - \eta^{\widehat{p}^i,\overline{r}^i,\widehat{\pi}_t^{*,i}}\Big)\right| \leqslant cH\sqrt{\frac{\log\frac{NT}{\delta}}{K}}.$$

Now, we compare distributions $\eta^{\widehat{p}^i,\overline{r}^i,\widehat{\pi}_t^{*,i}}$ and $\eta^{p^i,\overline{r}^i,\widehat{\pi}_t^{*,i}}$. Through straightforward passages, we can write:

$$|U^{E,\intercal}\Big(\eta^{\widehat{p}^i,\overline{r}^i,\widehat{\pi}_t^{*,i}} - \eta^{p^i,\overline{r}^i,\widehat{\pi}_t^{*,i}}\Big)|$$
$$= |J^{\widehat{\pi}_t^{*,i}}(\overline{U}^E; \widehat{p}^i, \overline{r}^i) - J^{\widehat{\pi}_t^{*,i}}(\overline{U}^E; p^i, \overline{r}^i)|$$
$$= \Big|\sum_{s'\in\mathcal{S}} p_1^i(s'|s_0^i, \widehat{\pi}_{t,1}^{*,i}(s_0^i))V_2^{\widehat{\pi}_t^{*,i}}(s'; p^i, \overline{r}^i)$$
$$- \sum_{s'\in\mathcal{S}} \widehat{p}_1^i(s'|s_0^i, \widehat{\pi}_{t,1}^{*,i}(s_0^i))V_2^{\widehat{\pi}_t^{*,i}}(s'; \widehat{p}^i, \overline{r}^i)\Big|$$
$$\leqslant \Big|\sum_{s'\in\mathcal{S}} \Big(p_1^i(s'|s_0^i, \widehat{\pi}_{t,1}^{*,i}(s_0^i)) - \widehat{p}_1^i(s'|s_0^i, \widehat{\pi}_{t,1}^{*,i}(s_0^i))\Big)V_2^{\widehat{\pi}_t^{*,i}}(s'; p^i, \overline{r}^i)\Big|$$
$$+ \sum_{s'\in\mathcal{S}} \widehat{p}_1^i(s'|s_0^i, \widehat{\pi}_{t,1}^{*,i}(s_0, 0))\Big|V_2^{\widehat{\pi}_t^{*,i}}(s'; p^i, \overline{r}^i) - V_2^{\widehat{\pi}_t^{*,i}}(s'; \widehat{p}^i, \overline{r}^i)\Big|$$
$$\leqslant \ldots$$
$$\leqslant \underset{\widehat{p}^i,\overline{r}^i,\widehat{\pi}_t^{*,i}}{\mathbb{E}}\Big[\sum_{h'=1}^{H}\Big|\sum_{s'\in\mathcal{S}}\Big(p_{h'}^i(s'|s_{h'}, a_{h'}) - \widehat{p}_{h'}^i(s'|s_{h'}, a_{h'})\Big)V_{h'+1}^{\widehat{\pi}_t^{*,i}}(s'; p^i, \overline{r}^i)\Big|$$
$$\Big| s_1 = s_0^i\Big]$$
$$\leqslant H\underset{\widehat{p}^i,\overline{r}^i,\widehat{\pi}_t^{*,i}}{\mathbb{E}}\Big[\sum_{h'=1}^{H}\Big\|p_{h'}^i(\cdot|s_{h'}, a_{h'}) - \widehat{p}_{h'}^i(\cdot|s_{h'}, a_{h'})\Big\|_1\Big| s_1 = s_0^i\Big]$$
$$\leqslant H\underset{\widehat{p}^i,\overline{r}^i,\widehat{\pi}_t^{*,i}}{\mathbb{E}}\Big[\sum_{h'=1}^{H}\sqrt{2\mathrm{KL}(p_{h'}^i(\cdot|s_{h'}, a_{h'})\|\widehat{p}_{h'}^i(\cdot|s_{h'}, a_{h'}))}\Big| s_1 = s_0^i\Big],$$

where at the last passage we applied the Pinsker's inequality. Note that the previous derivation was possible as long as as policy $\widehat{\pi}_t^{*,i}$ is defined over all the possible pairs state-cumulative reward $(s, y) \in \mathcal{S} \times \mathcal{Y}_h$ for all $h \in [\![H]\!]$. Since we construct it through policy $\widehat{\psi}_t^{*,i}$, obtained at Line 13, i.e., over the entire enlarged state space $\{\mathcal{S} \times \mathcal{Y}_h\}_h$, then policy $\widehat{\pi}_t^{*,i}$ satsifies

such property. Now, in the proof of Lemma E.16 we used Lemma E.14, in which event $\mathcal{E}$ bounds the KL-divergence between transition models. Therefore, under the application of Lemma E.16, it holds that:

$$|U^{E,\intercal}\big(\eta^{\widehat{p}^i,\overline{r}^i,\widehat{\pi}_t^{*,i}} - \eta^{p^i,\overline{r}^i,\widehat{\pi}_t^{*,i}}\big)| \leqslant H^2\sqrt{\frac{2}{n}\Big(\log\frac{SAHN}{\delta} + (S-1)\log\big(e(1+n/(S-1))\big)\Big)},$$

where $n$ is the number of samples takes at each $(s,a,h) \in \mathcal{S} \times \mathcal{A} \times \llbracket H \rrbracket$ in the $i \in \llbracket N \rrbracket$ MDP.

Therefore, we can finally write:

$$\overline{U}^{E,\intercal}\widehat{\eta}_t^i - U^{E,\intercal}\eta^{p^i,r^i,\overline{\pi}^i} \pm \overline{U}^{E,\intercal}\eta^{\widehat{p}^i,\overline{r}^i,\widehat{\pi}_t^{*,i}} \pm \overline{U}^{E,\intercal}\eta^{p^i,\overline{r}^i,\widehat{\pi}_t^{*,i}}$$

$$= U^{E,\intercal}\big(\eta^{p^i,\overline{r}^i,\widehat{\pi}_t^{*,i}} - \eta^{p^i,r^i,\overline{\pi}^i}\big) + \overline{U}^{E,\intercal}\big(\eta^{\widehat{p}^i,\overline{r}^i,\widehat{\pi}_t^{*,i}} - \eta^{p^i,\overline{r}^i,\widehat{\pi}_t^{*,i}}\big)$$

$$+ \overline{U}^{E,\intercal}\big(\widehat{\eta}_t^i - \eta^{\widehat{p}^i,\overline{r}^i,\widehat{\pi}_t^{*,i}}\big)$$

$$\overset{(1)}{\leqslant} U^{E,\intercal}\big(\eta^{p^i,\overline{r}^i,\widehat{\pi}_t^{*,i}} - \eta^{p^i,r^i,\overline{\pi}^i}\big) + cH\sqrt{\frac{\log\frac{NT}{\delta}}{K}}$$

$$+ c'H^2\sqrt{\frac{2}{n}\Big(\log\frac{SAHN}{\delta} + (S-1)\log\big(e(1+n/(S-1))\big)\Big)}$$

$$\overset{(2)}{\leqslant} LH\epsilon_0/2 + cH\sqrt{\frac{\log\frac{NT}{\delta}}{K}}$$

$$+ c'H^2\sqrt{\frac{2}{n}\Big(\log\frac{SAHN}{\delta} + (S-1)\log\big(e(1+n/(S-1))\big)\Big)},$$

where at (1) we have used the bounds derived earlier, and at (2) we have applied Lemma E.18, noticing that we can choose policy $\overline{\pi}^i$ as we wish, and using that $k \leqslant \epsilon_0/2$.

$\square$

**Lemma E.18.** *Let $\mathcal{M}_1 = (\mathcal{S}, \mathcal{A}, H, s_0, p, r^1)$ and $\mathcal{M}_2 = (\mathcal{S}, \mathcal{A}, H, s_0, p, r^2)$ be two MDPs with deterministic rewards that differ only in the reward function $r^1 \neq r^2$, and assume that, for all $(s,a,h) \in \mathcal{S} \times \mathcal{A} \times \llbracket H \rrbracket$, it holds that $|r_h^1(s,a) - r_h^2(s,a)| \leqslant k$, for some $k \geqslant 0$. Let $\pi^1$ be an arbitrary (potentially non-Markovian) policy that induces, in $\mathcal{M}_1$, the distribution over returns $\eta^{p,r^1,\pi^1}$. Then, there exists a policy $\pi^2$ that induces in $\mathcal{M}_2$ the distribution $\eta^{p,r^2,\pi^2}$ such that:*

$$\sup_{U \in \underline{\mathfrak{U}}_L}\Big|\mathbb{E}_{G \sim \eta^{p,r^1,\pi^1}}[U(G)] - \mathbb{E}_{G \sim \eta^{p,r^2,\pi^2}}[U(G)]\Big| \leqslant LHk.$$

*Proof.* A non-Markovian policy like $\pi^1$, in its most general form, prescribes actions at stages $h \in \llbracket H \rrbracket$ depending on the sequence of state-action-reward $(s_1, a_1, r_1, s_2, a_2, r_2, \ldots, s_{h-1}, a_{h-1}, r_{h-1}, s_h)$ received so far. Since, by hypothesis, the reward functions are deterministic (see also Section 2), then it is clear that the information contained in the rewards received so far ($\{r_1, r_2, \ldots, r_{h-1}\}$) is already contained in the state-action pairs received $(s_1, a_1, s_2, a_2, \ldots, s_{h-1}, a_{h-1}, s_h)$ (indeed, for deterministic reward $r^1$, we have that $r_1 = r_1^1(s_1, a_1), r_2 = r_2^1(s_2, a_2)$, and so on). This means that, for any non-Markovian policy in the MDP $\mathcal{M}_1$, since it coincides with $\mathcal{M}_2$ except for the deterministic reward function, it is possible to construct a policy $\pi^2$ that induces the same distribution over *state-action* trajectories, i.e., for any state-action trajectory $\omega = (s_1, a_1, s_2, a_2, \ldots, s_{H-1}, a_{H-1}, s_H, a_H, s_{H+1}) \in \Omega$, it holds $\mathbb{P}_{p,r^1,\pi^1}(\omega) = \mathbb{P}_{p,r^2,\pi^2}(\omega)$.

Therefore, we can write:

$$\sup_{U \in \underline{\mathfrak{U}}_L}\Big|\mathbb{E}_{G \sim \eta^{p,r^1,\pi^1}}[U(G)] - \mathbb{E}_{G \sim \eta^{p,r^2,\pi^2}}[U(G)]\Big|$$

$$\overset{(1)}{=} \sup_{U \in \underline{\mathfrak{U}}_L}\Big|\sum_{\omega \in \Omega}\mathbb{P}_{p,r^1,\pi^1}(\omega)U\Big(\sum_{(s,a,h) \in \omega}r_h^1(s,a)\Big)$$

$$- \sum_{\omega \in \Omega}\mathbb{P}_{p,r^2,\pi^2}(\omega)U\Big(\sum_{(s,a,h) \in \omega}r_h^2(s,a)\Big)\Big|$$

$$\overset{(2)}{=} \sup_{U \in \underline{\mathfrak{U}}_L} \Big| \sum_{\omega \in \Omega} \mathbb{P}_{p,r^1,\pi^1}(\omega) U \Big( \sum_{(s,a,h) \in \omega} r_h^1(s,a) \Big)$$

$$- \sum_{\omega \in \Omega} \mathbb{P}_{p,r^1,\pi^1}(\omega) U \Big( \sum_{(s,a,h) \in \omega} r_h^2(s,a) \Big) \Big|$$

$$= \sup_{U \in \underline{\mathfrak{U}}_L} \Big| \sum_{\omega \in \Omega} \mathbb{P}_{p,r^1,\pi^1}(\omega) \Big( U \Big( \sum_{(s,a,h) \in \omega} r_h^1(s,a) \Big) - U \Big( \sum_{(s,a,h) \in \omega} r_h^2(s,a) \Big) \Big) \Big|$$

$$\overset{(3)}{\leqslant} \sup_{U \in \underline{\mathfrak{U}}_L} \sum_{\omega \in \Omega} \mathbb{P}_{p,r^1,\pi^1}(\omega) \Big| U \Big( \sum_{(s,a,h) \in \omega} r_h^1(s,a) \Big) - U \Big( \sum_{(s,a,h) \in \omega} r_h^2(s,a) \Big) \Big|$$

$$\overset{(4)}{\leqslant} \sum_{\omega \in \Omega} \mathbb{P}_{p,r^1,\pi^1}(\omega) L \Big| \sum_{(s,a,h) \in \omega} (r_h^1(s,a) - r_h^2(s,a)) \Big|$$

$$\overset{(5)}{\leqslant} \sum_{\omega \in \Omega} \mathbb{P}_{p,r^1,\pi^1}(\omega) L \sum_{(s,a,h) \in \omega} \Big| r_h^1(s,a) - r_h^2(s,a) \Big|$$

$$\overset{(6)}{\leqslant} \sum_{\omega \in \Omega} \mathbb{P}_{p,r^1,\pi^1}(\omega) L \sum_{(s,a,h) \in \omega} k$$

$$= LHk,$$

where at (1) we use the fact that the expected utility w.r.t. the distribution over returns can be computed using the probability distribution over state-action trajectories (since the rewards are deterministic), at (2) we use that policy $\pi^2$ is constructed exactly to match the distribution over state-action trajectories, at (3) we apply triangle inequality, at (4) we use the fact that all utilities $U \in \underline{\mathfrak{U}}_L$ are $L$-Lipschitz, i.e., for all $x, y \in [0, H]$: $|U(x) - U(y)| \leqslant L|x - y|$, at (5) we apply again the triangle inequality, and at (6) we use the hypothesis that $r^1, r^2$ are close to each other by parameter $k$. $\qquad \square$

# F. Experimental Details

In this appendix, we collect additional information about the experiments described in Section 6. Appendix F.1 presents formally the MDP used for the collection of the data along with the questions posed to the participants. Appendix F.2 contains additional details on Experiment 2. Finally, Appendix F.3 presents an additional experiment conducted on the collected data.

## F.1. Data Description

Below, we describe the data collected.

### F.1.1. CONSIDERED MDP

The 15 participants analyzed in the study have been provided with complete access to the MDP in Figure 11, which we will denote by $\mathcal{M}$. In other words, the participants *know the transition model and the reward function of $\mathcal{M}$ everywhere*.

Intuitively, states L (Low), M (Medium), H (High), and T (Top), represent 4 "levels" so that the received reward increases when playing actions in "higher" states instead of "lower" states. Formally, MDP $\mathcal{M} = (\mathcal{S}, \mathcal{A}, H, s_0, p, r)$ has four states $\mathcal{S} = \{L, M, H, T\}$, and three actions for each state $\mathcal{A} = \{a_0, a_+, a_-\}$. The horizon is $H = 5$, i.e., the agent has to take 5 actions. The initial state is $s_0 = M$. The transition model $p$ is stationary, i.e., it does not depend on the stage $h \in [\![H]\!]$. Specifically, $p$ is depicted in Table 1. The intuition is that action $a_0$ keeps the agent in the same state deterministically, while action $a_+$ tries to bring the agent to the higher state with probability $1/3$, and action $a_-$ sometimes make the agent "fall down" to the lower state with probability $1/5$.

The reward function $r : \mathcal{S} \times \mathcal{A} \times [\![H]\!] \to \mathbb{R}$ is deterministic, stationary, and depends only the state-action pair played. The specific values are depicted in Table 2. Note that we have written the reward values as numbers in $[0€, 1000€]$, to provide a monetary interpretation. Nevertheless, we will rescale the interval to $[0, 1]$ during the analysis for normalization. Observe that the same actions played in "higher" states (e.g., $H$ or $T$) provide higher rewards than when played in "lower" states (e.g., $L$ or $M$). Moreover, notice that action $a_+$, which is the only action that tries to increase the state, does not provide reward at all, while the risky action $a_-$, which sometimes decreases the state, always provides double the reward

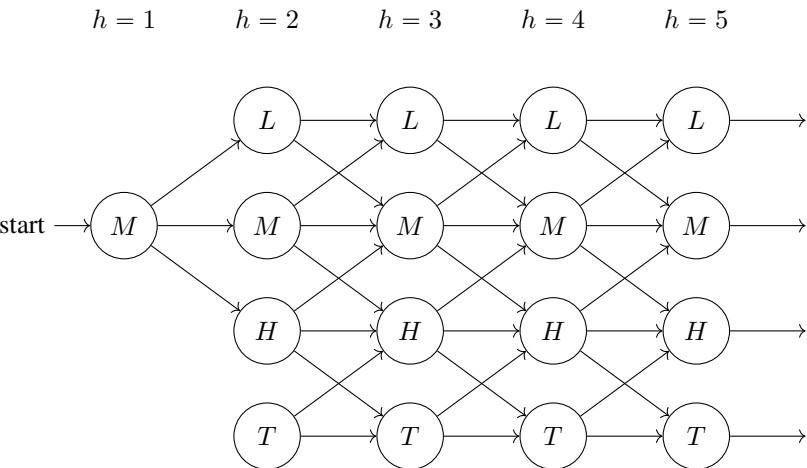

Figure 11. The MDP used for data collection.

| $p$ | $L$ | $M$ | $H$ | $T$ |
|---|---|---|---|---|
| $(L, a_0)$ | 1 | 0 | 0 | 0 |
| $(L, a_+)$ | 2/3 | 1/3 | 0 | 0 |
| $(L, a_-)$ | 1 | 0 | 0 | 0 |
| $(M, a_0)$ | 0 | 1 | 0 | 0 |
| $(M, a_+)$ | 0 | 2/3 | 1/3 | 0 |
| $(M, a_-)$ | 1/5 | 4/5 | 0 | 0 |
| $(H, a_0)$ | 0 | 0 | 1 | 0 |
| $(H, a_+)$ | 0 | 0 | 2/3 | 1/3 |
| $(H, a_-)$ | 0 | 1/5 | 4/5 | 0 |
| $(T, a_0)$ | 0 | 0 | 0 | 1 |
| $(T, a_+)$ | 0 | 0 | 0 | 1 |
| $(T, a_-)$ | 0 | 0 | 1/5 | 4/5 |

Table 1. The transition model $p$ of MDP $\mathcal{M}$.

than "default" action $a_0$.

| | $L$ | $M$ | $H$ | $T$ |
|---|---|---|---|---|
| $a_0$ | 0€ | 30€ | 100€ | 500€ |
| $a_+$ | 0€ | 0€ | 0€ | 0€ |
| $a_-$ | 0€ | 60€ | 200€ | 1000€ |

Table 2. The reward function $r$ of MDP $\mathcal{M}$.

### F.1.2. INTUITION BEHIND AGENTS BEHAVIOR

The reward is interpreted as money. Playing MDP $\mathcal{M}$ involves a trade-off between playing action $a_+$, which gives no money but potentially allows to collect more money in the future (by reaching "higher" states), and action $a_-$, which provides the greatest amount of money immediately, but potentially reduces the amount of money which can be earned in the future. Action $a_0$, being deterministic, provides a reference point, so that deterministically playing action $a_0$ for all the $H = 5$ stages gives to the agent $30 \times 5 = 150$€. Thus, playing actions $a_+, a_-$ other than $a_0$ means that the agent accepts some risk to try to increase its earnings.

### F.1.3. QUESTIONS ASKED TO THE PARTICIPANTS

We remark that the participants have enough background knowledge to understand the MDP described. To each participant, we ask which action in $\{a_0, a_+, a_-\}$ it would play if it was in a certain state $s$, stage $h$, with cumulative reward up to now $y$, for many different values of triples $(s, h, y) \in \mathcal{S} \times [\![H]\!] \times [0€, 5000€]$. Specifically, the values of triples $s, h, y$ considered are:

$$
\begin{array}{ccccc}
(M, 1, 0€) & (M, 2, 0€) & (M, 2, 30€) & (M, 2, 60€) & (H, 2, 0€) \\
(M, 3, 0€) & (M, 3, 30€) & (M, 3, 60€) & (M, 3, 200€) & (H, 3, 0€) \\
(H, 3, 30€) & (H, 3, 60€) & (H, 3, 200€) & (T, 3, 0€) & (M, 4, 0€) \\
(M, 4, 30€) & (M, 4, 60€) & (M, 4, 90€) & (M, 4, 120€) & (M, 4, 150€) \\
(M, 4, 180€) & (M, 4, 300€) & (M, 4, 400€) & (H, 4, 0€) & (H, 4, 30€) \\
(H, 4, 60€) & (H, 4, 100€) & (H, 4, 130€) & (H, 4, 200€) & (H, 4, 300€) \\
(H, 4, 1000€) & (T, 4, 0€) & (T, 4, 60€). & &
\end{array}
$$

From state $L$, we assume all participants always play action $a_+$ since it is the only rational strategy. Moreover, from stage $h = 5$, we assume that all participants always play action $a_-$ since, again, it is the only rational strategy.

In all other possible combinations of values of $s, h, y$, we "interpolate" by considering the action recommended by the participant in the closest $y'$ to $y$, in the same $s, h$.

### F.1.4. THE RETURN DISTRIBUTION OF THE PARTICIPANTS' POLICIES

We now present the return distribution of the policies prescribed by the participants. Specifically, we have simulated 10000 times the policies of the participants, and we have computed the empirical estimate of their return distributions. Such values are reported in Figures 12, 13, 14, 15, and 16, where we use notation $\eta_i^E$ to denote the return distribution of participant $i$, with $i \in [\![15]\!]$.

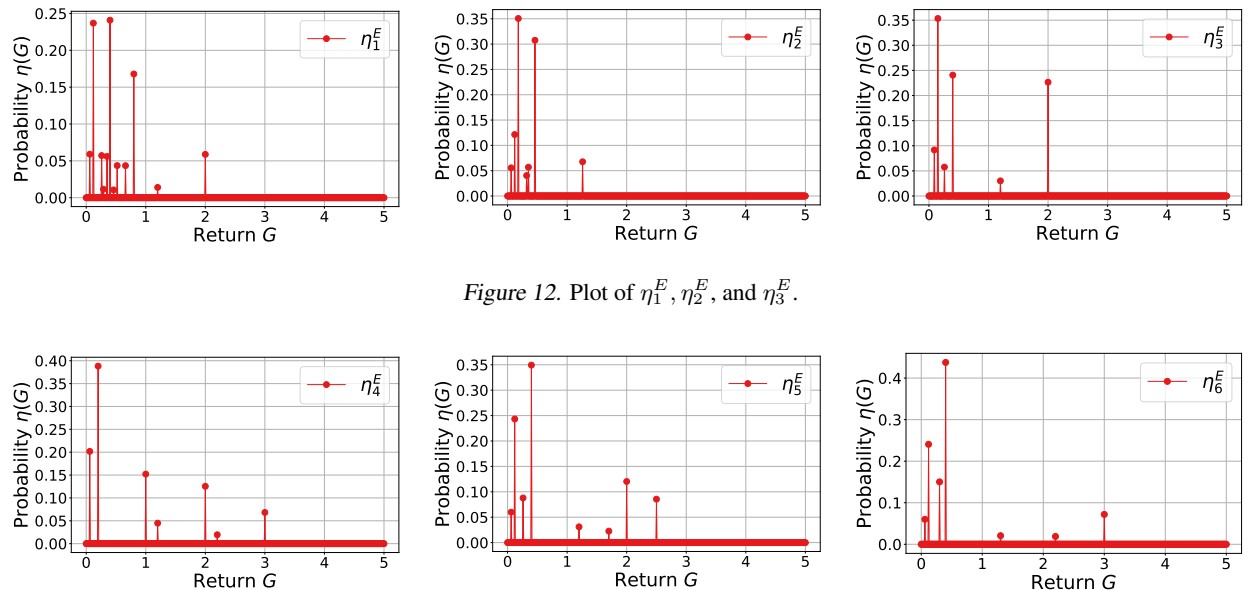

*Figure 12.* Plot of $\eta_1^E$, $\eta_2^E$, and $\eta_3^E$.

*Figure 13.* Plot of $\eta_4^E$, $\eta_5^E$, and $\eta_6^E$.

## F.2. Details Experiment 2

Experiment 2 is made of two parts, the first in which we execute **TRACTOR-UL** on the MDP (and data) adopted also in Experiment 1, and the other where we use simulated data. We describe here the former, while we present the latter more in detail in Appendix F.2.3.

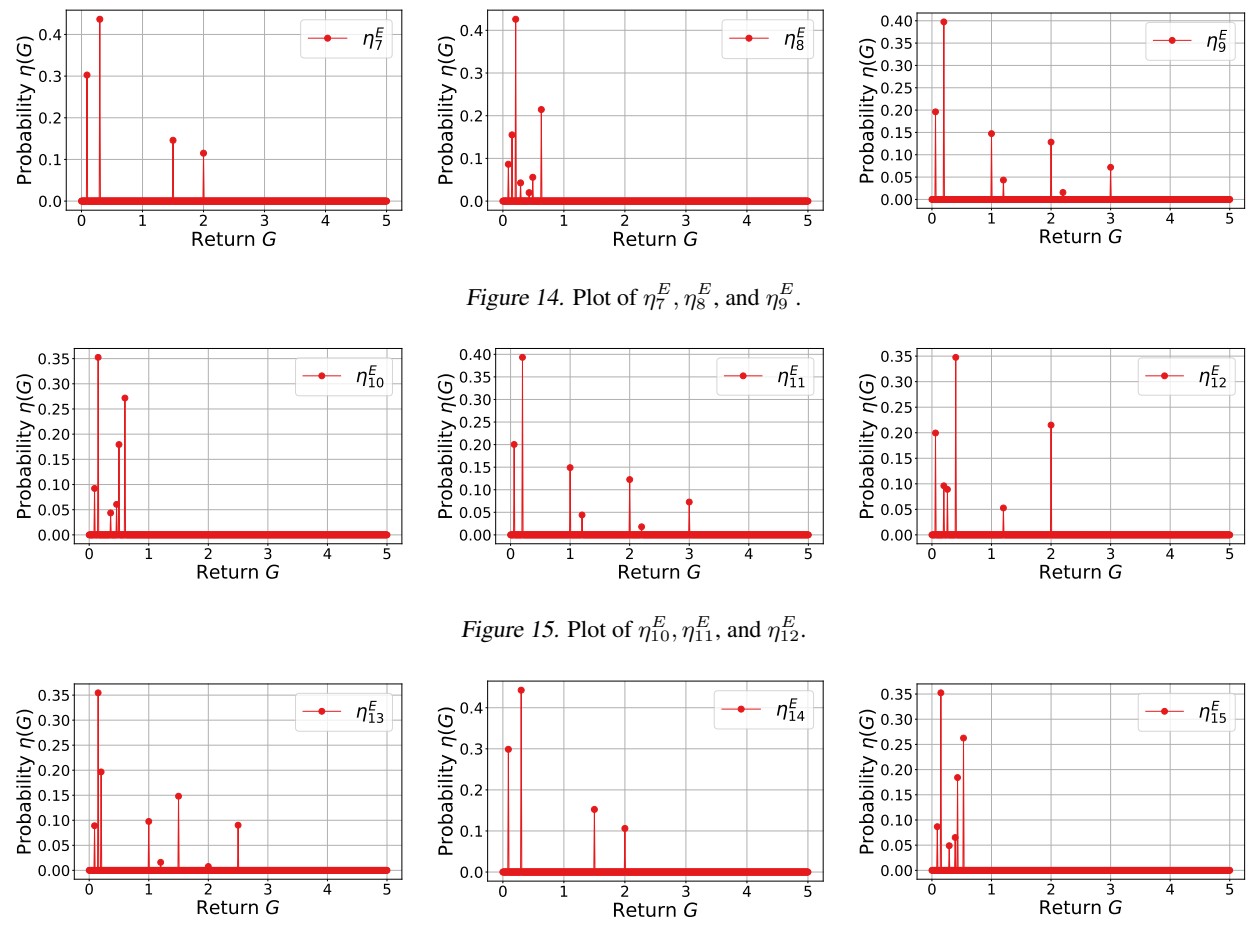

Figure 14. Plot of $\eta_7^E$, $\eta_8^E$, and $\eta_9^E$.

Figure 15. Plot of $\eta_{10}^E$, $\eta_{11}^E$, and $\eta_{12}^E$.

Figure 16. Plot of $\eta_{13}^E$, $\eta_{14}^E$, and $\eta_{15}^E$.

We consider the policy of the 10th participant (chosen arbitrarily) to the survey, and we execute **TRACTOR-UL** multiple times with varying values of the input parameters, specifically: we always use $K = 10000$ trajectories for estimating the return distribution of the 10th participant's policy, and the return distribution of the optimal policies computed along the way; we make 5 runs with each combination of parameters with different seeds. We execute for $T = 70$ iterations using Lipschitz constant $L = 10$, which means that we consider only utilities $U \in \overline{\mathfrak{U}}_L$ satisfying $|U(G) - U(G')| \leqslant 10|G - G'|$ for all $G, G' \in [0, 5]$ (the horizon is 5). As initial utility $\overline{U}_0$, we try $U_{\text{sqrt}}$, $U_{\text{square}}$, and $U_{\text{linear}}$ (see Appendix F.3), and as learning rates we try $0.01, 0.5, 5, 100, 1000, 10000$.

The experiment has been conducted in some hours on a personal computer with processor AMD Ryzen 5 5500U with Radeon Graphics (2.10 GHz), with 8,00 GB of RAM.

We note that the choice of $\overline{U}_0$ is rather irrelevant for the shape of the extracted $\widehat{U}$, but it matters for its "location", as shown in Fig. 17.

To view the sequence of utilities extracted by **TRACTOR-UL** during the run, see Appendix F.2.1, while in Appendix F.2.2 we explain better why the best learning rate is large.

### F.2.1. THE SEQUENCE OF UTILITIES EXTRACTED BY **TRACTOR-UL** ON THE COLLECTED DATA

We now present some plots representing the sequence of utilities extracted by **TRACTOR-UL** during its execution. Specifically, we consider initial utility $\overline{U}_0 = U_{\text{square}}$, and we use learning rates $\alpha \in [0.01, 0.5, 5, 100, 1000, 10000]$. We plot the sequence of utilities considered by **TRACTOR-UL** during its execution in Figures 18, 19, and 20, where we adopt notation that $U_t$ denotes the utility extracted at iteration $t$, and the number in the legend represents the (non)compatibility of that utility. We consider again participant 10.

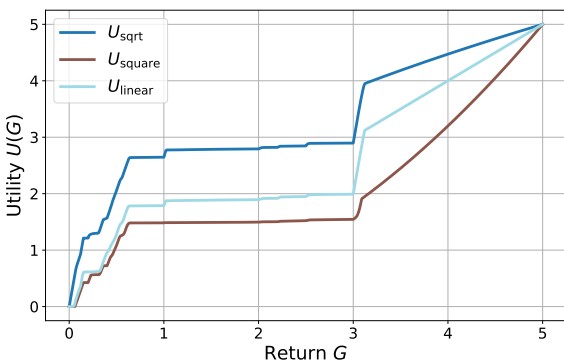

*Figure 17.* Utilities computed by `TRACTOR-UL` starting with the $\overline{U}_0$ in the legend ($\alpha = 100$).

We observe that, for smaller learning rates (e.g., $\alpha \in [0.01, 0.5, 5]$), the utilities as well as the (non)compatibilities) do not change much (Figure 18 and Figure 19 left), while for larger learning rates, we obtain more consistent changes (Figure 19 left and Figure 20).

Clearly, larger learning rates require less iterations to achieve small values of (non)compatibilities. Nevertheless, too large values (e.g., $\alpha = 10000$) are outperformed by intermediate values (e.g., $\alpha = 100$).

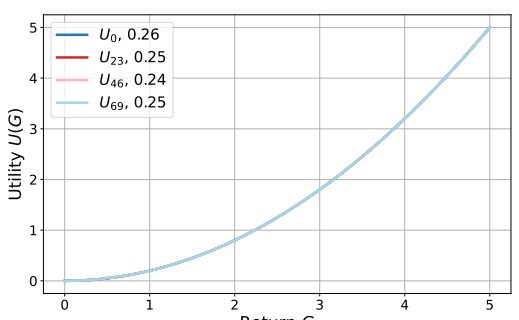
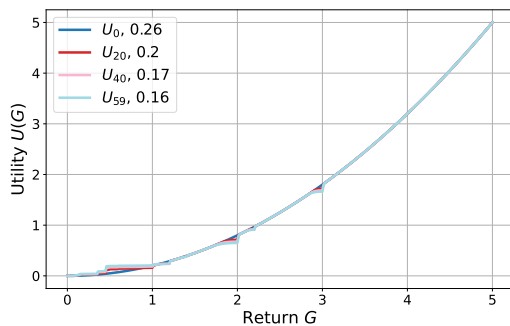

*Figure 18.* (Left) $\alpha = 0.01$. (Right) $\alpha = 0.5$.

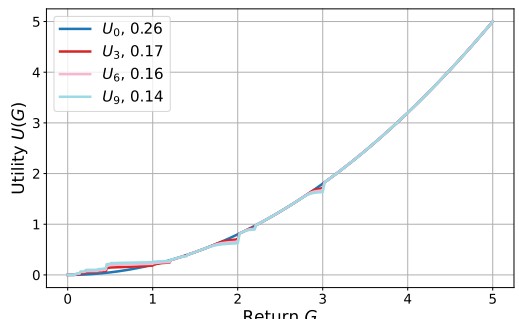
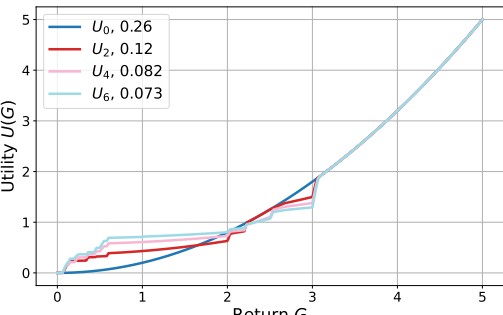

*Figure 19.* (Left) $\alpha = 5$. (Right) $\alpha = 100$.

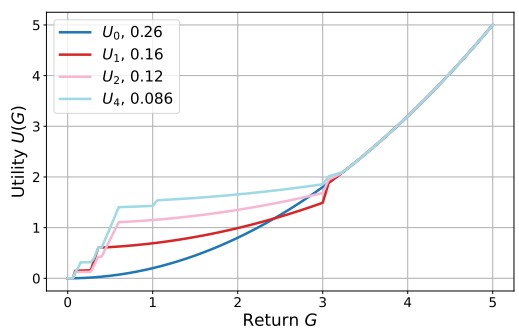 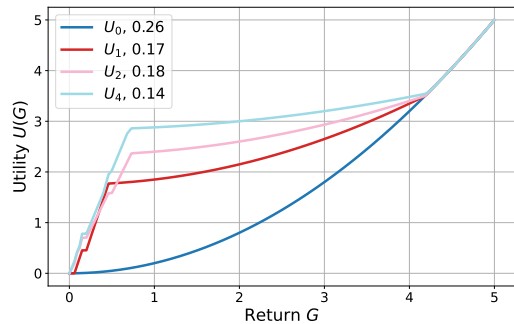

*Figure 20.* (Left) $\alpha = 1000$. (Right) $\alpha = 10000$.

### F.2.2. AN EXPLANATION FOR A LARGE LEARNING RATE

As mentioned in the main paper, there two reasons why a large learning rate is required: $(i)$ the feasible set is large and contains utilities that lie on the boundaries of set $\overline{\mathfrak{U}}_L$,[13] causing larger step sizes to converge sooner; $(ii)$ the projection onto $\overline{\mathfrak{U}}_L$ results in minimal changes of utility even with very large steps.

Now, we show visually that the projection update represented by operator $\Pi_{\overline{\mathfrak{U}}_L}$ crucially neglects small variations in the (non-projected) utilities, requiring us to increase the step size.

Thus, the intuition is that we need a large learning rate because the projection step neglects small variations. To show this, we take as initial utility $\overline{U}_0 = U_{\mathrm{sqrt}}$, two return distributions $\eta_0^*, \eta^E$, where $\eta^*$ coincides with the distribution of an optimal policy for $U_{\mathrm{sqrt}}$, and $\eta^E$ is the return distribution of the policy played by participant 10. These distributions are plotted in Figure 21 left, and their difference is plotted in Figure 21 right. In particular, we note that the two distributions are rather different, with the expert's distribution $\eta^E$ that is more risk-averse, in that it provides higher probability to returns around $G = 0.5$, while the optimal distribution $\eta_0^*$ is more risk-lover, in that it assigns some probability to higher returns $G \geqslant 1$, but suffering from also high probability to small returns $G \leqslant 0.3$.

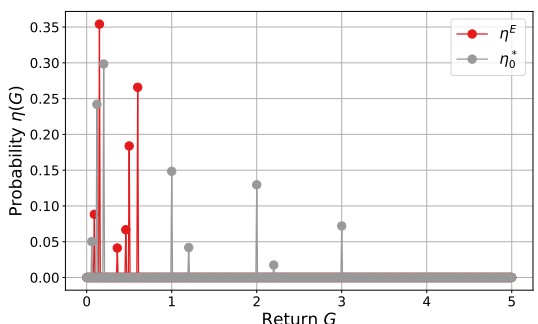 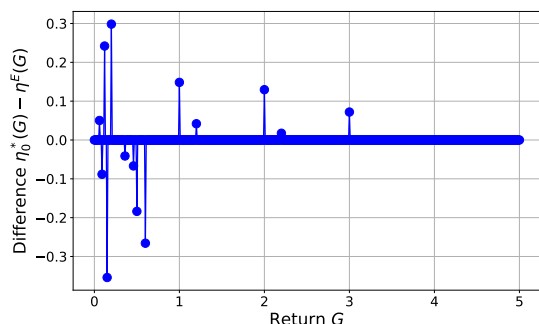

*Figure 21.* (Left) Plot of $\eta_0^*$ and $\eta^E$. (Right) Plot of $\eta_0^* - \eta^E$.

We aim to perform the **TRACTOR-UL** update rule:

$$\overline{U}_1' \leftarrow \overline{U}_0 - \alpha(\eta^* - \eta^E),$$

with some learning rate $\alpha$, and then to perform the projection:

$$\overline{U}_1 \leftarrow \Pi_{\overline{\mathfrak{U}}_L}[\overline{U}_1'].$$

---

[13]$\overline{\mathfrak{U}}_L$ forces utilities to be *increasing*, i.e., with constraints $U(G_1) \leqslant U(G_2) \, \forall G_1 \leqslant G_2$. The plateau in Fig. 17 (right) indicates that $U(G_1) = U(G_2) \, \forall G_1 \leqslant G_2, G_1, G_2 \in [1, 3]$, thus, it represents a boundary.

We execute the update with the following values of steps size: $\alpha \in \{0.01, 0.5, 5, 100, 1000, 10000\}$, and we plot the corresponding updated utilities $\overline{U}'_1$ and $\overline{U}_1$ in Figures 22, 23, and 24.

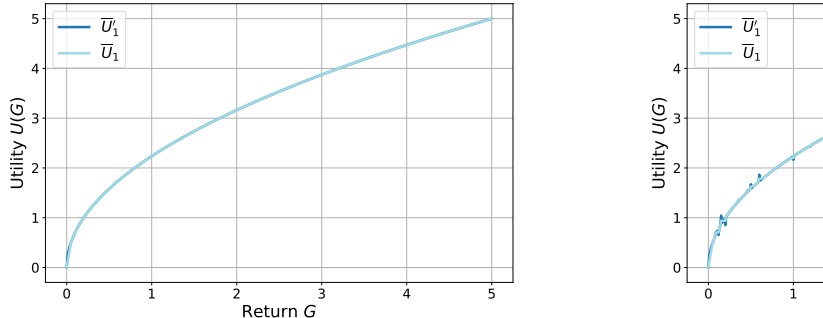
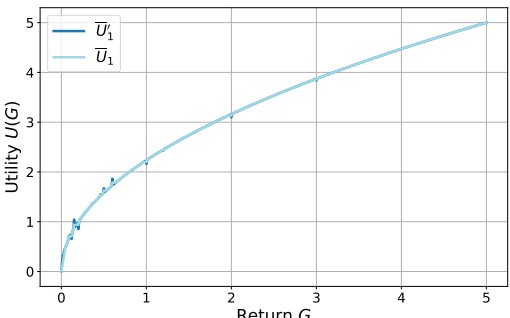

*Figure 22.* (Left) $\alpha = 0.01$. (Right) $\alpha = 0.5$.

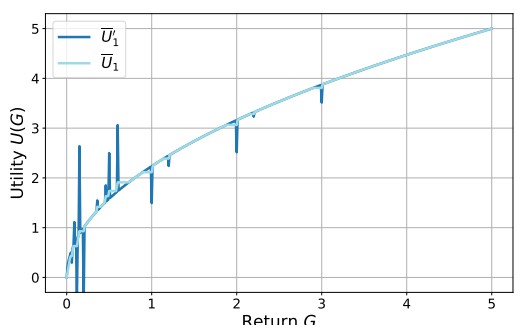
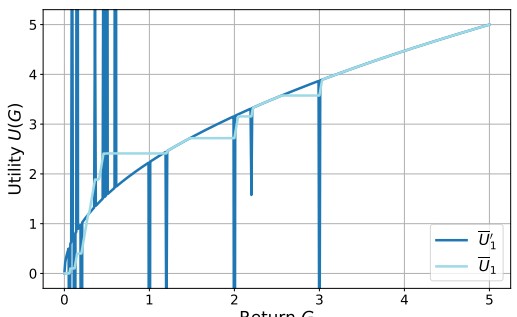

*Figure 23.* (Left) $\alpha = 5$. (Right) $\alpha = 100$.

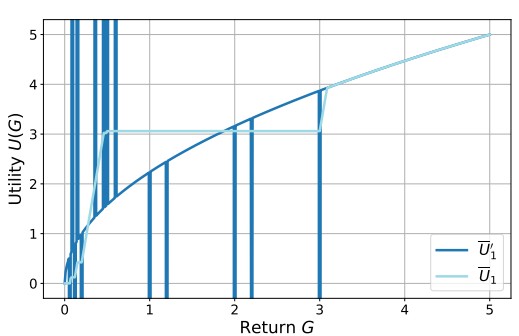
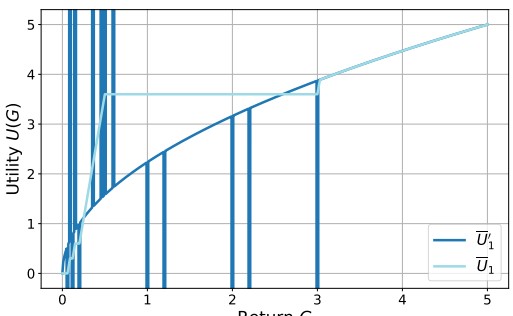

*Figure 24.* (Left) $\alpha = 1000$. (Right) $\alpha = 10000$.

As we can see from Figures 22, 23, and 24, the update $\overline{U}_0 \rightarrow \overline{U}_1$ obtained with step sizes $< 5$ are rather neglectable, so that the return distribution of the new optimal policy $\eta_1^*$ for $\overline{U}_1$ still coincides with the previous one $\eta_0^*$, and the gradient at the next step is the same. For $\alpha = 5$, we begin to notice some changes. See Figure 25.

Instead, with larger gradients, we observe a non-neglectable change in utility, which provides a consistent change in the return distribution for $\alpha = 100$, and a huge change for $\alpha \in [1000, 10000]$ (see Figure 26).

Since neglectable changes in both the utility and the optimal return distribution (obtained with small learning rates) mean that we have to update the utility many times along the same direction, then the update is equivalent to performing a single update in that direction with a huge step size. This justifies the use of large learning rates.

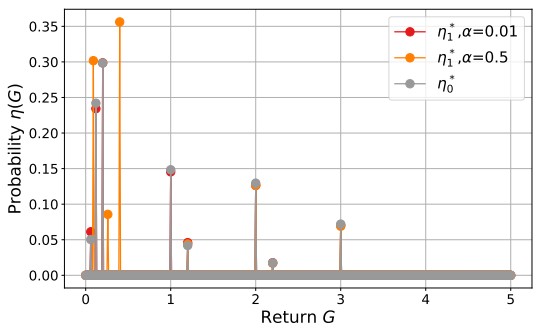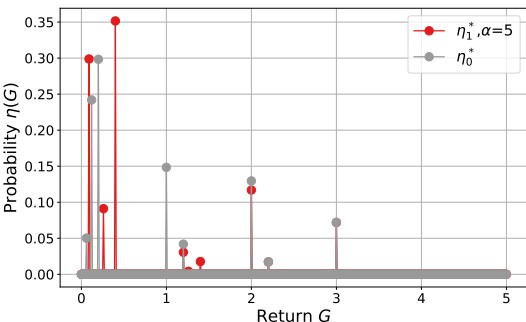

*Figure 25.* (Left) Comparison of the return distributions $\eta_1^*$ obtained with $\alpha = 0.01$ and $\alpha = 0.5$, with $\eta_0^*$. (Right) Comparison of the return distribution $\eta_1^*$ obtained with $\alpha = 5$, with $\eta_0^*$.

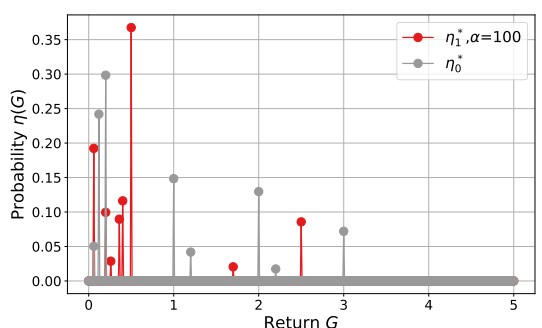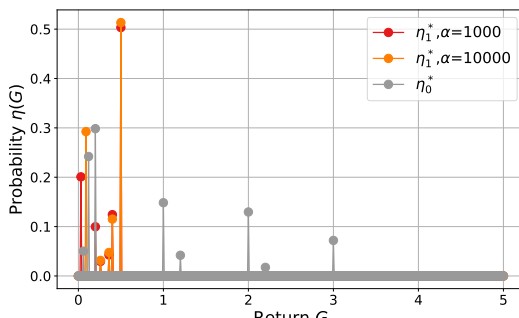

*Figure 26.* (Left) Comparison of the return distribution $\eta_1^*$ obtained with $\alpha = 100$, with $\eta_0^*$. (Right) Comparison of the return distributions $\eta_1^*$ obtained with $\alpha = 1000$ and $\alpha = 10000$, with $\eta_0^*$.

### F.2.3. ANALYSIS ON SIMULATED DATA

We have executed **TRACTOR-UL** on MDPs generated at random. Below (Figures 27-29), we report the truncated (non)compatibility values of the utilities extracted by the algorithm as a function of the number of iterations, in the five different experiments conducted. For the experiments, we executed for $T = 70$ gradient iterations, with parameters $K = 10000$ and $L = 10$, as in the first part of the experiment. We found that the best learning rate is $\alpha = 1$.

To comply with the assumption that there exists a utility function for which the expert's policy is (almost) optimal, we compute, in each environment, the optimal policy for an S-shaped utility function that is convex for small returns, and concave for large returns, and then we inject some noise.

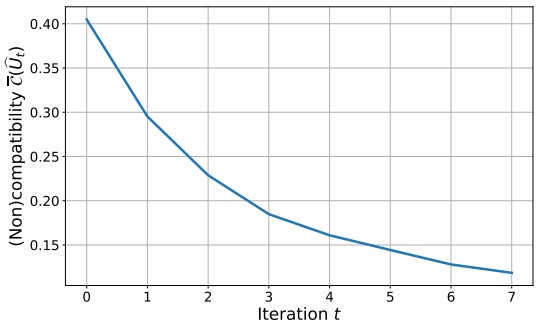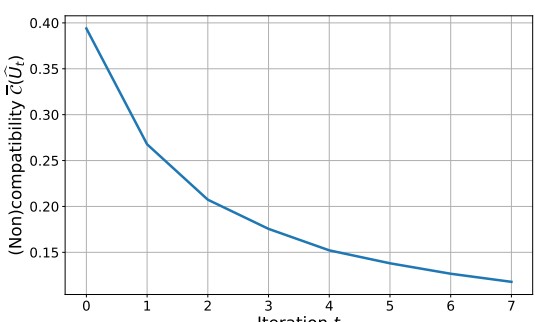

*Figure 27.* (Left) Simulation with $S = 20$ and $A = 5$. (Right) Simulation with $S = 100$ and $A = 10$.

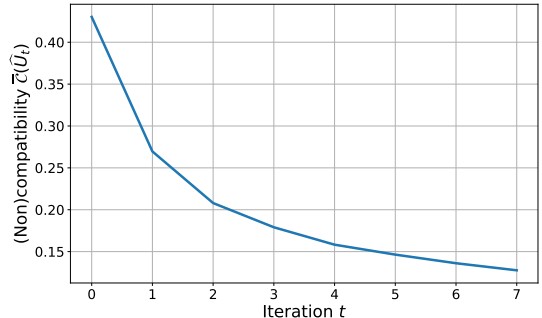 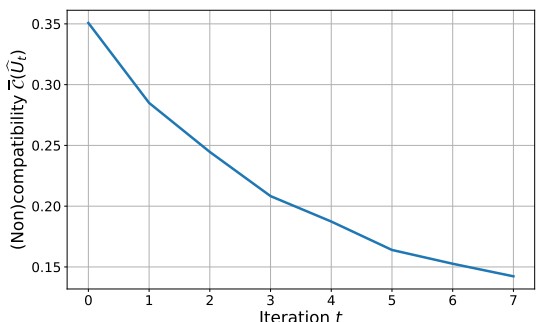

*Figure 28.* (Left) Simulation with $S = 1000$ and $A = 20$. (Right) Simulation with $N = 5$.

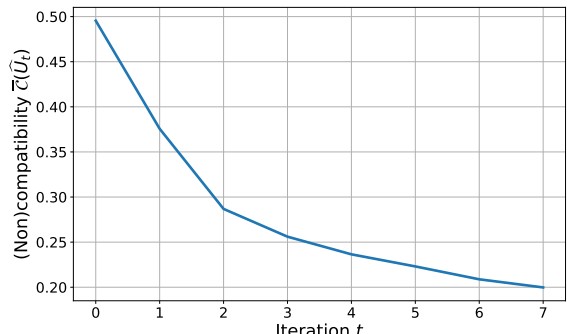

*Figure 29.* Simulation with $N = 20$.

## F.3. Additional Experiment

We conducted an additional experiment using the collected data to understand which utility is more representative of all the participants' behaviors under the model of Eq. (1).

The utilities considered for comparison are: $U_{\text{sqrt}}$, $U_{\text{square}}$, $U_{\text{linear}}$, and $U_{\text{SG}}$. The first three can be formally defined as: $U_{\text{sqrt}}(G) := \sqrt{5G}$, $U_{\text{square}}(G) := G^2/5$, $U_{\text{linear}}(G) := G$, and they are depicted in Figure 30. Instead, utility $U_{\text{SG}}$ differs from each participant and is defined in the next section.

### F.3.1. STANDARD GAMBLE DATA

Utility $U_{\text{SG}}$ corresponds to the utility of each participant as fitted using the *standard gamble method* (Wakker, 2010).

**Standard Gamble (SG).** The Standard Gamble (SG) method (e.g., see Section 2.5 of (Wakker, 2010)) is a common method for inferring the von Neumann-Morgenstern (vNM) utility function of an agent. Observe Figure 32. In a SG, the agent has to decide between two options: a sure option (e.g., $x = 30€$), in which the prize is obtained with probability 1, and a lottery between two prizes (e.g., 5000€ and 0€), in which the best prize (5000€) is received with probability $p$. For any value of $x$, the agent has to answer what is the probability $p$ that, from his perspective, makes the two options (i.e., $x$ for sure, or the lottery) *indifferent*.

Given the probability $p$, we have that the utility $U$ of the agent for $x$ is:

$$U(x) = p \cdot U(5000) + (1 - p) \cdot U(0) = p,$$

since, by normalization conditions, we have $U(0) = 0$ and $U(5000) = 1$.

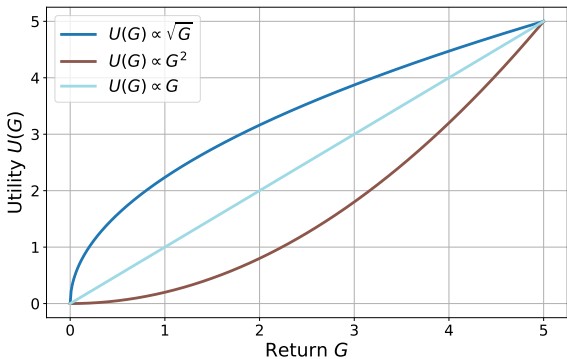

*Figure 30.* A plot of utilities $U_{\text{sqrt}}, U_{\text{square}}, U_{\text{linear}}$.

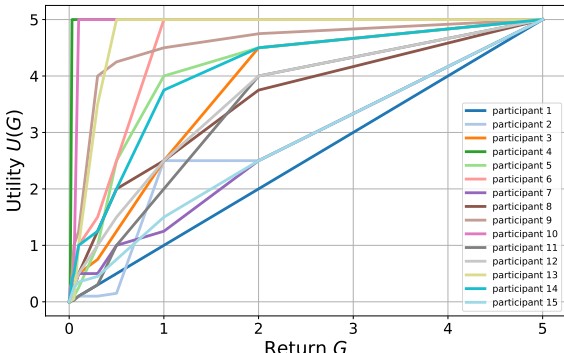

*Figure 31.* The SG utilities of the participants.

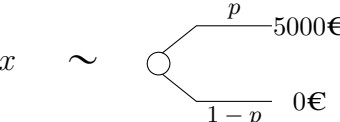

*Figure 32.* The SG used for data collection.

**Our SG.** We have asked the 15 participants to the study to answer some SG questions, which allows us to fit a vNM utility function $U_{\text{SG}}$ for each of them. Specifically, we have asked to answer 8 different SG questions, in which the $x$ value in Figure 32 has been replaced by:

$$10\text{\euro}, 30\text{\euro}, 50\text{\euro}, 100\text{\euro}, 300\text{\euro}, 500\text{\euro}, 1000\text{\euro}, 2000\text{\euro}.$$

Next, we linearly interpolate the computed utilities, obtaining the functions in Figure 31.

It should be remarked that this model considers single decisions (i.e., $H = 1$), while in MDPs there is a sequence of decisions to be taken over time, specifically over a certain time horizon $H$.

F.3.2. RESULTS

To measure the fitness of a utility $U$ to the data (policy $\pi$) *fairly*, we consider a *relative* notion of (non)compatibility (we omit $p, r$ for simplicity): $\overline{\mathcal{C}}^{\text{r}}_{\pi}(U) := (J^*(U) - J^{\pi}(U))/J^*(U)$. Intuitively, $\overline{\mathcal{C}}^{\text{r}}_{\pi}(U)$ measures the *quality* of $\pi$ as perceived by the demonstrating agent, *if $U$ was its true utility function*.

| | 1 | 2 | 3 | 4 | 5 | 6 | 7 | 8 | 9 | 10 | 11 | 12 | 13 | 14 | 15 | mean |
|---|---|---|---|---|---|---|---|---|---|---|---|---|---|---|---|---|
| $U_{\text{linear}}$ | 39 | 58 | 18 | 1 | 9 | 33 | 25 | 62 | 1 | 56 | 1 | 16 | 16 | 25 | 60 | **28±22** |
| $U_{\text{sqrt}}$ | 16 | 28 | 8 | 1 | 3 | 16 | 11 | 30 | 1 | 25 | 1 | 6 | 8 | 11 | 28 | **13±10** |
| $U_{\text{square}}$ | 70 | 86 | 32 | 1 | 19 | 41 | 44 | 91 | 1 | 88 | 1 | 35 | 28 | 44 | 91 | **45±32** |
| $U_{\text{SG}}$ | 39 | 76 | 11 | 0 | 5 | 28 | 20 | 34 | 10 | 2 | 1 | 8 | 21 | 17 | 51 | **22±21** |

*Table 3.* Values of $\overline{\mathcal{C}}_\pi^{\text{r}}$ of various utilities with the demonstrations of the participants in percentage.

We execute `CATY-UL` (without exploration) for the 15 participants comparing the IRL risk-neutral utility $U_{\text{linear}}$ with 3 "baselines": A risk-averse $U_{\text{sqrt}}$ (concave) and a risk-lover $U_{\text{square}}$ (convex) utilities, and the utility $U_{\text{SG}}$ fitted through the SG method (see Appendix F for details). We report the (non)compatibilities in *percentage* in Table 3, where we have used colors to highlight the best and worst values for each participant (the last column contains the average over the participants).

Some observations are in order. First, *this* data shows that $U_{\text{linear}}$ (i.e., IRL) is overcome by $U_{\text{sqrt}}$, which reduces $\overline{\mathcal{C}}_\pi^{\text{r}}(\cdot)$ from 28% to 13% on the average of the participants. Next, note that $U_{\text{sqrt}}$ outperforms the $U_{\text{SG}}$ of *each* participant. This is due to both the bounded rationality of humans, who can *not* apply the $H = 1$ utility $U_{\text{SG}}$ to $H > 1$ problems, and the fact that $U_{\text{sqrt}}$ probably "overfits" the simple MDP considered, but it might generalize worse than $U_{\text{SG}}$ to new environments. [14] Finally, observe that all the utilities are compatible with policies 4 and 11, providing empirical evidence on the *partial identifiability* of the expert's utility from single demonstrations.

The experiment has been conducted on the same personal computer as experiment 2, in less than one hour.

The experiment has been conducted collecting 10000 trajectories to estimate the return distribution of each participant's policy, and 10000 trajectories for estimating the return distribution of the optimal policy, which has been computed exactly through value iteration. We have executed 5 simulations with different seeds, and the relative (non)compatibility values written in Table 3 are the average over the 5 simulations.

For the experiment, we used the true transition model, and we remark that the reward function considered, when discretized, coincides with itself, i.e., we did not incur in estimation error of the transition model nor in approximation error for the discretization.

---

[14]Further analysis should be carried out on this, that we leave to future works.

