# OpenReview forum: "Learning Utilities from Demonstrations in Markov Decision Processes"
_ICML.cc/2025/Conference — ICML 2025 poster_

### Official Review · Reviewer_4S8c · 2025-03-02

**Overall Recommendation:** 4

**Summary:**

This paper considers learning a utility function from demonstration using inverse reinforcement learning and risk sensitive RL. The reward function is assumed to be known. The utility function mapping cumulative rewards to a scalar value is to be inferred. The authors proof the partial identifiability of utility function and improved identifiability with multiple environments. Two algorithms are proposed: 1) CATY-UL is used to classify whether a utility function lies in the compatibility set (i.e., compatible with observed behavior), 2) TRACTOR-UL is used to find a utility function that lies in the compatibility set. Practical implementation uses a discretization based approach for policy evaluation and utility representation and update. Experiments on real and simulated data show TRACTOR-UL's ability to find compatible utility functions.

**Claims And Evidence:**

This paper proposes a method to learn utility functions from demonstration. The results demonstrate the claimed capability.

**Essential References Not Discussed:**

I am not aware of essential references being missed.

**Experimental Designs Or Analyses:**

The experimental design is sound, as the main focus is showing the ability to search compatible utility functions.

A few questions:
* Is the purpose of including human data only to show that human data is not Markovian? Numerical experiment could have been conducted completely in simulation.
* Did the authors demonstrate improved identifiability with more environments, i.e., larger N? Sorry if I missed it but it was hard to find.

**Methods And Evaluation Criteria:**

The proposal to focus on learning utility with known reward in line 201 makes sense. The definition of compatibility in (3) makes sense and is the main driver of the algorithms. Learning from multiple environments to improve identifiability makes sense.

**Other Comments Or Suggestions:**

NA

**Other Strengths And Weaknesses:**

**Weakness**
* This paper is quite difficult to read, perhaps due to a lot of notations. Ultimately the results and algorithms are pretty straightforward. I think the authors could present it in a much easier way.

**Questions For Authors:**

* On line 18 in algorithm 2, what does the $\Pi$ symbol mean? Is it a policy or a projection operator?

**Relation To Broader Scientific Literature:**

This paper address the problem of model risk-sensitivity in inverse RL. Prior work has mainly focused on average return criteria, ignoring the full return distribution. This paper addresses the gap.

**Theoretical Claims:**

I checked the counter examples used to proof partial identification, i.e., proposition 4.1-4.5. I briefly checked the derivations of the sample complexity bounds, i.e., theorem 5.1.

---

> ### Author Rebuttal · Authors · 2025-03-28
>
> We thank the Reviewer for recognizing the validity of our proposal, specifically of learning a utility with known reward and to use demonstrations from multiple environments to reduce identifiability issues. Below, we answer to the Reviewer's comments and questions.
>
> > Is the purpose of including human data only to show that human data is not Markovian? Numerical experiment could have been conducted completely in simulation.
>
> Yes. Since one of the main contributions of the paper is to present the *first* model of human behavior that complies with non-Markovian policies in MDPs (see Eq. (1)), then, we have included human data to provide some empirical evidence on the claim that human behavior is inherently non-Markovian. This is precisely the goal of Experiment 1.
>
> > Did the authors demonstrate improved identifiability with more environments, i.e., larger N? Sorry if I missed it but it was hard to find.
>
> From the *theoretical* viewpoint, our Proposition 4.5 demonstrates that the feasible set of utilities $\mathcal{U}$ reduces its size as demonstrations from multiple environments are observed, up to the limit in which it contains only the expert's utility $\mathcal{U}=\{U^E\}$. Simply put, a policy in an environment represents a constraint in the set of utilities $\mathfrak{U}$, thus, adding more environments we are adding more constraints, and the feasible set $\mathcal{U}$, i.e., the intersection of these subsets, reduces its size.
>
> From an *empirical* perspective, as mentioned in Experiment 2, we have observed that, increasing the number of environments $N$, the empirically-best step size reduces, which may be a symptom that the feasible set $\mathcal{U}$ is smaller, and, thus, we have to be more "precise" for spotting it inside the set of all utilities $\mathfrak{U}$. Anyway, note that Theorem 5.2 holds with a choice of step size $\alpha$ that decreases with $N$ (see line 2728), thus, our previous conjecture may be wrong.
>
> Nevertheless, we remark that an increment of $N$ does not necessary improve the identifiability of the expert's utility $U^E$, although it cannot worsen it, and, thus, this complicates the analysis of the relationship between $N$ and the identifiability of $U^E$. Indeed, intuitively, identifying $U^E$ using demonstrations from $N+M$ environments is "easier" than using only $N$ environments as long as the additional $M$ environments provide constraints that do not already appear in the first $N$ environments. For this reason, analyzing how the identifiability of $U^E$ improves as $N$ increases is not immediate from a technical perspective. Thus, we leave it for future works.
>
> > This paper is quite difficult to read, perhaps due to a lot of notations. Ultimately the results and algorithms are pretty straightforward. I think the authors could present it in a much easier way.
>
> We agree with the Reviewer that the algorithms and the results in Section 5 could be presented using a simpler notation. However, note that the additional notation is necessary for providing a sketch of the proofs of Theorems 5.1 and 5.2 in the main paper. We will try to adjust this trade-off by moving the notation not strictly necessary to the appendix, in order to improve the clarity and the readability of the paper.
>
> > On line 18 in algorithm 2, what does the $\Pi$ symbol mean? Is it a policy or a projection operator?
>
> Yes, $\Pi_{\overline{\underline{\mathfrak{U}}}_L}$ denotes the Euclidean projection onto set $\overline{\underline{\mathfrak{U}}}_L$, as defined on line 083.

---

> > ### Comment · Reviewer_4S8c · 2025-04-03
> >
> > Thank the authors for the responses. I don't have more questions. The paper is solid other than being very dense. I am increasing the score.

---

### Official Review · Reviewer_mjRi · 2025-03-13

**Overall Recommendation:** 3

**Summary:**

This paper introduces a framework for learning utility functions from expert demonstrations inMDPs, where the utility function captures the agent’s risk sensitivity. As the utility may not be identifiable in a single environment, the authors consider learning from demonstrations in multiple environments. The paper proposes two methods and provide sample complexity guarantees for both. Finally, the authors conduct experiments on toy environment (both with real-world participants and simulated).

**Claims And Evidence:**

The claims made in the submission are sufficiently supported.

**Essential References Not Discussed:**

As far as I can tell, related work is appropriately addressed by the authors.

**Experimental Designs Or Analyses:**

I skimmed through the additional experimental details in Appendix F. The experimental design with the participants makes sense, though, it is not clear who the 15 participants are or whether these participants are representative/unbiased.

One concern is the lack of baselines. The authors do not compare against existing methods in their experiments.

**Methods And Evaluation Criteria:**

The conducted experiments are in very small toy environments which seem inspired by simple financial decision-making situations. As a result, the experiments are not fully convincing, but still make sense for the motivation of learning risk-sensitive utility functions.

**Other Comments Or Suggestions:**

1. Even though the paper is overall well-written, it is still quite tedious to parse due to the sometimes seemingly exessive use of notation. It is nearly impossible to keep track of everything and you might want to consider finding ways to reduce the notational burden for the sake of the reader.

**Other Strengths And Weaknesses:**

### Strengths
1. The problem of learning risk-sensitive utility functions is well-motivated and addresses an important limitation of traditional IRL, which typically assumes risk-neutral behavior.
2. The paper answers several fundamental questions about the problem setup, including the identifiability of utility functions.


### Weaknesses
1. The experiments lack comparisons to baselines from existing work on risk-sensitive IRL or utility learning. Could you please clarify whether existing methods are unsuitable for direct comparison or are there other reasons for having no baselines?
2. The experiments are conducted on small toy environments, making it unclear how well the proposed methods scale and whether they can be applied in practice. Additionally, the dependence on the horizon $H$ appears to be a bottleneck and could potentially limiting the practicality of the approach in problems with longer horizons.

**Questions For Authors:**

See above.


**Post-Rebuttal Update**

I maintain my original score and I am still in favor acceptance.

**Relation To Broader Scientific Literature:**

This paper differentiates itself by explicitly modeling the risk attidude of individuals, whereas most prior work on IRL assumes risk-neutral experts (i.e., individuals). It is closely related to the literature on risk-senstive IRL (Majumdar et al. (2017), Singh et al. (2018), Chen et al. (2019), Ratliff & Mazumdar (2020), Cheng et al. (2023)) and primarily differs from the existing work in that it does not assume that the expert is Markovian.

**Theoretical Claims:**

I did not check the correctness of the proofs.

---

> ### Author Rebuttal · Authors · 2025-03-28
>
> We are glad that the Reviewer appreciated the significance of the problem setting considered and the analysis conducted. Below, we report answers to the Reviewer's comments.
>
> > On the experiments conducted.
>
> The ultimate goal of the paper is to introduce a new problem setting, and to characterize its properties and challenges (e.g., identifiability issues). The presentation of algorithms CATY-UL and TRACTOR-UL serves the main objective of demonstrating that, from a theoretical viewpoint, the utility learning problem can be solved efficiently even in the worst case, which is not obvious given the non-Markovianity of the expert's policy. This result paves the way to more practical utility learning algorithms, whose development would be "risky" in absence of theoretical guarantees of polynomial sample and computational complexity (e.g., see Theorems 5.1, 5.2). For these reasons, we believe that, although interesting and important in order to comprehensively corroborate the proposed model and algorithms, an extensive empirical validation that goes beyond the simple tabular settings considered in Section 6 has limited relevance for the scope of this work.
>
> In brief, we stress that since the paper is the first to consider this problem setting and since it already provides significant contributions, we conducted only illustrative experiments in the tabular setting. The development of more complex and practical algorithms able to scale should be conducted in future works.
>
> > On the participants to the experiments.
>
> The 15 participants are **lab members**, as mentioned at the beginning of Section 6 (line 399). We will highlight this fact in the paper.
>
> > On the absence of baselines.
>
> Concerning **Experiment 1**, the objective is to understand which model of human behavior is the most appropriate. Since the main novel feature of the model of behavior that *we* propose in the paper (see Eq. (1)) is the *non-Markovianity* of the expert's policy, then, in the experiment, we compare with the baseline represented by the *Markovian* policy.
>
> The goal of **Experiment 2** is to test the efficiency of TRACTOR-UL at recovering a utility function under the newly-proposed UL problem setting. Since no IRL algorithm in literature, neither the common ones like [1,2,3] nor the risk-sensitive ones like [4,5,6], aims to learn a utility function of this kind, then comparing the efficiency with which existing (risk-sensitive) IRL algorithms converge to their learning targets with the efficiency with which TRACTOR-UL converges to a utility function would not be much meaningful.
>
> > On the dependence on $H$.
>
> We agree with the Reviewer that the dependence on $H^4$ and $\frac{1}{\epsilon^4}$ in the theoretical guarantee of Theorem 5.2, although being polynomial, can be prohibitive in problems with very long horizons for which we require accurate estimates. However, note that this guarantee holds *in the worst case* and using the discretization approach that, while offering the advantage of simplifying the theoretical analysis of the algorithm, might be less efficient in practice than other methods, e.g., estimating the utility of the expert through function approximation and some fixed set of basis functions. We leave this interesting direction to future works.
>
> > On the notation.
>
> We agree with the Reviewer that part of the notation introduced for instance in Section 2 and Section 5 is rather cumbersome and marginal for conveying the main ideas of the paper. However, it is necessary for providing a sketch of the proofs of Theorems 5.1 and 5.2 in the main paper. We will try to adjust this trade-off by moving the notation not strictly necessary to the appendix, in order to improve the clarity and the readability of the paper.
>
> [1] Andrew Y. Ng and Stuart J. Russell. Algorithms for inverse reinforcement learning.
>
> [2] Brian D. Ziebart. Modeling purposeful adaptive behavior with the principle of maximum causal entropy
>
> [3] Deepak Ramachandran and Eyal Amir. Bayesian Inverse Reinforcement Learning.
>
> [4] Lillian J. Ratliff and Eric Mazumdar. Inverse risk-sensitive reinforcement learning.
>
> [5] Sumeet Singh, Jonathan Lacotte, Anirudha Majumdar, and Marco Pavone. Risk-sensitive inverse reinforcement learning via semi- and non-parametric methods.
>
> [6] Haoyang Cao, Zhengqi Wu, and Renyuan Xu. Inference of utilities and time preference in sequential decision-making.

---

### Official Review · Reviewer_ZJB5 · 2025-03-14

**Overall Recommendation:** 3

**Summary:**

The paper introduces a new risk-sensitive model for inverse reinforcement learning  in MDPs, explicitly accounting for the non-Markovian policies induced by risk-sensitive utility functions. The main contributions include formulating the Utility Learning problem to learn an agent's risk attitude, characterizing its partial identifiability, and proposing two algorithms, CATY-UL and TRACTOR-UL, for efficient utility learning from finite demonstrations. The authors validate their methods through theoretical analysis and proof-of-concept experiments.

**Claims And Evidence:**

The paper provides strong theoretical justifications for its claims, with formal propositions  showing the limitations of identifiability in the single-environment setting. The propositions supporting the value of multi-environment data (Proposition 4.5) are correct but somewhat weak, as they confirm possibility rather than providing explicit complexity or a lower bound on the number of required environments.

**Essential References Not Discussed:**

Essential references related to risk-sensitive MDPs, particularly Online Risk-sensitive MDP, including ERM and CVaR-based, were not fully cited or adequately discussed. Incorporating such references would improve clarity and contextualize this work further within the broader risk-sensitive reinforcement learning literature. To name a few

Online ERM-MDP
- Fei, Y., Yang, Z., Chen, Y., Wang, Z., and Xie, Q. Risk-sensitive reinforcement learning: near-optimal risksample tradeoff in regret. In Proceedings of the 34th International Conference on Neural Information Processing Systems, NIPS ’20, Red Hook, NY, USA, 2020. Curran Associates Inc. ISBN 9781713829546.
- Fei, Y., Yang, Z., Chen, Y., and Wang, Z. Exponential bellman equation and improved regret bounds for risksensitive reinforcement learning. Advances in neural information processing systems, 34:20436–20446, 2021.
- Liang, H., & Luo, Z. Q. (2024). Bridging distributional and risk-sensitive reinforcement learning with provable regret bounds. Journal of Machine Learning Research, 25(221), 1-56.

Online CVaR-MDP
- Bastani, O., Ma, J. Y., Shen, E., & Xu, W. (2022). Regret bounds for risk-sensitive reinforcement learning. Advances in Neural Information Processing Systems, 35, 36259-36269.
- Wang, K., Kallus, N., and Sun, W. Near-minimax-optimal risk-sensitive reinforcement learning with cvar. In International Conference on Machine Learning, pp. 3586435907. PMLR, 2023.
- Wang, K., Liang, D., Kallus, N., and Sun, W. Risk-sensitive rl with optimized certainty equivalents via reduction to standard rl. arXiv preprint arXiv:2403.06323, 2024.

**Experimental Designs Or Analyses:**

The experimental results offers simple proof-of-concept demonstrations with real human data, validating the non-Markovian nature of human decision-making. However, the experiments are preliminary and somewhat limited in scope, particularly lacking evaluation in larger-scale or varied environments.

**Methods And Evaluation Criteria:**

The methods and evaluation criteria chosen (partial identifiability, regret, and compatibility metrics) make sense and are relevant to the problem of learning risk-sensitive utilities. The paper provides both bounds on the feasibility of learned utilities and empirical validations.

**Other Comments Or Suggestions:**

- Clarify and simplify Section 2’s notation, as some elements appear dense and are not essential for the main text.
- More clearly state computational complexity and practical scalability concerns.

**Other Strengths And Weaknesses:**

Strengths:
- Novel problem formulation clearly motivated by real-world risk-sensitive behaviors.
- Strong theoretical contributions on identifiability.
- Clear, well-written manuscript.

Weaknesses:
- Limited empirical validation; experiments are preliminary and conducted only in simple settings.
- Ambiguity in demonstrating the complexity and requirements for multi-environment utility identifiability.
- Missing key discussions and comparisons with related risk-sensitive formulations (ERM, CVaR).

**Questions For Authors:**

Please see the above concerns/weakness. In addition
- Proposition 4.5 currently states only the possibility of unique identifiability. Can you provide insight into how many environments typically might be required for practical identifiability?

**Relation To Broader Scientific Literature:**

The paper situates itself within inverse reinforcement learning and risk-sensitive MDP literature, highlighting how it generalizes previous IRL models (Ng & Russell, 2000) and connects to expected utility theory. However, the authors should more explicitly discuss connections to the broader literature on risk-sensitive MDPs, including Expected Risk Measure (ERM) MDPs and Conditional Value-at-Risk (CVaR) MDPs, which are not sufficiently discussed. Some references related to Maximum Entropy IRL (Ziebart, 2010) seems to be missing as well.

**Theoretical Claims:**

The theoretical claims, particularly the identifiability results  and regret bounds for algorithms, appear sound. I also checked the correctness of Proposition 4.1, Proposition 4.2, and Theorem 5.1; no issues were identified in the proofs. However, the complexity of the augmented state space approach is not well analyzed.

---

> ### Author Rebuttal · Authors · 2025-03-28
>
> We thank the Reviewer for recognizing the novelty of the proposed problem formulation, and the strength of our theoretical contributions on the identifiability problem of utilities. Below, we answer to the Reviewer's comments and questions.
>
> > On the comparison with literature.
>
> We thank the Reviewer for the references. We will incorporate a discussion on both ERM-MDPs and CVaR-MDPs in the paper. Although the primary focus of our work is the *inverse* problem, we agree with the Reviewer that including a discussion on algorithms for solving the *forward* problem can improve the quality and clarity of the paper.
>
> > On the empirical validation.
>
> The ultimate goal of the paper is to introduce a new problem setting, and to characterize its properties and challenges (e.g., identifiability issues). The presentation of CATY-UL and TRACTOR-UL serves the main objective of demonstrating that, from a theoretical viewpoint, the UL problem can be solved efficiently even in the worst case, which is not obvious given the non-Markovianity of the expert's policy. This result paves the way to more practical UL algorithms, whose development would be "risky" in absence of theoretical guarantees of polynomial sample and computational complexity (see Theorems 5.1, 5.2). For these reasons, we believe that, although interesting, an extensive empirical validation of the proposed algorithms that goes beyond the simple tabular settings considered in Section 6 has limited relevance for the scope of this work.
>
> In brief, we stress that, since the paper is the first to consider this problem setting and since it already provides significant contributions, then we conducted only illustrative experiments in the tabular setting. The development of more complex and practical algorithms able to scale should be conducted in future works.
>
> > On the notation.
>
> We agree with the Reviewer that part of the notation introduced in Section 2 is rather cumbersome and marginal for conveying the main ideas of the paper. However, it is necessary for providing a sketch of the proofs of Theorems 5.1 and 5.2 in the main paper. We will try to adjust this trade-off by moving the notation not strictly necessary to the appendix, in order to improve the clarity and the readability of the paper.
>
> > On the computational complexity.
>
> For the subroutines:
>
> - EXPLORE: *time* = $\mathcal{O}(N\tau)$; *space* = $\mathcal{O}(SAHN)$.
> - ERD: *time* = $\mathcal{O}(H\tau^E+H/\epsilon_0)$; *space* = $\mathcal{O}(H/\epsilon_0)$.
> - PLANNING: *time* = $\mathcal{O}(S^2AH^2/\epsilon_0)$; *space* = $\mathcal{O}(SAH^2/\epsilon_0)$.
> - ROLLOUT: *time* = $\mathcal{O}(KH)$; *space* = $\mathcal{O}(K)$.
>
> Thus:
> - **CATY-UL**: *time* = $\mathcal{O}(N\tau+MN(H\tau^E+S^2AH^2/\epsilon_0))$, where $M$ denotes the number of input utilities to which CATY-UL is applied; *space* = $\mathcal{O}(SAHN+SAH^2/\epsilon_0)$.
> - **TRACTOR-UL**: *time* = $\mathcal{O}(N\tau+NH\tau^E+T(NS^2AH^2/\epsilon_0+NKH+Q_{time}))$, where $Q_{time}$ represents the number of iterations of the optimization solver adopted for the Euclidean projection; *space* = $\mathcal{O}(NH/\epsilon_0+SAH^2/\epsilon_0+K+Q_{space})$, where $Q_{space}$ is the space used for the projection.
>
> Note that these complexities are polynomial in all the quantities of interest $S,A,H$, $N,$ $\frac{1}{\epsilon}$, $\log\frac{1}{\delta},Q_{time},Q_{space}$, and this holds even if we replace $\epsilon_0,\tau^E,\tau,T,K$ with the values that provide the theoretical guarantees in Theorems 5.1, 5.2.
>
> We remark that the assumption made in Theorem 5.2 that the Euclidean projection is *exact* is made just for simplifying the theoretical analysis, but it is *not necessary*. If the Reviewer desires details on the proof, we can provide them.
>
> We will add all these considerations to the paper.
>
> >  On the multi-environment utility identifiability.
>
> This is an interesting point. Intuitively, it is not trivial to compute a minimum number of environments $N\ge\overline{N}$ that suffices for the "practical" identifiability of the expert's utility $U^E$, for two reasons:
> - It depends on how much "*informative*" are the observed $N$ environments. For instance, if the $N$ environments provide similar constraints in the space of utilities $\mathfrak{U}$, then identifying $U^E$ remains difficult.
> - Depending on what we want to do with the expert's utility $U^E$ once that we have recovered it, we might be satisfied with less accurate estimates. For instance, if we want to recover $U^E$ for transferring it to a difficult environment, then, intuitively, we need a very good estimate $\widehat{U}\approx U^E$, because the target environment is difficult, and so we expect $N$ to be large; instead, if the goal is to do planning in a simple environment, then we can tolerate $N$ to be small.
>
> For these reasons, analyzing how the identifiability of $U^E$ improves as $N$ increases is not immediate from a technical perspective and we leave it for future works.

---

> > ### Comment · Reviewer_ZJB5 · 2025-04-04
> >
> > Thank the authors for their detailed response and clarifications, which address most of my concern. I maintain my positive evaluation for this paper.

---

### Official Review · Reviewer_HXF4 · 2025-03-14

**Overall Recommendation:** 4

**Summary:**

The paper considers a specific type of risk-sensitive MDPs where the risk sensitivity is captured via a continuous, strictly increasing utility function. Given a set of optimal expert demonstrations and the expert reward, the aim is to recover the expert's utility function. The authors first provide a few impossibility results, showing that similar to the IRL, the utility learning problem is ill-posed in general. However, they show that if given access to all deterministic optimal policies under any transition kernel, we could identify the expert's utility function. Motivated by this observation, they provide two algorithms for learning from expert data collected in multiple environments: CATY-UL, a classification-based method to check the approximate compatibility of a given utility, and TRACTOR-UL, a more practical algorithm that outputs a single candidate utility, much like traditional IRL methods. Finally, the applicability of TRACTOR-UL is showcased on a tabular toy problem.

**Claims And Evidence:**

Generally, the claims are clear, and complete proofs are provided in the appendix.

**Essential References Not Discussed:**

The authors stress the ability of their approach to induce non-Markovian behavior. The claim on line 409 that this is the first IRL model to induce non-Markovian behavior is too strong. Majumdar et al. (2017) already address IRL with coherent risk measures such as CVaR, which leads to non-Markovian policies (Chow, 2017). Moreover, most risk measures, except for ERM, EVaR, and time-consistent ones [Chow, Theorem 1.3.8], inherently yield non-Markovian policies.

- Majumdar, Anirudha, et al. "Risk-sensitive Inverse Reinforcement Learning via Coherent Risk Models." Robotics: science and systems. Vol. 16. 2017.
- Chow, Yinlam. Risk-sensitive and data-driven sequential decision making. Diss. Stanford University, 2017.

**Experimental Designs Or Analyses:**

The experimental validation is limited to a toy problem, which may have limited practical relevance. Nevertheless, the results are well-documented, and the authors considered human expert data.

**Methods And Evaluation Criteria:**

The problem definition, the algorithms, and the theoretical results are clearly presented and consistent.

**Other Comments Or Suggestions:**

Typos: in line 197, 281, 381: contained into -> contained in

**Other Strengths And Weaknesses:**

Strengths:
- The paper introduces an interesting problem setting by focusing on learning the utility function for a known reward.
- The theoretical contribution providing both identifiability results and convergence guarantees is solid.

Weaknesses:
- The paper is quite notation-heavy, which can make it difficult to follow. If you can reduce the density of math symbols in the main text, that could significantly improve the overall readability.

**Questions For Authors:**

When learning from a single environment, Theorem 5.2 basically guarantees that for the recovered utility, the expert is $\varepsilon$ optimal. Is it possible to also say something in terms of Hausdorff distance to the set of feasible utilities?

**Relation To Broader Scientific Literature:**

While IRL in the risk-neutral setting has been addressed extensively, and some risk-sensitive approaches exist, the specific setting of identifying the utility given the reward seems to be novel.

**Theoretical Claims:**

I reviewed the proofs of Proposition 4.5 and Theorem 5.2, and they seemed to be sound.

---

> ### Author Rebuttal · Authors · 2025-03-28
>
> We are glad that the Reviewer appreciated the novelty and the significance of the problem setting introduced, and that the Reviewer recognized the solidity of the theoretical results presented. Below, we report answers to the Reviewer's comments.
>
> > The authors stress the ability of their approach to induce non-Markovian behavior. The claim on line 409 that this is the first IRL model to induce non-Markovian behavior is too strong. Majumdar et al. (2017) already address IRL with coherent risk measures such as CVaR, which leads to non-Markovian policies (Chow, 2017). Moreover, most risk measures, except for ERM, EVaR, and time-consistent ones [Chow, Theorem 1.3.8], inherently yield non-Markovian policies.
>
> We agree with the Reviewer that the claim on line 409 is imprecise, and we will change it to "... the *first IRL model that contemplates non-Markovian policies **in MDPs***". Indeed, as we explain in Section 7, Majumdar et al. (2017) consider the much simpler "prepare-react model" as environment intead of an MDP.
>
> Also, we note that, even though most risk measures yield non-Markovian policies, there is no IRL algorithm for MDPs in the literature that models the expert's policy as the result of the optimization of a risk measure, and, as such, as non-Markovian. Indeed, as mentioned in Section 7 and explained in Appendix A, works like [1] model the expert's policy as Boltzmann rational, i.e., as stochastic Markovian.
>
> > The paper is quite notation-heavy, which can make it difficult to follow. If you can reduce the density of math symbols in the main text, that could significantly improve the overall readability.
>
> We agree with the Reviewer that the main text is notationally dense. We will try to simplify some passages to improve the readability in the final version of the paper, moving the notation not strictly necessary in the main paper to the appendix.
>
> > Typos: in line 197, 281, 381: contained into -> contained in
>
> We thank the Reviewer for pointing out, we have fixed them.
>
> > When learning from a single environment, Theorem 5.2 basically guarantees that for the recovered utility, the expert is optimal. Is it possible to also say something in terms of Hausdorff distance to the set of feasible utilities?
>
> Answering to this question is not trivial. The reason is that, for the feasible utility set, it is not immediate to find an explicit representation for the feasible utility set $\mathcal{U}$ that permits to construct an estimator $\widehat{\mathcal{U}}$ for which it is simple to carry out a sample complexity analysis.
>
> Consider the sample complexity analysis conducted for the estimation of the feasible reward set [2,3,4,5]. In [2,3], it is proved that each reward $r$ in the feasible reward set $\mathcal{R}$ can be parameterized as a function of the optimal value $V_h$ and advantage $A_h$ functions as:
> $$
> r_h(s,a)=V_h(s)-\sum\limits_{s'\in\mathcal{S}}p_h(s'|s,a)
> V_{h+1}(s')+1\\{\pi_h^E(s)=a\\}A_h(s,a).\qquad (1)
> $$
> Thus, using as estimator for $\mathcal{R}$ the set $\widehat{\mathcal{R}}$ where each reward $\widehat{r}\in\widehat{\mathcal{R}}$ can be parametrized as a function of the optimal value $\widehat{V}_h$ and advantage $\widehat{A}_h$ functions in the estimated MDP as:
>
> $$
> \widehat{r}\_h(s,a)=\widehat{V}\_h(s)-\sum\limits\_{s'\in\mathcal{S}}\widehat{p}\_h(s'|s,a) \widehat{V}\_{h+1}(s')+1\\{\widehat{\pi}\_h^E(s)=a\\}\widehat{A}\_h(s,a),
> $$
>
> then it is possible to bound the Hausdorff distance in max norm by the 1-norm of the transition models:
> $$
> \mathcal{H}(\mathcal{R},\widehat{\mathcal{R}})\le \max_{s,a,h}\\|\widehat{p}\_h(\cdot|s,a)-p_h(\cdot|s,a)\\|_1,
> $$
>
> as long as $\widehat{\pi}^E=\pi^E$ with high probability (see also [4,5]). Then, the analysis follows rather directly by applying standard concentration inequalities.
>
> To adapt this analysis to sets of utilities, we require a simple representation of the feasible utility set $\mathcal{U}$ analogous to that in (1). However, it is not immediate to us how to obtain it, since utilities are objects rather different from reward functions. Nevertheless, studying the feasible utility sets and the problem of learning them is an interesting future research direction.
>
> [1] Ratliff, L. J. and Mazumdar, E. Inverse risk-sensitive reinforcement learning.
>
> [2] Metelli, A. M., Ramponi, G., Concetti, A., and Restelli, M. Provably efficient learning of transferable rewards.
>
> [3] Lindner, D., Krause, A., and Ramponi, G. Active exploration for inverse reinforcement learning.
>
> [4] Metelli, A. M., Lazzati, F., and Restelli, M. Towards theoretical understanding of inverse reinforcement learning.
>
> [5] Zhao, L., Wang, M., and Bai, Y. Is inverse reinforcement learning harder than standard reinforcement learning?

---

### Decision · Program_Chairs · 2025-05-01

**Decision:**

Accept (poster)

**Comment:**

This paper considers learning utilities from demonstrations for risk-sensitive agents. A utility function is introduced that transforms rewards/returns of a trajectory to a real-valued utility for the agent. When this function is linear, classical inverse reinforcement learning methods result. When the function is nonlinear, risk-sensitivity is introduced. The paper specifically focuses on learning this utility function given the reward function is known/given. Identifiability is characterized for this task, and this characterization is used to guide the development of two algorithms: CATY-UL and TRACTOR-UL and provide convergence guarantees. The approach is evaluated on a simple tabular MDP.

This paper provides a good formulation and perspective for learning risk-sensitivity, and provides strong theoretical analysis. The empiricial demonstrations are weaker and it is not clear that the approach will be applicable to practical problems of much larger scale.

I dislike the title. It makes sense based on the authors' perspective, but would seem to imply learning both r^E and U^E without additional qualification: e.g., "Learning Risk-Sensitive Utilities from Demonstrations and Rewards in Markov Decision Processes" might be more appropriate.

Overall, I concur with the reviewers that this is a solid theoretical paper with a useful perspective and recommend acceptance.